# AnyTouch 2: General Optical Tactile Representation Learning For Dynamic Tactile Perception

**Ruoxuan Feng**[1,2,3]    **Yuxuan Zhou**[4]    **Siyu Mei**[4]    **Dongzhan Zhou**[5]    **Pengwei Wang**[3]
**Shaowei Cui**[6,3]    **Bin Fang**[7,3]    **Guocai Yao**[3,8]    **Di Hu**[1,2,3,†]

[1]Gaoling School of Artificial Intelligence, Renmin University of China, Beijing, China
[2]Beijing Key Laboratory of Research on Large Models and Intelligent Governance
[3]Beijing Academy of Artificial Intelligence    [4]Beijing Jiaotong University
[5]Shanghai AI Laboratory    [6]Institute of Automation, Chinese Academy of Sciences
[7]Beijing University of Posts and Telecommunications
[8]State Key Laboratory of Multimedia Information Processing, Peking University

## Abstract

Real-world contact-rich manipulation demands robots to perceive temporal tactile feedback, capture subtle surface deformations, and reason about object properties and force dynamics. Although optical tactile sensors are uniquely capable of providing such rich information, existing tactile datasets and models remain limited. These resources primarily focus on object-level attributes (*e.g.*, material) while largely overlooking fine-grained temporal dynamics. We consider that advancing dynamic tactile perception requires a systematic hierarchy of dynamic perception capabilities to guide both data collection and model design. To address the lack of tactile data with rich dynamic information, we present **ToucHD**, a large-scale tactile dataset spanning tactile atomic actions, real-world manipulations, and touch-force paired data. Beyond scale, ToucHD establishes a comprehensive dynamic data ecosystem that explicitly supports hierarchical perception capabilities from the data perspective. Building on it, we propose **AnyTouch 2**, a general tactile representation learning framework for diverse optical tactile sensors that unifies object-level understanding with fine-grained, force-aware dynamic perception. The framework captures both pixel-level and action-specific deformations across frames, while explicitly modeling physical force dynamics, thereby learning multi-level dynamic perception capabilities from the model perspective. We evaluate our model on benchmarks that covers static object properties and dynamic physical attributes, as well as real-world manipulation tasks spanning multiple tiers of dynamic perception capabilities—from basic object-level understanding to force-aware dexterous manipulation. Experimental results demonstrate consistent and strong performance across sensors and tasks, highlighting the framework's effectiveness as a general dynamic tactile perception model. The code, dataset and model are available at gewu-lab.github.io/AnyTouch2/.

## 1 Introduction

Tactile perception is a cornerstone of human interaction with the physical world, providing rich contact information that complements vision and audition. It enables fine-grained understanding of subtle deformations and force dynamics that are essential for various contact-rich tasks (Heng et al., 2025; Feng et al., 2025a; Xue et al., 2025; Iskandar et al., 2024). With the rapid progress of high-resolution optical tactile sensors (Lambeta et al., 2024; Zhao et al., 2025a), robotics is poised to enter a new era of *dynamic tactile perception*, where robots will be able to perceive temporal variations in contact, force, and material interactions to accomplish increasingly complex real-world tasks.

---

† Corresponding author.

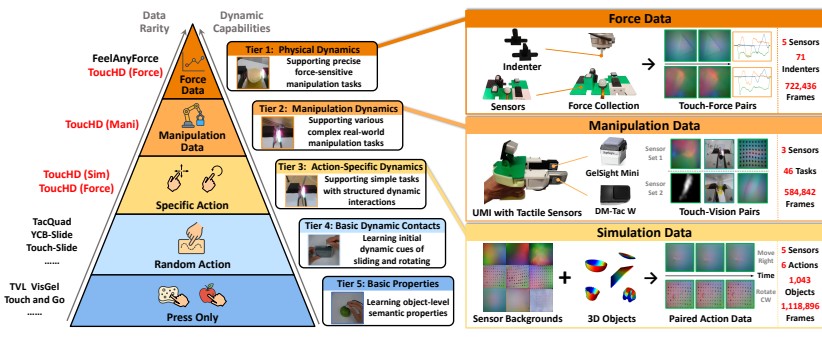

Figure 1: **Tactile Dynamic Pyramid and TouHD dataset.** We organize tactile pre-training data into 5 tiers based on data rarity and the complexity of the dynamic perception capabilities they support. Datasets shown in black font are existing ones. Most current datasets fall into the lower tiers (4 and 5), while higher tiers (1, 2, and 3) remain notably scarce. To bridge this gap, we present TouHD, a hierarchical dynamic tactile dataset spanning tactile atomic actions, real-world manipulations, and touch–force paired data. TouHD is designed to enrich high-tier data and establish a complete dynamic data ecosystem, thereby comprehensively supporting dynamic tactile perception.

In stark contrast, existing tactile datasets and models remain largely limited to static object-level properties, due to the absence of a systematic perspective on dynamic tactile perception, thereby overlooking the rich temporal dynamics of touch and the underlying force-related physical principles. Many large-scale datasets primarily rely on press-only actions to collect material properties like texture and hardness (Yang et al., 2022; Fu et al., 2024), with limited extensions to random sliding or rotation (Suresh et al., 2023; Higuera et al., 2025a; Feng et al., 2025b). A recent press-based touch–force dataset (Shahidzadeh et al., 2025) provides preliminary physical grounding but still lacks richer dynamic interactions. Similarly, mainstream tactile pre-training models, often adapted from image-based self-supervised (He et al., 2022) or multi-modal alignment frameworks (Radford et al., 2021), struggle to capture fine-grained deformations and force-aware dynamics. Deficiencies in both datasets and models for supporting dynamic perception capabilities required by complex tasks ultimately limit the effectiveness of tactile pre-training in manipulation (Luu et al., 2025).

To establish a systematic paradigm for dynamic tactile perception, we first introduce a *tactile dynamic pyramid* that organizes tactile data into five tiers based on the complexity level of the perception capabilities they support, as shown in Fig. 1. Most existing datasets reside at the lowest Press Only and Random Action tiers, offering limited action diversity and supporting only static attributes or shallow surface-level dynamics. In contrast, higher tiers, though far more challenging to collect, enable richer perception capabilities: Specific Action data facilitate learning structured tactile dynamic semantics, Manipulation data capture temporally evolving contact patterns crucial for dexterous skills, and Force data explicitly ground tactile dynamics in physical force properties. To fill this critical gap, we introduce **TouHD**, a large-scale dataset with 2,426,174 contact samples, designed as a **T**actile **H**ierarchical **D**ynamic resource to enrich the higher tiers. By incorporating diverse tactile sensors and techniques, TouHD integrates simulated atomic action data, real-world manipulation data collected with a modified FastUMI (Wu et al., 2024), and extensive touch–force pairs obtained from 71 indenters. Together, these hierarchical components form a systematic dynamic data architecture that provides broad diversity in objects, sensors, and contacts, and establishes a comprehensive foundation for advancing dynamic tactile perception across all tiers.

Building on this foundation, we introduce **AnyTouch 2**, a general tactile representation learning framework that unifies sensor-invariant object properties understanding with progressively enhanced perception of fine-grained deformations, action-specific dynamics, and force-related physical properties. Beyond masked video reconstruction, multi-modal alignment, and cross-sensor matching, we incorporate multi-level modules to advance dynamic tactile perception along the hierarchical capabilities outlined by our dynamic pyramid. Concretely, we enhance sensitivity to subtle temporal deformations via frame-difference reconstruction, promote semantic-level action understanding through action matching, and model the underlying physical properties by predicting temporal force variations from large-scale touch–force pairs. Collectively, these components yield a unified representation that bridges object-level semantics, dynamic interaction modeling, and physical reasoning across all tiers, offering a solid foundation for diverse downstream tasks.

We evaluate AnyTouch 2 on benchmarks spanning static object properties, dynamic physical prediction, and real-world manipulation tasks across all tiers of the tactile dynamic pyramid. Experimental results show that our approach delivers consistently strong performance across both static and dynamic tactile perception tasks, validating its effectiveness as a general tactile representation framework. By grounding our framework in the tactile dynamic pyramid, we hope this work lays a solid foundation for advancing the era of dynamic tactile perception and inspires future research toward more dexterous, physically grounded robotic intelligence.

## 2 RELATED WORK

**Large-Scale Tactile Dataset.** Early tactile datasets were typically collected via handheld or robotic pressing, focusing on object-level semantic properties such as material and hardness (Yuan et al., 2018; Li et al., 2019; Yang et al., 2022; Gao et al., 2023; Fu et al., 2024). These press-only datasets exhibit limited dynamic variation and primarily support learning static tactile features. Some datasets expand this paradigm by applying simple random actions on object surfaces to capture basic dynamic interactions (Suresh et al., 2023; Yu et al., 2024; Higuera et al., 2025a; Feng et al., 2025b). While such data can help models gain an initial understanding of tactile dynamics, they remain insufficient for supporting complex dynamic tasks like dexterous manipulation. Luu et al. (2025) collected a touch–force paired dataset by pressing sensors with different indenters, offering initial insight into physical contact properties, but the dataset still lacks richer dynamics like sliding or rotation. In this work, we collect the largest hierarchical dynamic tactile dataset to address the scarcity of high-tier tactile data with rich dynamic interactions and paired force measurements.

**Optical Tactile Representation Learning.** Optical tactile sensors can capture high-resolution spatio-temporal deformations of contact surfaces, enabling fine-grained perception of object properties and interaction dynamics. Leveraging the image-based nature of optical tactile data, recent studies have explored leveraging vision-related representation learning, using visual self-supervised learning methods (He et al., 2022) for fine-grained feature learning (Xu et al., 2025; Zhao et al., 2025b; Higuera et al., 2025a) and multi-modal alignment with vision and language for semantic-level understanding (Yang et al., 2024; Cheng et al., 2025; Ma et al., 2025; Feng et al., 2025b). To handle sensor heterogeneity, some works employ joint training (Zhao et al., 2025b), alignment (Yang et al., 2024; Gupta et al., 2025), or cross-sensor matching (Feng et al., 2025b). More recent works have explored dynamic tactile representation learning by transferring self-supervised video learning techniques (Higuera et al., 2025a; Feng et al., 2025b; Xie et al., 2025), allowing models to capture temporal deformation patterns. In this work, we unify the strengths of previous methods by integrating object-level feature understanding with hierarchical dynamic tactile perception capabilities, resulting in a general tactile representation capable of supporting a variety of downstream tasks.

**Dynamic Tactile Perception.** While early tactile models primarily focused on static object-level properties, real-world contact-rich manipulation requires perceiving the temporal tactile dynamics and reasoning about underlying physical principles (Xue et al., 2025; Higuera et al., 2025b). Recent studies have begun to explore dynamic tactile perception in both real and simulated environments. A common approach adapts visual models to process continuous tactile inputs and model temporal variation, but often without tailoring them to the unique characteristics of tactile data (Feng et al., 2025a; Hao et al., 2025; Zhang et al., 2025). (Heng et al., 2025) enhanced dynamic perception for manipulation tasks by forecasting future tactile signals. (Xie et al., 2025) proposed a masking strategy tailored to tactile videos, enhancing the capture of simple physical properties. (Li et al., 2025a) further incorporated force prediction as an auxiliary task to better model interaction dynamics. In parallel, advances in tactile simulators have enabled simple dynamic interactions and manipulation with tactile feedback in simulation (Akinola et al., 2025; Sun et al., 2025). For instance, Luu et al. (2025) built a manipulation benchmark based on the TacSL (Akinola et al., 2025) simulator, providing a scalable platform to evaluate dynamic tactile perception in interactive manipulation scenarios. In this work, we go beyond these directions by introducing multi-level dynamic enhanced modules to more comprehensively capture interaction dynamics and their underlying physical principles.

## 3 TACTILE HIERARCHICAL DYNAMIC DATASET

As a primary medium of human interaction with the physical world, touch exhibits rich and intricate dynamic characteristics. Capturing these dynamics requires not only advanced sensors but

also large-scale, high-quality datasets that reflect the temporal and physical nature of tactile interactions. However, most existing tactile datasets remain limited to simple paradigms such as pressing or random sliding, providing insufficient support for complex dynamic perception. To address this gap, we systematically establish a hierarchy of dynamic perception capabilities and propose a *tactile dynamic pyramid* that stratifies tactile data into five tiers based on the complexity of the dynamic perception capabilities they support, as shown in Fig. 1. This pyramid provides a principled framework to guide the collection of more informative dynamic tactile data. Specifically: (T5) **Press Only** data mainly support recognition of object-level attributes with minimal temporal variation; (T4) **Random Action** data introduce limited temporal changes, enabling perception of surface-related dynamics but lacking task relevance; (T3) **Specific Action** data capture structured dynamics associated with atomic interactions, facilitating action-level tactile understanding; (T2) **Manipulation** data reflect task-driven, temporally evolving contact changes, essential for learning real-world manipulation skills; and (T1) **Force** data explicitly ground tactile dynamics in physical force principles, enabling reasoning about force–deformation relationships and supporting fine-grained, force-sensitive manipulation tasks. As the tier level increases, data collection becomes more challenging or requires stricter constraints, and the data rarity increases. However, higher-tier data provides richer annotations or more realistic manipulation scenarios, enabling the development of stronger dynamic tactile perception capabilities. Most existing tactile datasets reside in Tier 4 and 5, offering insufficient support for advanced dynamic perception tasks such as dexterous manipulation, while higher-tier data remain scarce. Shahidzadeh et al. (2025) introduced a press-based touch–force dataset, but it excludes complex interactions like sliding, restricting its support for complex dynamic perception.

To address this gap, we present **ToucHD**, a large-scale tactile dataset with 2,426,174 contact samples designed as a **T**actile **H**ierarchical **D**ynamic resource to enrich higher-tier dynamic tactile data. Specifically, the dataset comprises three subsets corresponding to the highest 3 tiers of the pyramid:

**Simulated Atomic Action Data (Sim)**. Using an IMPM-based simulator (Shen et al., 2024), we collect 1,118,896 multi-sensor contact frames from five optical tactile sensors performing four atomic actions—sliding left/right and rotating clockwise/counterclockwise—on 1,043 objects sourced from ObjectFolder (Gao et al., 2022) and OmniObject3D (Wu et al., 2023). We further augment the data by rotating the two sliding actions, thereby generating additional upward and downward sliding samples. This data corresponds to Tier 3 (Specific Action) of the tactile dynamic pyramid, supporting explicit learning of tactile variations induced by structured dynamic interactions.

**Real-World Manipulation Data (Mani)**. We modify FastUMI (Wu et al., 2024) by equipping its two grippers with different tactile sensors, enabling efficient collection of multi-sensor tactile manipulation data. Using two distinct sets of sensors, we collect 584,842 contact frames from 46 carefully designed manipulation tasks, while simultaneously recording the interaction videos. This portion of the data corresponds to Tier 2 (Manipulation Data) and explicitly supports tactile pretraining models in capturing fine-grained dynamic tactile variations during real manipulation tasks.

**Touch-Force Paired Data (Force)**. We collect 722,436 touch–force pairs using five carefully selected tactile sensors. All sensors are mounted on a fixed base, while 71 distinct indenters are sequentially attached to the end-effector of a robotic arm. Under programmatic control, each indenter performs sliding motions in four directions—forward, backward, left, and right—across the sensor surface, while a wrist-mounted force sensor records 3D contact force sequences. These touch–force pairs correspond to Tier 1 (Force Data), providing explicit supervision for models to perceive fine-grained contact forces and serving as evaluation benchmarks for physical understanding.

As illustrated in Fig. 1, ToucHD integrates action-specific, real-world manipulation, and force-paired data, offering broad coverage across objects, sensors, and interaction dynamics. Together with existing lower-tier datasets, it forms a complete dynamic tactile data ecosystem, systematically supporting hierarchical dynamic perception capabilities. More details are shown in Appendix A.2.

## 4 METHOD

Building on the dynamic tactile data ecosystem established by ToucHD, we introduce **AnyTouch 2**, a general tactile representation learning framework that unifies sensor-invariant object-level understanding with multi-level dynamic perception capabilities, as shown in Fig. 2. Specifically, we start from pixel-level dynamic detail learning as the foundation (Sec. 4.1), extend to semantic-level tac-

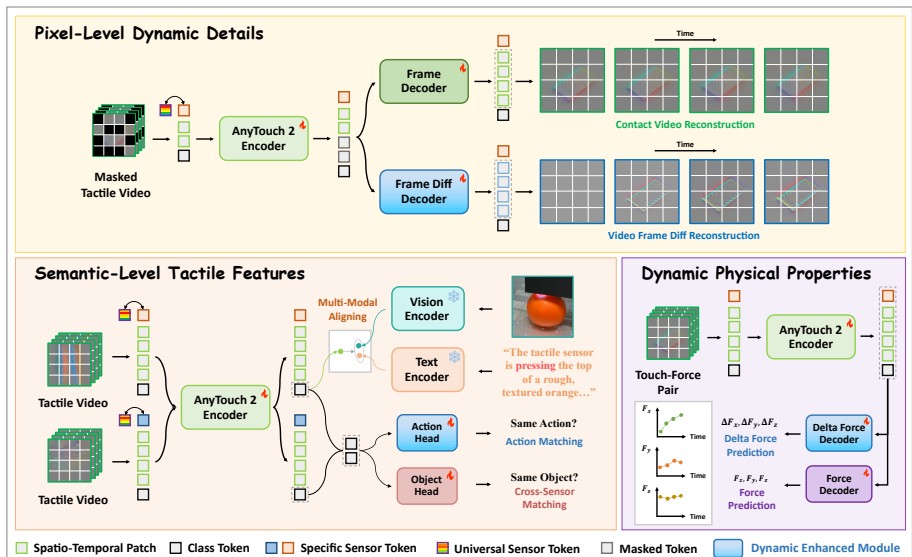

Figure 2: **Overview of AnyTouch 2.** Our model unifies object-level tactile semantics with fine-grained dynamic and physical perception, learning a general tactile representation that supports a broad spectrum of downstream tasks. By incorporating multi-level dynamic enhanced modules aligned with the tiers of the tactile dynamic pyramid, it strengthens sensitivity to subtle tactile variations and improves reasoning about the physical properties underlying dynamic interactions.

tile feature understanding (Sec. 4.2), and further advance to modeling dynamic physical properties (Sec. 4.3), aligning with the hierarchical tiers in our tactile dynamic pyramid.

## 4.1 PIXEL-LEVEL DYNAMIC DETAILS

Understanding pixel-level tactile deformations forms the basis of higher-level dynamic perception. To enhance the capacity for capturing fine-grained temporal changes, we employ a video masked autoencoder (Tong et al., 2022) to learn diverse deformation patterns from consecutive frames across multiple optical sensors. To focus on deformations rather than sensor-specific backgrounds, we subtract the background frame from each frame, yielding a normalized input $\mathbf{T} = (T_1, T_2, ..., T_N) \in \mathbb{R}^{N \times H \times W \times 3}$, where $N$ is the number of frames and $H \times W$ denotes the shape of tactile images. We partition $\mathbf{T}$ into non-overlapping 3D spatio-temporal tokens of size $s \times h \times w$ where s is the tube size and $h \times w$ denotes the patch size, yielding a token sequence of length $M = \frac{N}{s} \times \frac{H}{h} \times \frac{W}{w}$. We apply tube masking with a mask ratio $\rho$, and reconstruct the masked video into $\hat{\mathbf{T}}$ via a frame decoder. The training loss $\mathcal{L}_{\text{rec}}^{\text{ori}}$ is defined as the mean squared error (MSE) over masked tokens:

$$\mathcal{L}_{\text{rec}}^{\text{ori}} = \frac{1}{N|\Omega_M|} \sum_{n=1}^{N} \sum_{p \in \Omega_M} |\hat{T}_n(p) - T_n(p)|^2, \tag{1}$$

where $p$ is the token index and $\Omega_M$ is the set of masked tokens. Unlike natural videos, tactile deformations are highly localized and subtle, requiring explicit mechanisms to highlight small frame-to-frame changes. To this end, we further introduce frame-difference reconstruction to strengthen the model's sensitivity to fine-grained temporal variations. Specifically, we subtract the first frame $T_1$ of the video $\mathbf{T}$ from each subsequent frame to obtain the frame differences $\mathbf{D} = (D_2, ..., D_N) \in \mathbb{R}^{(N-1) \times H \times W \times 3}$, where $D_n = T_n - T_1, n = 2, ..., N$. A frame-difference decoder is simultaneously trained to reconstruct $\mathbf{D}$ from masked tokens with an MSE loss:

$$\mathcal{L}_{\text{rec}}^{\text{dif}} = \frac{1}{N|\Omega_M|} \sum_{n=2}^{N} \sum_{p \in \Omega_M} |\hat{D}_n(p) - D_n(p)|^2. \tag{2}$$

The total pixel-level loss is defined as $\mathcal{L}_{\text{Pixel}} = \mathcal{L}_{\text{rec}}^{\text{ori}} + \mathcal{L}_{\text{rec}}^{\text{dif}}$. By jointly reconstructing both the original frames and their frame differences, the model learns to capture both global deformation patterns and subtle fine-grained temporal variations essential for dynamic perception. This dual reconstruction strategy establishes a strong foundation for higher-level semantic and physical property perception.

## 4.2 SEMANTIC-LEVEL TACTILE FEATURES

While pixel-level deformation modeling lays the foundation for dynamic tactile perception, a general tactile representation also requires capturing semantic-level features that generalize across objects, sensors, and actions. To achieve this, we first leverage multi-modal alignment to embed tactile data into a shared semantic space grounded in perceptual and linguistic concepts such as object identity, material properties, and interaction descriptions. Following the CLIP paradigm (Radford et al., 2021; Feng et al., 2025b), tactile features are aligned with their paired visual and textual features as:

$$\mathcal{L}_{\text{Align}} = \frac{\alpha_{TV}}{2}(\mathcal{L}_{T \to V} + \mathcal{L}_{V \to T}) + \frac{\alpha_{TL}}{2}(\mathcal{L}_{T \to L} + \mathcal{L}_{L \to T}), \tag{3}$$

where $\mathcal{L}_{T \to V}, \mathcal{L}_{V \to T}$ and $\mathcal{L}_{T \to L}, \mathcal{L}_{L \to T}$ are tactile–visual and tactile–language contrastive losses respectively, while $\alpha_{TV}, \alpha_{TL}$ control their aligning strength. The full formulations are provided in Appendix A.7. In parallel, we employ cross-sensor matching (Feng et al., 2025b) to align tactile signals from different sensors that contact the same object, promoting sensor-invariant object-level feature learning. For each tactile video $\mathbf{T}$ from TacQuad, ToucHD (Sim), or ToucHD (Force), we sample a positive $\mathbf{T}_{\text{obj}}^{+}$ within these datasets that contacts the same object but originates from a different sensor. Additionally, a negative $\mathbf{T}_{\text{obj}}^{-}$ from a different object is randomly drawn from the batch. For each triplet $(\mathbf{T}, \mathbf{T}_{\text{obj}}^{+}, \mathbf{T}_{\text{obj}}^{-})$, the model predicts similarity scores between $\mathbf{T}$ and the other samples, and is trained with a binary cross-entropy loss to distinguish these pairs as:

$$\mathcal{L}_{\text{obj}} = -\log \sigma(sim(\mathbf{T}, \mathbf{T}_{\text{obj}}^{+})) - \log\left(1 - \sigma(sim(\mathbf{T}, \mathbf{T}_{\text{obj}}^{-}))\right), \tag{4}$$

where $\sigma(\cdot)$ denotes the Sigmoid function and $sim(\cdot, \cdot)$ represents the similarity score computed from the CLS tokens of the two samples through a linear head.

While existing components mainly focus on static attribute learning, we introduce action matching to capture the semantics of structured dynamic tactile interactions. In particular, this objective guides the model to embed atomic action information into the representation space. The tactile videos from ToucHD (Sim) and ToucHD (Force) are grouped into 8 atomic actions, including pressing, leaving, sliding (4 directions), and rotating (2 directions). The model is trained to cluster representations of the same action while separating different ones. This encourages the encoder to recognize the characteristic temporal patterns, motion directions, and frame-to-frame deformations associated with each action, effectively embedding semantic-level action information into the tactile representation. Concretely, for a tactile video $\mathbf{T}$, we sample a positive $\mathbf{T}_{act}^{+}$ from the same action class (potentially across different objects or sensors) within these datasets, and a negative $\mathbf{T}_{act}^{-}$ from a different action class within the batch. Similar to the cross-sensor matching, we train the model to pull together frame sequences of the same action while pushing apart sequences of different actions:

$$\mathcal{L}_{\text{act}} = -\log \sigma(sim(\mathbf{T}, \mathbf{T}_{\text{act}}^{+})) - \log\left(1 - \sigma(sim(\mathbf{T}, \mathbf{T}_{\text{act}}^{-}))\right). \tag{5}$$

This objective explicitly incorporates semantic-level action information into the tactile representation, improving the model's understanding of dynamic interactions and supporting downstream manipulation tasks that depend on action-aware perception. The total matching loss is then $\mathcal{L}_{\text{Match}} = \mathcal{L}_{\text{obj}} + \mathcal{L}_{\text{act}}$. By jointly optimizing these objectives, the model captures both static object-level and dynamic action-aware semantic features, effectively bridging low-level tactile signals with high-level perceptual understanding. However, the model still falls short of fully understanding the underlying physical properties that drive these interactions.

## 4.3 PHYSICAL-LEVEL DYNAMIC PROPERTIES

Understanding the physical properties underlying tactile interactions requires integrating knowledge of both object-level attributes and action dynamics. Among these properties, contact force is fundamental, as it directly governs how objects deform, slip, or respond during manipulation (Huang et al., 2025). Accurately modeling force dynamics not only provides explicit supervision for the temporal evolution of tactile signals but also grounds the learned representations in the underlying physics of interactions. Therefore, we introduce the force prediction task to explicitly model the physical properties underlying tactile interactions. Using the large-scale touch–force pairs $(T_n, F_n)$ from ToucHD (Force), the model is trained to predict the 3D contact force $\mathbf{F} \in \mathbb{R}^{(N-1) \times 3}$ for each frame in a tactile video $\mathbf{T}$, excluding the first frame. This enables the model to directly associate dynamic

tactile deformations with their physical magnitudes. To further enhance sensitivity to fine-grained dynamic deformations, we introduce delta-force prediction, which focuses on capturing the temporal variations of contact forces. The model is trained to predict the force increments $\mathbf{\Delta F} \in \mathbb{R}^{(N-1) \times 3}$ where $\mathbf{\Delta F}_n = F_n - F_{n-1}, n = 2, ..., N$. This shifts the focus from static force values to dynamic transitions, encouraging the encoder to attend to subtle temporal cues and continuous deformation patterns. The force and delta-force decoders are jointly trained with an L1 Loss:

$$\mathcal{L}_{\text{Force}} = \frac{1}{3(N-1)}||\hat{\mathbf{F}} - \mathbf{F}||_1 + \frac{1}{3(N-1)}||\hat{\mathbf{\Delta F}} - \mathbf{\Delta F}||_1. \tag{6}$$

By explicitly predicting the 3D contact forces and their temporal variations from tactile videos, the model can bridge high-level semantic understanding with fine-grained dynamic properties. This enables a comprehensive and physically grounded representation across all tiers of the tactile dynamic pyramid, supporting dexterous manipulation and robust generalization across tasks and objects.

## 4.4 TRAINING RECIPE

Our model integrates tactile perception tasks spanning the hierarchical tiers of the tactile dynamic pyramid, from low-level pixel deformations to high-level semantic and force-sensitive interactions. To jointly optimize these multi-level objectives while mitigating task interference, we adopt a curriculum task scheduling strategy with task-specific start iterations and gradually increasing weights. Concretely, pixel-level reconstruction, as the foundation of tactile perception, is trained from the beginning with the highest weight. Higher-level tasks, including semantic tactile feature learning and dynamic physical property modeling, are introduced after several iterations $i$ with gradually increasing weights $\lambda_{\text{task}}^i$. This strategy ensures the model first captures robust low-level tactile patterns before learning more complex capabilities. The total loss $\mathcal{L}$ of our framework is defined as:

$$\mathcal{L}_{\text{total}} = \mathcal{L}_{\text{Pixel}} + \lambda_{\text{Align}}^i \mathcal{L}_{\text{Align}} + \lambda_{\text{Match}}^i \mathcal{L}_{\text{Match}} + \lambda_{\text{Force}}^i \mathcal{L}_{\text{Force}},$$
$$\lambda_{\text{task}}^i = \frac{\max(0, \ i - i_{\text{task}})}{i_{\text{total}} - i_{\text{task}}} \lambda_{\text{task}}^{\max}, \quad \text{task} \in \{\text{Align}, \text{Match}, \text{Force}\}, \tag{7}$$

where $i_{\text{task}}$ is the task start iteration and $\lambda_{\text{task}}^{\max}$ denotes the maximum task-specific weight.

## 5 EXPERIMENTS

In this section, we comprehensively evaluate our model's general tactile perception. We first test it on benchmarks covering object-level properties and dynamic physical attributes (Sec. 5.2), then on four real-world manipulation tasks spanning multiple tiers of the tactile dynamic pyramid, assessing its ability to generalize across hierarchical dynamic capabilities (Sec. 5.3).

### 5.1 DATASETS AND BASELINES

During pre-training, we filtered contact samples from 9 different tactile datasets, including: Touch and Go (TAG) (Yang et al., 2022), VisGel (Li et al., 2019), ObjectFolder Real (Gao et al., 2023) , TVL (Fu et al., 2024), YCB-Slide (Suresh et al., 2023), SSVTP (Kerr et al., 2022), Octopi (Yu et al., 2024), TacQuad (Feng et al., 2025b), and ToucHD. For downstream evaluation, we adopt TAG and Cloth (Yuan et al., 2018) for object property understanding, and Sparsh (Higuera et al., 2025a) together with ToucHD Bench (10 unseen indenters) for dynamic physical understanding, covering 3 mainstream optical tactile sensors: GelSight (Yuan et al., 2017), DIGIT (Lambeta et al., 2020), and GelSight Mini (Inc.). We compare the AnyTouch 2 model with representative tactile representation learning methods: UniTouch (Yang et al., 2024) and T3 (Zhao et al., 2025b) (single-frame input), and MAE (Sparsh), VJEPA (Sparsh) (Higuera et al., 2025a), and AnyTouch 1 (Feng et al., 2025b) (multi-frame input). Single-frame models are fed two consecutive frames along the batch dimension to handle temporal data without architecture changes. To fairly compare and simultaneously evaluate the benefits of our ToucHD dataset, we also train an MAE (Sparsh)† model on the same training data, including ToucHD as AnyTouch 2. The detailed introduction is provided in Appendix A.3 and A.4.

### 5.2 OFFLINE BENCHMARK EVALUATION

To evaluate both object-level and dynamic physical perception, we conduct extensive experiments on Object Bench (TAG Material and Cloth Textile Classification), Sparsh Bench (Force Prediction,

Table 1: Evaluation of object-level attribute understanding on ObjectBench and physical-level dynamic perception on SparshBench and our ToucHD Bench. The evaluation covers three mainstream optical tactile sensors: GelSight (GS), DIGIT (DG), and GelSight Mini (Mini). Green rows indicate static models that take a single frame as input, while blue rows denote dynamic models that process multiple consecutive frames. (S) marks the pre-trained Sparsh model, and † indicates the use of additional training data including ToucHD. Underlined numbers denote the second-best results.

| Method | Object Bench | | Sparsh Bench | | | | | ToucHD Bench | |
|---|---|---|---|---|---|---|---|---|---|
| | TAG | Cloth | Pose | Slip (Delta Force) | | Force | | Force | |
| | Acc(↑) | Acc(↑) | Acc(↑) | F1 Score(↑) / RMSE(↓) | | RMSE(↓) | | RMSE(↓) | |
| | GS | GS | DG | DG | Mini | DG | Mini | DG | Mini |
| CLIP | 51.65 | 26.76 | 54.54 | 33.13 / 174.39 | 85.47 / 177.67 | 1278.08 | 553.19 | 4880.94 | 4492.77 |
| UniTouch | 61.27 | 20.43 | 54.92 | 35.43 / 169.26 | 87.73 / 211.81 | 1540.76 | 652.61 | 4146.55 | 4400.57 |
| T3 | 52.51 | (Seen) | 55.01 | 52.12 / 152.55 | 77.65 / 210.39 | 1535.84 | 640.39 | 4805.63 | 4877.66 |
| VJEPA (S) | 54.67 | 18.66 | 55.09 | 83.33 / 105.63 | 97.00 / 121.31 | 957.73 | 428.56 | 4766.11 | 3208.10 |
| MAE (S) | 59.47 | 19.40 | 55.92 | 83.30 / 98.33 | 97.50 / 102.64 | 821.26 | 297.96 | 1953.82 | 3655.39 |
| MAE (S)† | 63.32 | 36.84 | 57.09 | 85.67 / 92.47 | 97.40 / 98.85 | 741.67 | 239.98 | 1714.86 | 2467.42 |
| AnyTouch 1 | **80.82** | (Seen) | 56.22 | 40.60 / 169.42 | 88.92 / 162.41 | 1235.11 | 488.31 | 3968.81 | 4050.45 |
| **AnyTouch 2** | 76.97 | **42.31** | **57.83** | **86.66 / 87.80** | **97.96 / 80.83** | **624.26** | **202.14** | **894.32** | **1051.03** |

Pose Estimation and Slip Detection) and our ToucHD Bench (Force Prediction). For the Sparsh Force Prediction task, we evaluate the models on the unseen flat indenter. To more comprehensively evaluate the model's understanding of force, we further conduct comparisons on the ToucHD Bench, which consists of 10 unseen indenters, and select 3 of them as testing indenters. To further probe fine-grained dynamic understanding, we add an additional evaluation within the Slip Detection task, where the model predicts 3D force changes across the input contact frame sequences. All reported root mean squared error (RMSE) values are measured in mN.

As shown in Tab. 1, our AnyTouch 2 model achieves performance comparable to AnyTouch 1 on Object Bench, which primarily emphasizes static semantic features. At the same time, AnyTouch 2 consistently outperforms prior approaches across all other evaluation tasks requiring fine-grained dynamics and force-sensitive reasoning. This demonstrates its ability to unify object-level understanding with action-aware and force-grounded dynamic perception. Models leveraging multiple consecutive frames show clear advantages on the two dynamic benchmarks. In contrast, single-frame baselines sometimes perform even worse than CLIP model on Force Prediction and Slip Detection, largely because they lack temporal position embeddings and thus cannot capture the ordering of tactile inputs. This highlights the indispensable role of dynamic tactile perception and reveals the limitations of training solely on lower-tier datasets, which lack the temporal richness needed for capturing fine-grained dynamics. Interestingly, while MAE (Sparsh) and VJEPA (Sparsh) achieve competitive results on dynamic tasks, they still fall behind CLIP and UniTouch, which benefit from semantic-level multi-modal alignment, on Cloth classification. This further underscores the value of AnyTouch 2: enhancing dynamic perception while preserving robust static understanding, achieving a general tactile representation. Finally, augmenting MAE (Sparsh) with more training data, including our ToucHD dataset, yields consistent improvements across all tasks—even without additional objectives—highlighting the unique value of ToucHD as a high-tier dynamic tactile dataset.

## 5.3 ONLINE REAL-WORLD MANIPULATION

To evaluate our model in realistic scenarios, we design four challenging real-world manipulation tasks that explicitly span the tactile dynamic pyramid: Tactile Grasping (Tier 5), Whiteboard Wiping (Tier 4 & 3), USB Insertion (Tier 2) and Chip Moving (Tier 1), as shown in Fig. 3. These tasks comprehensively cover all tiers of the dynamic pyramid, from force-sensitive precision manipulation to object-level property recognition, providing a holistic benchmark for validating the model's dynamic tactile perception capabilities in real-world environments. We adopt Diffusion Policy (Chi et al., 2023) as the policy head and freeze all tactile encoders during training. Each task is tested 20 times, and we report the average success rate. Detailed task setups are provided in Appendix A.6.

As shown in Fig. 4, static single-frame models perform significantly worse than dynamic models in real-world manipulation, particularly on higher-tier tasks, highlighting the necessity of dynamic

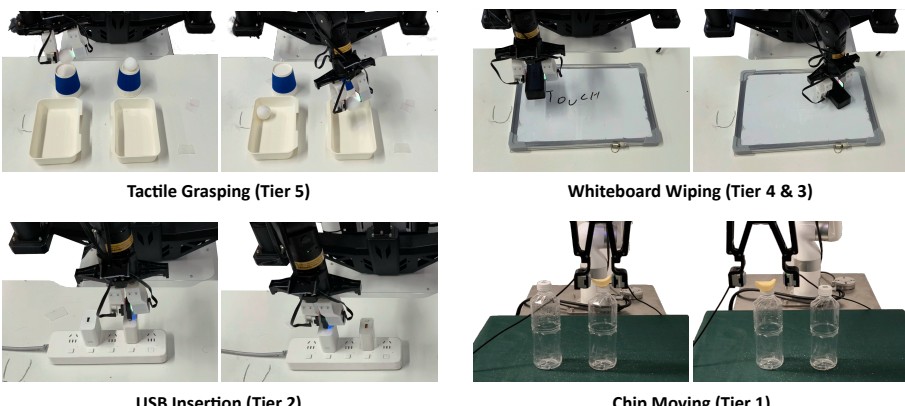

Figure 3: **Real-world manipulation tasks.** We evaluate models on real-world manipulation tasks that span the dynamic capabilities of different tiers in our tactile dynamic pyramid: Tactile Grasping (Tier 5), Whiteboard Wiping (Tiers 4 & 3), USB Insertion (Tier 2), and Chip Moving (Tier 1).

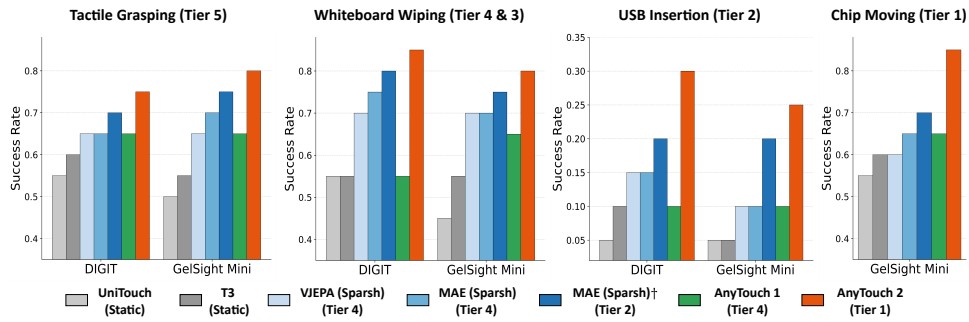

Figure 4: **Evaluation of real-world manipulation tasks.** This evaluation spans DIGIT and Gel-Sight Mini. Each dynamic model that takes consecutive tactile frames as input has a corresponding dynamic tier, which denotes the highest level of the training data and objectives used in our tactile dynamic pyramid shown in Fig. 1, reflecting the model's dynamic perception capability. † denotes additional training data including ToucHD.

perception for contact-rich manipulation. Moreover, depending on the tier of the training data and objectives, different dynamic perception models exhibit varying performance across different tiers of tasks. The three Tier 4 dynamic perception models achieve comparable performance on the Tier 5 Tactile Grasping task, while AnyTouch 1, which focuses more on static object attributes, lags behind MAE (S) and VJEPA (S), which better capture inter-frame variations on the Tier 4 & 3 task. However, all three models perform poorly on the higher-level Tier 1 and Tier 2 tasks that are not covered by their training data, revealing the limits of using only lower-tier dynamic data. By further incorporating ToucHD into the training data of MAE (S), the model gains dynamic perception capabilities across all other Tier 2 and lower-tier tasks, except accurate force perception for Tier 1, achieving significant improvements over the original MAE (S) in all tasks. Ultimately, by integrating the ToucHD dataset with multi-level dynamic enhanced modules, AnyTouch 2 achieves the strongest Tier-1 dynamic perception capability, outperforming all baselines across all 4 real-world tasks, including the most delicate and challenging Tier 1 Chip Moving task. This demonstrates that the hierarchical dynamic data provided by ToucHD effectively supports higher-tier dynamic capabilities, and that our AnyTouch 2 framework effectively bridges all tiers of the tactile dynamic pyramid, establishing a solid foundation for general tactile perception in real-world manipulation.

Beyond model comparisons, we also observe notable differences between the two optical tactile sensors. GelSight Mini, with its cleaner background and sharper deformation imaging, excels at capturing fine-grained details, outperforming DIGIT on the Tier-5 task using AnyTouch 2. In contrast, DIGIT's higher acquisition frequency (30 Hz vs. GelSight Mini's 18 Hz) provides more training samples and denser dynamic information, leading to superior performance on higher-tier manipulation tasks. These findings underscore not only the complementary strengths of different sensors but also the importance of models that can effectively integrate data from diverse tactile sensors.

Table 2: The impact of the modules in AnyTouch 2 on offline benchmarks. This evaluation spans three mainstream optical tactile sensors: GelSight (GS), DIGIT (DG), and GelSight Mini (Mini). The red arrow ↓ indicates a significant drop in performance.

| Method | Object Bench | | Sparsh Bench | | | | ToucHD Bench | |
|---|---|---|---|---|---|---|---|---|
| | TAG | Cloth | Slip (Delta Force) | | Force | | Force | |
| | Acc(↑) | Acc(↑) | F1 Score(↑) / RMSE(↓) | | RMSE(↓) | | RMSE(↓) | |
| | GS | GS | DG | Mini | DG | Mini | DG | Mini |
| **AnyTouch 2** | **76.97** | **42.31** | 86.66 / 87.80 | 97.96 / **80.83** | 624.26 | 202.14 | **894.32** | 1051.03 |
| - Diff Recon | 76.19 | 41.33 | 84.39↓ / 94.88↓ | 97.81 / 100.84↓ | 687.13↓ | 225.18↓ | 1009.44↓ | 1123.47 |
| - Action Match | 76.93 | 42.05 | 84.42↓ / 87.98 | 97.68↓ / 83.84 | 643.75 | 203.61 | 896.21 | 1082.39 |
| - Force Pred | 76.46 | 41.45 | 86.35 / 90.72 | 97.88 / 96.34↓ | 770.44↓ | 254.10↓ | 1646.95↓ | 2008.38↓ |
| - MM Aligning | 63.84↓ | 37.61↓ | **87.31 / 81.44** | **98.16** / 85.89 | **589.13** | **193.73** | 976.73↓ | **972.37** |
| - ToucHD (Sim) | 76.54 | 41.97 | 84.68↓ / 88.78 | 97.83 / 108.25↓ | 624.39 | 207.83 | 992.96↓ | 1113.56 |
| - ToucHD (Mani) | 76.43 | 41.01 | 86.13 / 88.12 | 97.93 / 80.96 | 655.56 | 208.46 | 1118.49↓ | 1193.84 |
| - ToucHD (Force) | 74.33↓ | 40.87↓ | 84.91↓ / 107.43↓ | 97.85 / 109.37↓ | 777.41↓ | 266.43↓ | 1792.49↓ | 2424.68↓ |
| - ToucHD | 68.92↓ | 40.39↓ | 84.16↓ / 110.68↓ | 97.67↓ / 136.36↓ | 783.64↓ | 257.95↓ | 2448.89↓ | 2982.46↓ |

## 5.4 ABLATION STUDY

To comprehensively evaluate the contributions of each module in our model to its general tactile perception capabilities, we conduct extensive ablation studies on three benchmarks. The experimental results are shown in Tab. 2. When the action matching module is removed, the model's performance on the slip detection task decreases. Similarly, removing the force prediction module leads to reduced performance on the force prediction and delta force prediction tasks. Furthermore, when the frame-difference reconstruction task, which serves as a fundamental fine-grained dynamic perception objective, is removed, the model exhibits decreased performance across all dynamic tasks. These results demonstrate the effectiveness of our designed multi-tier dynamic enhancement modules in improving dynamic perception capabilities. However, when the multi-modal alignment module is removed, we observe an interesting phenomenon: the model shows some performance improvement across most dynamic perception tasks, while exhibiting a noticeable decline on Object Bench, which focuses more on object-level static semantic features. This is because multi-modal alignment inherently emphasizes static tactile features, bringing together different possible actions on the same object, which can somewhat compromise the model's fine-grained dynamic perception capabilities. This essentially reflects a trade-off between perceiving static tactile object properties and dynamic tactile features, as both are crucial for general tactile perception. We further investigate the contribution of the ToucHD dataset and its subsets to the dynamic perception capabilities. When we remove the ToucHD (Sim) subset which contains a large number of atomic tactile actions, the model's performance on the two slip tasks decreases. This indicates that this Tier 3 dataset does primarily supports the perception of structured dynamic tactile deformations. When the ToucHD (Mani) subset is removed, the model also shows a consistent performance drop. However, since this subset primarily supports dynamic perception in real-world manipulation tasks corresponding to Tier 2, the magnitude of the decrease is relatively small. In contrast, when the ToucHD (Force) subset is removed, the model loses data support for perceiving Tier 1 dynamic physical properties, resulting in a performance drop across all benchmarks. Finally, when the entire ToucHD dataset is removed, the model exhibits a significant performance drop across all tasks, highlighting the crucial role of the ToucHD dataset in supporting general dynamic tactile perception capabilities. More ablation and hyper-parameter experiments are shown in Appendix A.9 and A.10.

## 6 CONCLUSION

In this work, we advance dynamic tactile perception by introducing the tactile dynamic pyramid as a systematic paradigm to guide both data collection and model design for hierarchical tactile perception capabilities. From the data perspective, the proposed ToucHD dataset serves as the final missing piece, completing a comprehensive dynamic tactile data ecosystem that supports multiple tiers of perception. From the model perspective, our AnyTouch 2 general representation learning framework integrates multi-level objectives across all tiers, endowing it with comprehensive dynamic tactile perception capabilities. We believe this work establishes a solid foundation for general tactile perception and will push tactile intelligence into the new era of dynamic perception.

ACKNOWLEDGMENTS

This work was supported by Beijing Natural Science Foundation (4262050) and by fund for building world-class universities (disciplines) of Renmin University of China. It was also supported by the Open Foundation of the State Key Laboratory of Precision Space-time Information Sensing Technology No.STSL2025-B-07-01 (C). We would like to extend our special thanks to Daimon Robotics and Prof. Huanbo Sun for their support with tactile sensing devices. We would also like to thank Denghang Huang, Mingxin Wang, Shaoxuan Xie, Boyue Zhang, and Yuhao Sun for their help with 3D component design and data collection.

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

# A APPENDIX

In the appendix, we first provide a detailed description of the structure of Tactile Dynamic Pyramid (A.1) and the ToucHD data collection process (A.2), followed by comprehensive statistics and characteristics of the training dataset (A.3). We then present the details of benchmarks and baselines (A.4), implementation details (A.5), and the setup of real-world tasks (A.6). In addition, we include the formulation of the complete multi-modal alignment loss (A.7) and detailed figures for force prediction evaluation (A.8). We also report an extensive ablation study (A.9) and a hyper-parameter study (A.10), conduct cross-sensor generation experiments (A.11), and discuss limitations and future work (A.12). Finally, we provide a statement regarding the usage of LLMs (A.13).

## A.1 STRUCTURE OF TACTILE DYNAMIC PYRAMID

In this section, we further clarify the criteria of the tiered structure of our Tactile Dynamics Pyramid in Fig. 1. These tiers are defined based on the data collection efforts, the types of actions, and the difficulty of obtaining labels:

- **Tier 5 (Press-Only)**: This tier of data is collected by **only pressing the sensor against objects** using either handheld operation or a robot arm. No detailed action-type annotations or paired force labels are provided.

- **Tier 4 (Random Action)**: This tier of data is collected by **pressing the sensor against objects, followed by random sliding and rotation** using either handheld operation or a robot arm. No detailed action-type annotations or paired force labels are provided.

- **Tier 3 (Specific Action)**: This tier of data is collected by **programmatically controlling the sensor to press and slide** along the object surface following specific predefined actions. Detailed action-type labels are available, but no paired force data is provided.

- **Tier 2 (Manipulation Data)**: This tier of data is collected during **real object manipulation tasks** using a robot arm or a UMI device. No paired force data is provided.

- **Tier 1 (Force Data)**: This tier of data is collected by **a robot arm equipped with a force sensor**, with either an indenter or an object interacting with the tactile sensor. This is the only tier that contains paired force labels.

As the tier level increases, the corresponding data collection process becomes more challenging or requires stricter constraints, and the data rarity increases. However, higher-tier data provides richer annotations or more realistic manipulation scenarios, enabling the development of stronger dynamic tactile perception capabilities.

## A.2 DETAILS OF TOUCHD COLLECTION

### A.2.1 SIMULATED DATA

With the advancement of tactile simulators, simple dynamic contact can now be rendered with high fidelity (Shen et al., 2024; Sun et al., 2025). Moreover, simulators allow easy replacement of sensors and objects, enabling the collection of large-scale multi-sensor paired dynamic contact data at low cost. Therefore, we employ an IMPM (Improved Material Point Method) optical tactile simulation platform (Shen et al., 2024), which consists of two main components: elastomer–object contact simulation and rendering. The input objects are point clouds sourced from ObjectFolder 2 (Gao et al., 2022) and OmniObject3D (Wu et al., 2023). The total number of objects reaches over 1000, and These objects cover more than 10 different material types across five major environments—household, office, video, industrial, and natural, surpassing the material diversity of several existing large-scale tactile datasets such as YCB-Slide and ObjectFolder Real. Each object is first converted into a standardized NumPy format. We then initialize the grids and particles based on the object's initial position. Specifically, the 3D grid dimensions are manually specified, including the number of nodes, their velocities, masses, and the grid size. Particle initial parameters are also defined, which consist of particle number, position $x \in \mathbb{R}^3$, velocity $v \in \mathbb{R}^3$, mass $m \in \mathbb{R}^+$, affine velocity field $C \in \mathbb{R}^3$, deformation gradient $F \in \mathbb{R}^3$, density $\rho \in \mathbb{R}^+$, Young's modulus $E \in \mathbb{R}^+$ and Poisson's ratio $\nu \in \mathbb{R}$. From these, particle volumes and Lamé parameters are computed. To reduce the movement time, the object is placed so that its center aligns with the elastomer's center,

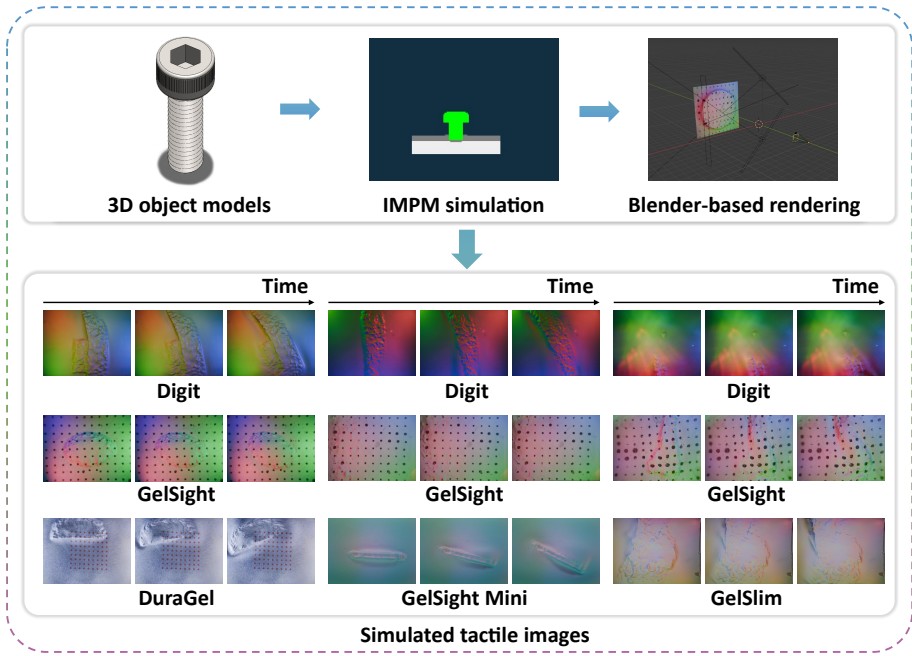

Figure 5: **Simulated data acquisition.** 3D object models are processed using an IMPM optical tactile simulation platform, which comprises two components: the IMPM simulator and a Blender-based rendering module. Firstly, the IMPM simulator generates 3D elastomer models that capture deformations caused by object rotations and sliding motions. The Blender-based rendering module then converts these elastomer models into tactile images for different optical sensors.

and its bottom surface is tangent to the top surface of the elastomer. The object is then driven downward using IMPM until the elastomer reaches the target deformation depth. During this process, the simulation continues to advance the object step by step until the specified deformation threshold is met. We define six object motions including clockwise rotation, counter-clockwise rotation, and translation to the left and right as the atomic actions. These motions are simulated step by step using IMPM until the target pose is reached. Each simulated interaction produces 30 frames capturing the elastomer deformation throughout the motion. After these, the reconstructed triangle meshes are imported into Blender. Different tactile sensor backgrounds are then projected onto the mesh surface, thereby producing simulated images corresponding to five optical tactile sensors, including GelSight (Inc.), DIGIT (Lambeta et al., 2020), GelSight Mini (Inc.), GelSlim (Donlon et al., 2018), and DuraGel (Zhang et al., 2024). As surface geometry deforms during contact, the marker patterns deform accordingly, eliminating the need for manual annotation. LED lighting effects are then incorporated according to the sensor design, including LED positions, colors, and power settings, and the corresponding rendered images are generated. By rotating the left and right translation samples, we can additionally obtain upward and downward translation samples. These eight atomic actions are sufficient to serve as the minimal fundamental action units for most tasks, while combinations of these actions may occur in some complex tasks.

There are also tactile datasets that use implicit neural representations to store object-level tactile information (Gao et al., 2022; Li et al., 2025b; Dou et al., 2024). By providing a contact location as input, these neural fields can generate large numbers of tactile frames. While these datasets can increase material diversity, they cannot directly render tactile images during dynamic contact, providing only large numbers of static images. Therefore, these data essentially belong to Tier 5 of the tactile dynamics pyramid, offering few advantages compared to tactile simulators that can render dynamic contact processes.

### A.2.2 MANIPULATION DATA

The advent of UMI (Chi et al., 2024) has enabled the large-scale collection of real-world manipulation data at relatively low cost. Building on the FastUMI design (Wu et al., 2024), we adapt the grip-

Table 3: Manipulation task descriptions.

| Index | Task Name | Description |
|---|---|---|
| 1 | Cap a Pen | The UMI grips the pen body while the left hand places the cap back on. |
| 2 | Uncap a Pen | The UMI grips the pen body while the left hand pulls the cap off. |
| 3 | Insert Hex Wrench | The UMI inserts a hex wrench into a socket fixed by the left hand. |
| 4 | Insert USB | The UMI grips a USB cable and inserts it into the port. |
| 5 | Remove USB | The UMI grips the USB cable and pulls it out of the port. |
| 6 | Cut Paper | The UMI grips a cutter to cut a slit while the left hand holds the paper. |
| 7 | Assemble Pen | The UMI and left hand align and rotate pen parts to assemble. |
| 8 | Disassemble Pen | The UMI and left hand rotate to separate pen parts. |
| 9 | Detach Velcro | The UMI grips the Velcro end and pulls while the left hand holds the other side. |
| 10 | Seal Zip Bag | The UMI moves along the sealing strip to close the bag. |
| 11 | Install Drill Bit | The UMI inserts a bit into a screwdriver fixed by the left hand. |
| 12 | Remove Drill Bit | The UMI pulls the bit out of the screwdriver. |
| 13 | Close Box Lid | The UMI grips the lid and closes the plastic box. |
| 14 | Tear Paper | The UMI tears a paper sheet apart while the left hand holds the other side. |
| 15 | Slide Mouse | The UMI grips and slides a mouse steadily on a mousepad. |
| 16 | Rotate Glue Stick | The UMI rotates the bottom while the left hand holds the top. |
| 17 | Apply Glue Stick | The UMI applies glue on paper with the glue stick. |
| 18 | Open Bit Case | The UMI grips and opens the lid of a bit case. |
| 19 | Close Bit Case | The UMI grips and closes the lid of a bit case. |
| 20 | Insert Key | The UMI removes a key from a lock. |
| 21 | Unlock with Key | The UMI rotates the key to unlock. |
| 22 | Place Test Tube | The UMI places a test tube into a rack. |
| 23 | Sweep Fruit | The UMI sweeps fruit into a dustpan held by the left hand. |
| 24 | Fold Towel | The UMI and left hand fold a towel twice. |
| 25 | Twist Towel | The UMI and left hand twist a towel. |
| 26 | Seal Document Bag | The UMI grips and slides the bag seal to close it. |
| 27 | Pull Tissue | The UMI pulls and unfolds a tissue with left-hand assistance. |
| 28 | Assemble Chopsticks | The UMI and left hand rotate chopstick halves to assemble. |
| 29 | Open Fan | The UMI assists in unfolding a fan held by the left hand. |
| 30 | Wipe Table | The UMI grips a rag and wipes stains back and forth. |
| 31 | Rotate Rubik's Cube | The UMI rotates the top and left faces while the left hand fixes the base. |
| 32 | Stack Blocks | The UMI stacks blocks on a base held by the left hand. |
| 33 | Unstack Blocks | The UMI removes blocks one by one from a stacked tower. |
| 34 | Assemble Medicine Bottle | The UMI grips and seals a bottle cap. |
| 35 | Scoop Rice | The UMI scoops rice and places it on the desk. |
| 36 | Remove Scissor Cover | The UMI pulls off a scissor cover while the left hand holds the handle. |
| 37 | Pick up Chip | The UMI transfers a chip without breaking it. |
| 38 | Straighten Cable | The UMI grips and straightens a bent cable with the left hand. |
| 39 | Flatten Clay | The UMI flattens a clay ball into a disc with assistance. |
| 40 | Stretch Clay | The UMI stretches a clay ball into a strip with assistance. |
| 41 | Press Clay into Mold | The UMI presses clay into a mold held by the left hand. |
| 42 | Shape Clay | The UMI shapes clay into a cylinder with assistance. |
| 43 | Zip Bag | The UMI grips and pulls a zipper to close the bag. |
| 44 | Write Whiteboard | The UMI holds a marker and writing a few words on the whiteboard. |
| 45 | Wipe Whiteboard | The UMI grips an eraser and wipes in a straight line. |
| 46 | Pour Water | The UMI grabs a bottle and pours half a cup of water into another cup. |

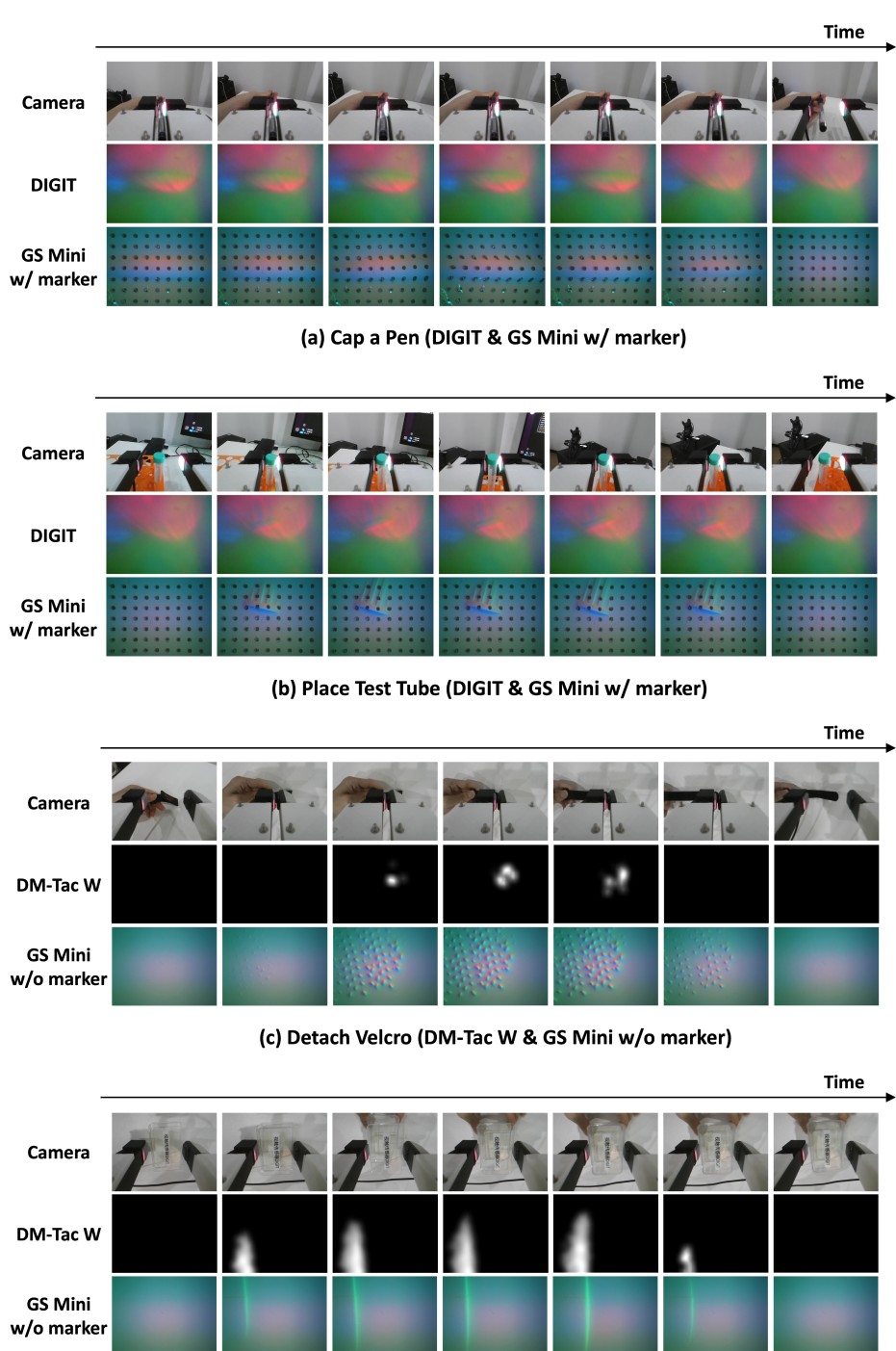

Figure 6: **Real-world manipulation data.** (a) and (b) were collected with a GelSight Mini (with markers) paired with DIGIT, corresponding to the tasks *Cap a Pen* and *Place Test Tube*, respectively. (c) and (d) were collected with a GelSight Mini (without markers) paired with DM-Tac W, corresponding to the tasks *Detach Velcro* and *Close Box Lid*, respectively. For each task, synchronized frames from the external camera and the two tactile sensors are shown to illustrate the dynamic tactile and visual changes during execution.

per structure to accommodate multiple tactile sensors for diverse data acquisition. Specifically, we employ three commercial optical tactile sensors: GelSight Mini (with and without markers) (Inc.), DIGIT (Lambeta et al., 2020), and DM-Tac W (Daimon , Shenzhen). These sensors exhibit complementary properties in terms of resolution, sensitivity, and dynamic response, allowing us to capture richer and more diverse tactile signals under the same manipulation scenarios. To facilitate the acquisition of paired tactile data, we divide the three sensors into two groups: GelSight Mini (with markers) with DIGIT and GelSight Mini (without markers) with DM-Tac W, and mount each group onto a pair of customized FastUMI grippers, enabling sensor-combination-based data collection.

In terms of task design, particular emphasis is placed on eliciting fine-grained dynamic tactile variations during the manipulation process. To this end, we design 46 manipulation tasks of varying difficulty that cover typical interaction patterns such as pushing, pulling, squeezing, rotating, sliding, and aligning. The detailed task specifications are summarized in the task description table provided in Tab.3. During data collection, both sensor groups perform the complete set of 46 tasks, ensuring direct comparability of tactile data across sensors under identical task conditions. For each task, we perform 4–10 repetitions, choosing different contact points whenever possible to manipulate the objects, thereby ensuring the diversity of the dataset. In total, we collect 584,842 real contact frames along with synchronized interaction videos. This portion of the dataset corresponds to Tier 2 Manipulation Data and is explicitly designed to support tactile pre-training models in perceiving fine-grained and dynamic tactile variations during real manipulation tasks. Representative synchronized visual and tactile data streams from the two different sensor groups across four example tasks are illustrated in Fig. 6.

It is worth noting that during data collection, we used the left hand in collaboration with the UMI device, rather than employing two UMI devices. This is because after we modified the UMI device by adding two tactile sensors, the overall setup became bulkier, and using dual UMIs to collect data would make many tasks difficult to perform. Therefore, we switched to a UMI+hand collaboration setup for large-scale data collection, which is essentially a trade-off. This may introduce some bias in the visual modality, but many existing studies (Yu et al., 2025; Wang et al., 2024; Zhou et al., 2025; Ye et al., 2024) have shown that even human-hand manipulation data can help improve the generalization ability of robotic manipulation.

Many existing works have collected tactile data using such specialized handheld devices (Liu et al., 2025; Zhu et al., 2025; Wu et al., 2025), but these were typically constrained to specific downstream tasks. In contrast, we are the first to collect tactile data across up to 46 diverse interaction tasks to support tactile representation learning.

### A.2.3 FORCE DATA

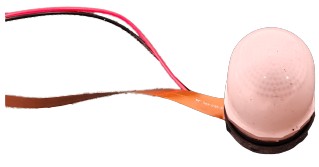

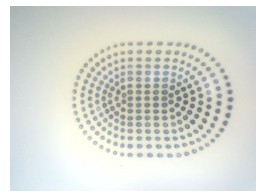 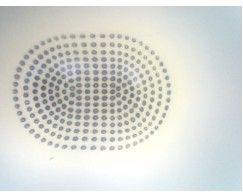

**Left Camera**     **Right Camera**

Figure 7: GelStereo BioTip Sensor.

Figure 8: Raw stereo camera data from the Gel-Stereo BioTip sensor.

Force represents one of the most essential physical properties in contact (Chen et al., 2025). Hence, equipping models with the ability to accurately perceive the force is key to achieving dexterous manipulation (Huang et al., 2025). Therefore, we collect paired touch–force data using five different optical tactile sensors, including GelSight Mini, DIGIT, DuraGel, DM-Tac W, and GelStereo

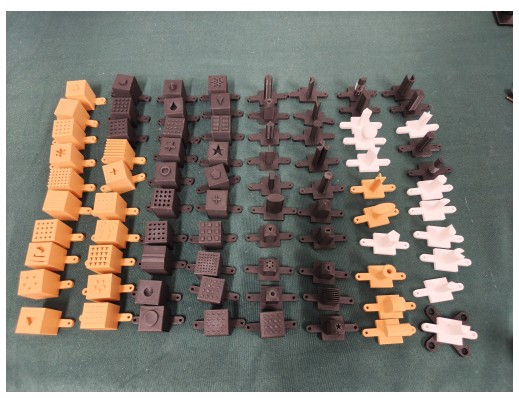

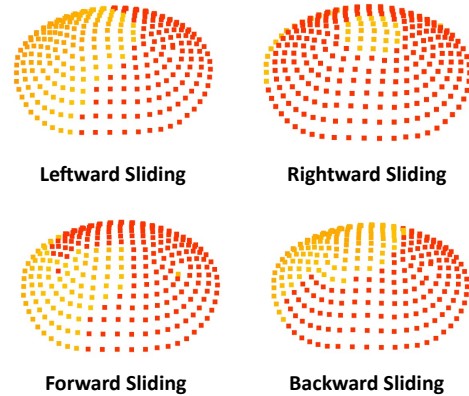

Figure 9: Illustration of the indenters. All of the indenters are made of 3D-printed materials.

Figure 10: Illustration of the 3D markers obtained as the indenter slides over the GelStereo BioTip sensor surface.

BioTip (Cui et al., 2023). Among these, DIGIT and Mini are widely used commercial sensors, Duragel is a laboratory-built sensor, and DM-Tac W and BioTip are marker-based optical sensors. Notably, BioTip is also a spherical sensor, as shown in Fig. 7 and Fig. 8. As a result, our collected touch–force paired dataset encompasses a wide variety of sensor types. We mount the five sensors on a unified base and design 71 indenters with different shapes, as shown in Fig. 9 and Tab. 4. Using a UFACTORY xArm 6 robotic arm, we performed pressing, sliding in forward, backward, left, and right directions, and lifting actions sequentially on each sensor. A six-axis force sensor is mounted on the robotic arm's wrist, enabling the collection of 3D contact forces (including both shear and normal forces) when the indenter makes contact with the sensor surface. Specifically, by tracking the marker captured by the stereo cameras inside the GelStereo BioTip sensor, we construct the 3D marker distributions on the sensor surface during the indenter pressing and sliding. Some examples are shown in Fig. 10. In addition, we provide a textual description for each sample, capturing the current motion state and indenter shape, forming a Touch–Force–Language dataset. We also locate segments in each trail where the forces along X, Y, and Z axes change smoothly, and determine the action type based on the direction of these changes. In this way, we add atomic action labels to some samples in this dataset and use them together with ToucHD (Sim) for the action matching task.

Although the sensors integrated into AnyTouch 2 are mainly planar optical tactile sensors, many non-planar and even non-optical tactile sensors are still widely used in practice. While their surface geometry (non-planar) and data representation (3D markers) differ from planar optical tactile sensors, these tactile sensors share the same fundamental principles of converting tactile signals into visual information (either 2D or 3D), indicating clear potential for further integration. Thus, the ToucHD dataset which contains both planar and non-planar optical tactile sensors can serve as a bridge between planar optical sensors and non-planar or non-optical tactile sensors, enabling future integration of a broader range of tactile sensors.

### A.3 TRAINING DATASET STATISTICS

In this section, we provide a detailed description of all the training datasets we used, including sensor types, paired modalities, sizes, and other relevant details. We use data from 9 different tactile datasets for model training, including: Touch and Go (TAG) (Yang et al., 2022), VisGel (Li et al., 2019), ObjectFolder Real (OF Real) (Gao et al., 2023) , TVL (Fu et al., 2024), YCB-Slide (Suresh et al., 2023), SSVTP (Kerr et al., 2022), Octopi (Yu et al., 2024), TacQuad (Feng et al., 2025b) and ToucHD. These datasets differ in terms of the tier in the tactile dynamic pyramid, the modalities paired with tactile data, the sensors used for collection, and the data scale. We summarize them in Tab. 5. Most of these open-source datasets are situated at the lower dynamic tiers 4 and 5, and contain a large number of contact-static frames. As a result, there is a substantial amount of redundant training data, particularly in the VisGel and ObjectFolder Real datasets. To address this issue, we compute the variance of the Laplacian for each frame relative to its preceding frame, and

Table 4: List of the 71 indenters and whether they are in ToucHD Bench.

| Index | Indenter | In ToucHD Bench | Index | Indenter | In ToucHD Bench |
|---|---|---|---|---|---|
| 1 | Semi-cylindrical | ✗ | 37 | Small Pentagrams Array | ✗ |
| 2 | Wavy Cylindrical | ✗ | 38 | Small Rectangular Bars | ✗ |
| 3 | Hexagonal | ✗ | 39 | Large Ring | ✗ |
| 4 | Isosceles Trapezoidal | ✗ | 40 | Rectangular Bar Array | ✗ |
| 5 | One-third cylindrical | ✗ | 41 | Rectangular Holes | ✗ |
| 6 | Five-sixths Cylindrical | ✗ | 42 | Star-shaped Holes | ✗ |
| 7 | Small Sphere | ✗ | 43 | Elliptical Holes | ✗ |
| 8 | Heart-shaped | ✗ | 44 | Radial Hole | ✗ |
| 9 | One-quarter Cylindrical | ✗ | 45 | Dense Circular Holes | ✗ |
| 10 | Regular Triangular | ✗ | 46 | Circular Holes | ✗ |
| 11 | Square Prism | ✗ | 47 | Irregular Holes | ✗ |
| 12 | Cylindrical | ✗ | 48 | Circular Hole Array | ✗ |
| 13 | Elliptical | ✗ | 49 | Regular Pentagonal Holes | ✗ |
| 14 | Rectangular | ✗ | 50 | Large Star-shaped Hole | ✗ |
| 15 | T-shaped | ✗ | 51 | Small Rectangular Holes | ✗ |
| 16 | U-shaped | ✗ | 52 | Teardrop-shaped Hole | ✗ |
| 17 | Cross-shaped | ✗ | 53 | Large Circular Hole | ✗ |
| 18 | Isosceles Triangular | ✗ | 54 | Cross-shaped Hole | ✗ |
| 19 | Ring-shaped | ✗ | 55 | Diamond-shaped Holes | ✗ |
| 20 | Raised Elliptical | ✗ | 56 | Dense Small Holes | ✗ |
| 21 | Five Small Spheres | ✗ | 57 | S-shaped Holes | ✗ |
| 22 | Small sphere Array | ✗ | 58 | Teardrop | ✗ |
| 23 | Square Holes | ✗ | 59 | Moon-shaped | ✗ |
| 24 | Triangular Hole | ✗ | 60 | Rectangular Bar | ✗ |
| 25 | Regular Hexagonal Hole | ✗ | 61 | Pentagram | ✗ |
| 26 | Moon-shaped Hole | ✗ | 62 | Elliptical | ✓ |
| 27 | Rectangular Holes | ✗ | 63 | Right-angled Trapezoidal | ✓ |
| 28 | Raised Small Sphere | ✗ | 64 | Small Square Array | ✓ |
| 29 | Small Ring | ✗ | 65 | Rectangular Bar | ✓ |
| 30 | Pentagram-shaped Holes | ✗ | 66 | Semicircular Hole Array | ✓ |
| 31 | Grid-like Gaps | ✗ | 67 | T-shaped Hole Array | ✓ |
| 32 | Small Trapezoid Array | ✗ | 68 | Dense Circular Holes | ✓ |
| 33 | Small Pentagons Array | ✗ | 69 | Large Triangular Hole | ✓ |
| 34 | Small ellipse Array | ✗ | 70 | Regular Octagonal | ✓ |
| 35 | Crescent-shaped | ✗ | 71 | Clover-shaped | ✓ |
| 36 | Sun-like Cylindrical | ✗ | | | |

apply a threshold to select frames that capture more informative dynamic contact events. In addition, to further reduce data redundancy and improve training efficiency, we perform interval sampling on the YCB-Slide, Touch-Slide, and ToucHD (Mani) datasets, thereby significantly reducing the overall volume of training data.

## A.4 BENCHMARK AND BASELINE DETAILS

For downstream evaluation, we adopt Touch and Go (Yang et al., 2022) and Cloth (Yuan et al., 2018) for object property understanding, and Sparsh (Higuera et al., 2025a) together with ToucHD Bench (10 unseen indenter) for dynamic physical understanding. These benchmarks cover three mainstream optical tactile sensors: GelSight (Yuan et al., 2017), DIGIT (Lambeta et al., 2020) and GelSight Mini (Inc.). Touch and Go is a dataset for material recognition, while Cloth focuses on clothing texture classification. Each of them contains 20 categories. Sparsh Bench comprises

Table 5: Training dataset statistics. V: Vision. L: Language. F: Force.

| Dataset | Dynamic Tier | Paired Modalities | Sensor | Total Size | Used Size |
|---|---|---|---|---|---|
| Touch and Go (Yang et al., 2022) | Tier 5 | V, L | GelSight | 250k | 250k |
| VisGel (Li et al., 2019) | Tier 5 | V | GelSight | 587k | 121k |
| TVL (Fu et al., 2024) | Tier 5 | V, L | DIGIT | 39k | 39k |
| SSVTP (Kerr et al., 2022) | Tier 5 | V, L | DIGIT | 4.5k | 4.5k |
| YCB-Slide (Suresh et al., 2023) | Tier 4 | / | DIGIT | 183k | 91k |
| Touch-Slide (Higuera et al., 2025a) | Tier 4 | / | DIGIT | 180k | 81k |
| ObjectFolder Real (Gao et al., 2023) | Tier 5 | V, L | GelSlim | 1165k | 71k |
| Octopi (Yu et al., 2024) | Tier 4 | L | GelSight Mini | 39k | 39k |
| TacQuad (Feng et al., 2025b) | Tier 4 | V, L | GelSight, DIGIT GelSight Mini DuraGel | 55k | 47k |
| ToucHD (Sim) | Tier 3 | / | GelSight, DIGIT GelSight Mini GelSlim DuraGel | 1119k | 252k |
| ToucHD (Mani) | Tier 2 | V | DIGIT, DuraGel GelSight Mini | 585k | 182k |
| ToucHD (Force) | Tier 1 | L, F | DIGIT, DuraGel GelSight Mini | 722k | 248k |

three dynamic perception tasks: force prediction, slip detection, and pose estimation. For the force prediction task, the training set consists of data collected with sphere and sharp indenters, while data from the unseen flat indenter is used for testing. In the slip detection task, an additional objective is included, namely predicting the total contact force change over tactile frames. For all Sparsh tasks, we use the official data splits.

Due to the limited diversity of indenter shapes in Sparsh Bench, we additionally select 10 probes from the full set of 71 collected touch–force paired probes to form the ToucHD Bench dataset (which is not included in the pre-training data), as shown in Tab. 4. Among these, 7 indenters are used for training and 3 for testing (Right-angled Trapezoidal, Small Square Array and Large Triangular Hole indenters). This setup allows us to more comprehensively evaluate the model's perception of force-related physical properties through the force prediction task.

We compare our AnyTouch 2 model with several representative tactile representation learning frameworks: UniTouch (Yang et al., 2024) and T3 (Zhao et al., 2025b), which use single tactile images as input, as well as MAE (Sparsh) (Higuera et al., 2025a), VJEPA (Sparsh) (Higuera et al., 2025a) and AnyTouch 1 (Feng et al., 2025b), which leverage multiple consecutive frames as input. UniTouch implicitly integrates multi-sensor representations into a unified space through tactile–visual alignment and learns tactile properties from vision. T3 is a multi-task, multi-sensor joint training framework in which all sensors share a common chunk. During training, both methods take single-frame tactile images as input and cannot directly handle multiple consecutive tactile frames. Therefore, when feeding two consecutive tactile frames to these models, we unfold them along the batch dimension. MAE (Sparsh) and VJEPA (Sparsh) are two visual self-supervised learning models trained on tactile data in (Higuera et al., 2025a). They take 2 frames and 4 frames as input, respectively, and thus possess preliminary dynamic perception capabilities. However, since their training data constitute only a subset of AnyTouch 2, to fairly compare and simultaneously evaluate the benefits of our ToucHD dataset, we additionally trained an MAE (Sparsh)† model on the same training data including ToucHD as AnyTouch 2 to serve as a baseline.

## A.5 IMPLEMENTATION DETAILS

We build our encoders on top of OpenCLIP-Base (Cherti et al., 2023). For the tactile decoder, we adopt a Vision Transformer (ViT) (Dosovitskiy et al., 2020) with 6 layers, 8 attention heads, and a hidden dimension of 512. Model optimization is performed using AdamW (Loshchilov, 2017) with a learning rate of $3 \times 10^{-4}$ and a batch size of 64. After a warm-up of 1 epoch, we apply a linear decay

schedule to the learning rate. All pre-training experiments are conducted on 4 NVIDIA H100 GPUs. For most tactile sensors operating at 30 Hz, we subsample every other frame and use a sequence of $N = 4$ frames as the input at time step $t$, *i.e.*, $\mathbf{T} = (T_{t-6}, T_{t-4}, T_{t-2}, T_t)$. For the GelSight Mini, which operates at approximately 18 Hz, we instead use four consecutive frames as input. In the two masked reconstruction tasks, we set the masking ratio to $\rho = 0.75$. During the alignment, we use alignment strengths of $\alpha_{TV} = \alpha_{TL} = 0.2$. The model is trained for a total of 40 epochs: at epoch 20, we introduce cross-sensor matching, action matching, and force prediction tasks ($i_{\text{Match}} = i_{\text{Force}}$), and at epoch 30, we further incorporate the aligning task ($i_{\text{Align}}$). The maximum task weights are set to $\lambda_{\text{Align}}^{\max} = 1.0$, $\lambda_{\text{Match}}^{\max} = 0.02$, and $\lambda_{\text{Force}}^{\max} = 0.1$. Following (Feng et al., 2025b), we use $L = 5$ sensor tokens for each type of sensor, with the probability of using universal sensor tokens increasing linearly from 0 to 0.75. During training, if a sample lacks the label required for a specific training objective, it is excluded from the loss computation for that objective. Since completing matching tasks requires feeding both positive and negative samples into the encoder simultaneously, we fix the proportion of samples in each batch that participate in the matching tasks to stabilize GPU memory usage. For the Cloth Task, Sparsh Bench, and ToucHD Bench, we freeze the tactile encoder and evaluate its representations using an attentive probe, following (Higuera et al., 2025a). In TAG and Cloth tasks, we input consecutive $N = 4$ frames ($T_{t-3}, T_{t-2}, T_{t-1}, T_t$. to our AnyTouch 2 models. For other dynamic models, we input $N = 2$ frames ($T_{t-3}, T_t$. For the static models that only accept single-frame input, we ensured a fair comparison by processing the same $N = 2$ frames for these models. Specifically, the $N$ frames were temporally unfolded into a batch of $B \times N$ independent samples for the static model. The final prediction was then obtained by averaging the output features across all $N$ frames for each original sample.

## A.6 REAL-WORLD TASK SETUP

To evaluate the practical effectiveness of our model, we design a set of four real-world manipulation tasks that comprehensively cover the dynamic tactile capabilities defined by our tactile dynamic pyramid. Each task targets different levels of tactile perception, ranging from object-level property understanding to fine-grained, force-sensitive dexterous manipulation:

**Tactile Grasping (Tier 5: Basic Tactile Properties).** In this task, the robot is required to grasp small balls of two different materials and textures and place them into the corresponding boxes. Successful completion demands an accurate perception of object tactile properties such as material stiffness, hardness, and surface texture during manipulation. A particular challenge arises from one ball's smooth surface, which requires the robot to continuously monitor fine-grained deformation feedback and adapt its gripping force in real time to prevent slippage. Furthermore, hesitation or oscillations in movement direction can destabilize the grasp and lead to dropping the ball. This task therefore evaluates the model's ability to differentiate objects based on static tactile attributes and leverage contact cues for stable manipulation in dynamic settings. We collect 50 human trajectories, with synchronized vision and tactile data recorded as task inputs.

**Whiteboard Wiping (Tier 4 & 3: Action-Specific Dynamics).** In this task, the robot must use an eraser to wipe letters off a whiteboard until the surface is completely clean. The process involves structured contact interactions characterized by directional motions and temporally evolving tactile feedback. A key challenge is that the robot has only a single opportunity to complete the wiping action: if the applied force is inadequate, the letters cannot be fully erased, leaving no chance for correction. This strict one-shot requirement forces the model to precisely perceive action-specific tactile cues (e.g., sliding direction and applied pressure) and to adapt its wiping motion dynamically throughout execution. It evaluates the model's capacity for action-specific understanding during manipulation. Since the dynamic perception capabilities corresponding to Tier 4 and Tier 3 are typically coupled in real-world manipulation, we integrate these two tiers for joint evaluation. We collect 50 human trajectories, simultaneously recording vision and tactile data as task inputs.

**USB Insertion (Tier 2: Complex Manipulation Dynamics).** In this task, the robot must extract a USB connector from one port and insert it into another. The manipulation involves complex, multi-directional deformations during both insertion and removal, and is particularly challenging due to the extremely small tolerance of USB sockets for misalignment. A further difficulty arises from the fact that collisions during extraction or re-insertion may alter the pose of the USB connector, requiring the robot to continuously monitor subtle deformation feedback and dynamically adjust its motion strategy in real time. Success depends on accurately perceiving the subtle temporal changes

in contact and adapting to dynamic shifts in alignment, thereby testing the model's ability to process continuous tactile deformations during the manipulation process. We collect 50 human trajectories, with synchronized vision and tactile data recorded as task inputs.

**Chip Moving (Tier 1: Force-Sensitive Manipulation Dynamics).** Here, the robot delicately picks up a single chip from the top of a bottle and transfers it to another bottle, ensuring the chip remains intact. This task involves small displacements between the chip and the sensor and requires extreme sensitivity to minute force variations and precise dynamic control during contact. During manipulation, visual observations are partially occluded, and the robotic arm must primarily rely on tactile feedback to control the gripper's closure and the downward placement depth, in order to prevent the chips from being crushed. It primarily tests the model's capacity for high-resolution, force-aware tactile perception and fine-grained dexterous manipulation. Since the surface of the DIGIT sensor is relatively rigid, the deformation is not clearly visible when grasping the chip with small forces. Therefore, we only use the GelSight Mini for testing in this task. We collect 50 human trajectories, simultaneously recording vision and tactile data as task inputs.

In the Tactile Grasping and Chip Moving tasks, the gripper of the robotic arm does not initially grasp the object but instead maintains a certain distance from it. This is because determining tactile attributes and grasping fragile objects based on tactile inputs at different contact locations is itself the core challenge of these two tasks. In contrast, for the Whiteboard Wiping and USB Insertion tasks, the primary role of the tactile modality lies in the manipulations performed after the object has been grasped, rather than in the grasping action itself. Therefore, in these two tasks, the gripper starts by firmly holding the object to be manipulated. Moreover, among the four real-world manipulation tasks, three of them inherently involve slip dynamics, including Whiteboard Wiping, USB Insertion, and Chip Moving. These displacements are subtle but critical, and cannot be perceived easily using force sensors. In summary, these four tasks provide a comprehensive evaluation of the model's ability to capture material properties, surface textures, fine-grained geometric details, and rich contact dynamics arising in real-world manipulation.

For the Tactile Grasping, Whiteboard Wiping, and USB Insertion tasks, experiments are conducted using the AGILEX Piper robotic arm equipped with GelSight Mini and DIGIT sensors on the fingertips. The Chip Moving task is performed on the uFactory xArm 6 for higher precision and embodiment diversity with GelSight Mini sensors on the fingertips, enabling comprehensive evaluation across different embodiments and sensor types. In each scenario, a third-person camera records visual information. For all real-world manipulation tasks, we used a frozen ImageNet-pretrained (Deng et al., 2009) ResNet-50 (He et al., 2016) as the visual encoder. We use an UNet-based Diffusion Policy (Chi et al., 2023) as our policy head and freeze all the tactile encoders during training. The diffusion policy adopted UNet channel sizes of [128,256,512], a positional encoding size of 256, a kernel size of 5, and 8 GroupNorm (Wu & He, 2018) groups. As the tactile encoder produces a large number of tokens, directly training the policy network on the full token sequence could bring unacceptable costs of GPU memory and time. Hence, we inserted a trainable attentive pooler between each tactile encoder and the diffusion policy. The pooler uses 30 learnable query tokens to extract information from the full tactile token sequence via cross-attention. These 30 pooled tokens then replace the full tactile token sequence as the input to the policy network and are concatenated with the visual features after flattening. We trained the policy network using the AdamW optimizer with a learning rate of $1 \times 10^{-4}$, for a total of 100 epochs and a batch size of 64. For each task, we randomly sampled 8 trajectories out of 50 as the validation set, and the model with the lowest validation loss was used for real-world evaluation. Due to the high real-time requirements of these tasks, we adopt an action chunking horizon of 8 and predict actions at a frequency of 3 Hz, executing only the first 2 actions at each inference step.

## A.7 MULTI-MODAL ALIGNING LOSS

Following (Feng et al., 2025b), we maximize the utilization of paired data by selecting, within each batch, the largest available subset for every modality combination to perform multi-modal alignment. Specifically, let $(x_T, x_V, x_L)$ denote uni-modal representations obtained from their respective encoders, where $x_T \in \mathbb{R}^d$ is the tactile representation, $x_V \in \mathbb{R}^d \cup \varnothing$ is the visual representation, and $x_L \in \mathbb{R}^d \cup \varnothing$ is the textual representation. We then conduct multi-modal alignment (Radford et al., 2021) within the batch as:

$$\mathcal{L}_{T \to V} = -\frac{1}{|\Omega_V|} \sum_{i \in \Omega_V} \log \frac{\exp(x_{T,i}^\top \cdot x_{V,i}/\tau)}{\sum_{j \in \Omega_V} \exp(x_{T,i}^\top \cdot x_{V,j}/\tau)},$$

$$\mathcal{L}_{V \to T} = -\frac{1}{|\Omega_V|} \sum_{i \in \Omega_V} \log \frac{\exp(x_{V,i}^\top \cdot x_{T,i}/\tau)}{\sum_{j \in \Omega_V} \exp(x_{V,i}^\top \cdot x_{T,j}/\tau)},$$

$$\mathcal{L}_{T \to L} = -\frac{1}{|\Omega_L|} \sum_{i \in \Omega_L} \log \frac{\exp(x_{T,i}^\top \cdot x_{L,i}/\tau)}{\sum_{j \in \Omega_L} \exp(x_{T,i}^\top \cdot x_{L,j}/\tau)},$$

$$\mathcal{L}_{L \to T} = -\frac{1}{|\Omega_L|} \sum_{i \in \Omega_L} \log \frac{\exp(x_{L,i}^\top \cdot x_{T,i}/\tau)}{\sum_{j \in \Omega_L} \exp(x_{L,i}^\top \cdot x_{T,j}/\tau)}.$$

$$(8)$$

Here, $B$ denotes the batch size, $\Omega_V$ and $\Omega_L$ are the index sets corresponding to samples that contain visual and textual inputs, respectively, and $\tau$ is the temperature parameter. Finally, the overall multi-modal alignment loss is defined as the weighted sum of all directional objectives:

$$\mathcal{L}_{\text{Align}} = \frac{\alpha_{TV}}{2}(\mathcal{L}_{T \to V} + \mathcal{L}_{V \to T}) + \frac{\alpha_{TL}}{2}(\mathcal{L}_{T \to L} + \mathcal{L}_{L \to T}),$$

$$(9)$$

where $\alpha_{TV}, \alpha_{TL}$ are hyper-parameters to control the aligning strength.

## A.8 FORCE PREDICTION EVALUATION

To provide a more intuitive comparison of the performance of different baselines and our AnyTouch 2 model on the force prediction task in ToucHD Bench, we visualize the 3D force probe results of all models on the DIGIT and GelSight Mini subsets. The results are shown in Fig. 11, 12, 13, 14, 15, and 16. Although the T3 model is pre-trained on a large amount of tactile data, this data comes from the lower tiers (Tier 4 and 5) of the tactile dynamics pyramid and does not involve training with consecutive frames for dynamic perception. Consequently, the model shows no advantage over the CLIP model without tactile pre-training in the force prediction task. Compared with the CLIP model, the prediction results of MAE(Sparsh) and VJEPA(Sparsh), which take multi-frame inputs, are noticeably more accurate. However, they still exhibit considerable bias in predicting tangential forces along the X and Y directions, indicating insufficient perception of sliding dynamics. For the AnyTouch series, the AnyTouch 1 model, which primarily focuses on static tactile features, achieves relatively accurate predictions in the Z-axis normal direction but performs poorly on tangential force prediction in the X and Y directions. In contrast, our AnyTouch 2 model, equipped with multi-level dynamic enhanced modules that incorporate force-related tactile dynamics and trained on the higher-tier ToucHD dataset, demonstrates superior performance on our ToucHD Bench, achieving precise force prediction across all three directions.

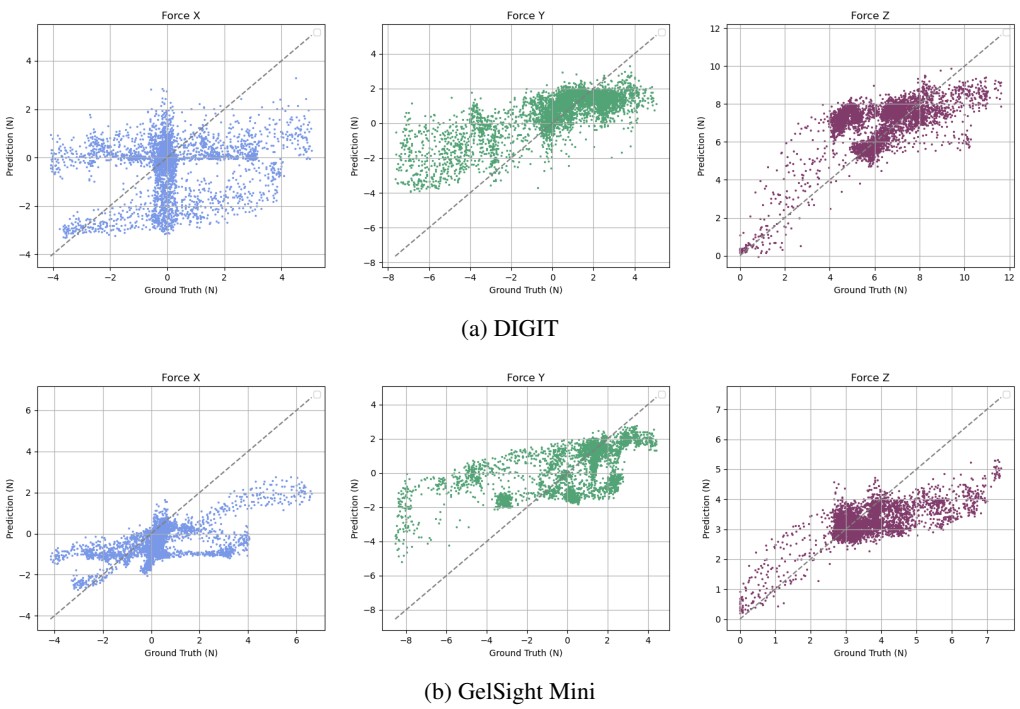

(a) DIGIT

(b) GelSight Mini

Figure 11: 3D Force Probe Results of **CLIP** on ToucHD Bench Force Prediction.

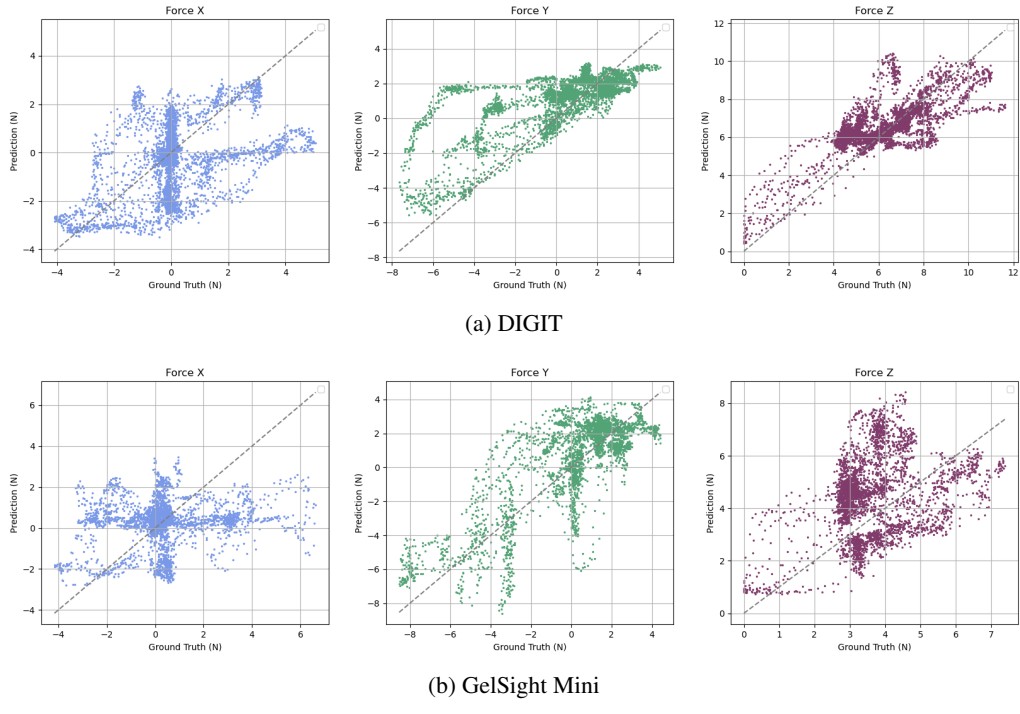

(a) DIGIT

(b) GelSight Mini

Figure 12: 3D Force Probe Results of **T3** on ToucHD Bench Force Prediction.

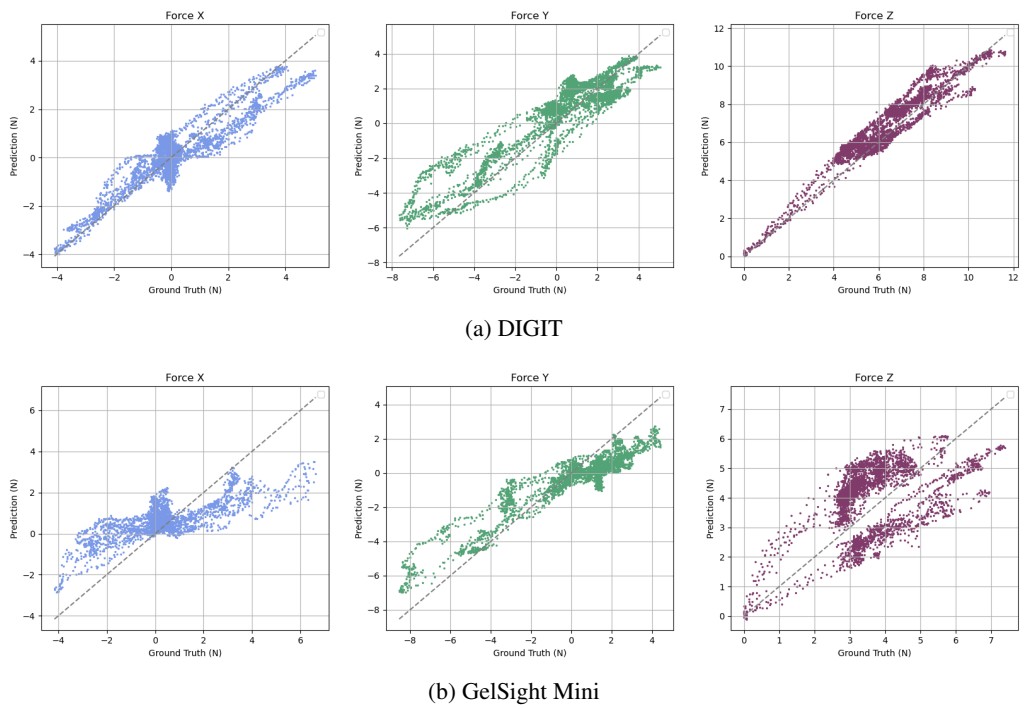

(a) DIGIT

(b) GelSight Mini

Figure 13: 3D Force Probe Results of **MAE (Sparsh)** on ToucHD Bench Force Prediction.

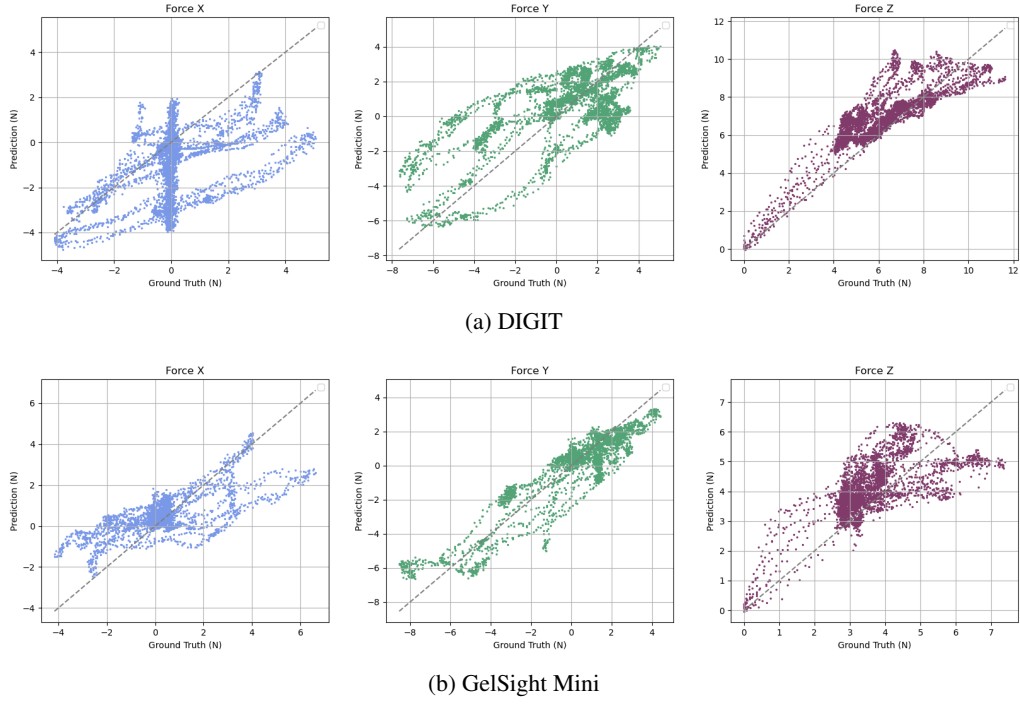

(a) DIGIT

(b) GelSight Mini

Figure 14: 3D Force Probe Results of **VJEPA (Sparsh)** on ToucHD Bench Force Prediction.

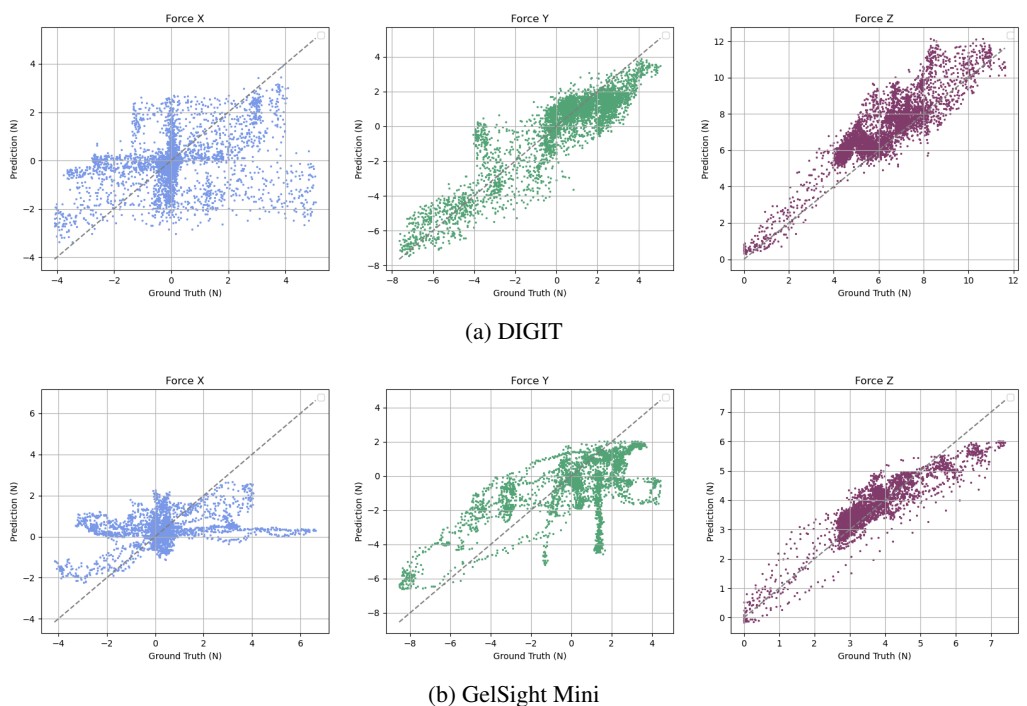

(a) DIGIT

(b) GelSight Mini

Figure 15: 3D Force Probe Results of **AnyTouch 1** on ToucHD Bench Force Prediction.

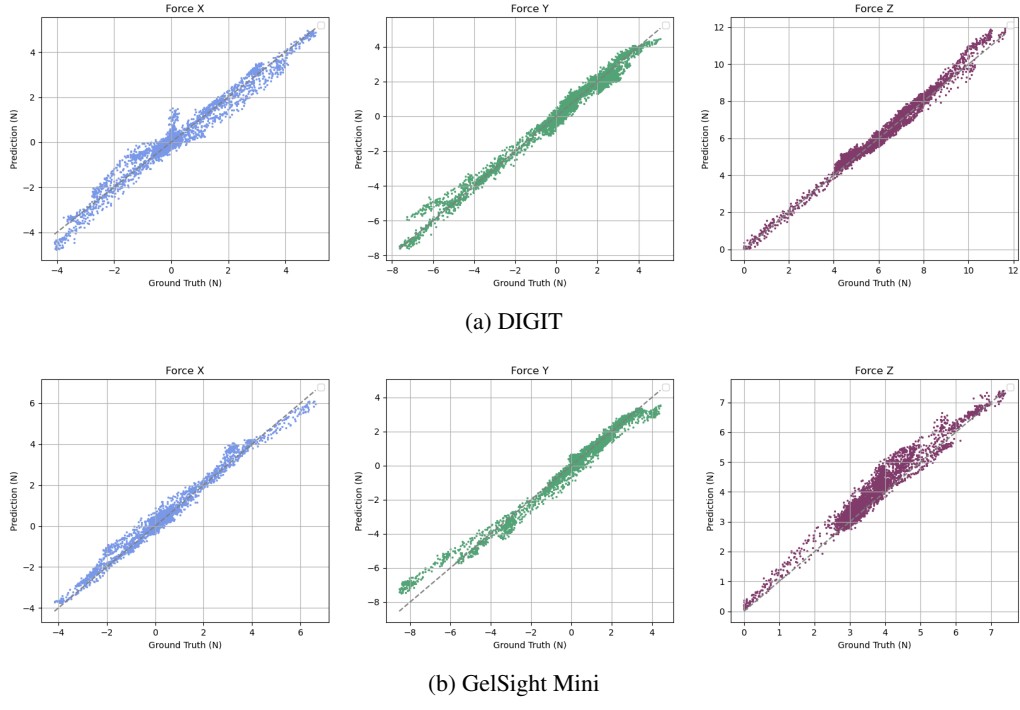

(a) DIGIT

(b) GelSight Mini

Figure 16: 3D Force Probe Results of **AnyTouch 2** on ToucHD Bench Force Prediction.

Table 6: Comparison between AnyTouch 2 trained with our pyramid-driven strategy and non–pyramid-driven baselines.

| Method | Data Size | Object Bench | | Sparsh Bench | | | | ToucHD Bench | |
|---|---|---|---|---|---|---|---|---|---|
| | | TAG | Cloth | Slip (Delta Force) | | Force | | Force | |
| | | Acc(↑) | Acc(↑) | F1 Score(↑) / RMSE(↓) | | RMSE(↓) | | RMSE(↓) | |
| | | GS | GS | DG | Mini | DG | Mini | DG | Mini |
| **AnyTouch 2** | 248k | **69.97** | **40.82** | **85.13 / 94.34** | **97.80 / 106.89** | **679.77** | **232.45** | **960.58** | **1153.19** |
| → Tier 4&5 Only | 744k | 68.92 | 40.39 | 84.16 / 110.68 | 97.67 / 136.36 | 783.64 | 257.95 | 2448.89 | 2982.46 |
| → Tier 1 Only | 248k | 61.81 | 36.62 | 84.45 / 98.26 | 97.60 / 115.32 | 699.12 | 240.26 | 987.91 | 1172.69 |
| - task scheduling | 248k | 69.24 | 39.91 | 76.92 / 100.34 | 97.20 / 139.13 | 690.39 | 252.68 | 1023.67 | 1342.75 |

Table 7: The impact of ToucHD in AnyTouch 2 on real-world manipulation tasks.

| Method | Dynamic Tier | Tier 5 Tactile Grasping | | Tier 4 & 3 Whiteboard Wiping | | Tier 2 USB Insertion | | Tier 1 Chip Moving |
|---|---|---|---|---|---|---|---|---|
| | | DG | Mini | DG | Mini | DG | Mini | Mini |
| **AnyTouch 2** | **Tier 1** | **0.75** | **0.80** | **0.85** | **0.80** | **0.30** | **0.25** | **0.85** |
| - ToucHD | Tier 4 | 0.70 | 0.75 | 0.75 | 0.70 | 0.20 | 0.15 | 0.70 |

## A.9 ABLATION STUDY

In the ablation study shown in Tab. 2, we found that removing the multi-modal alignment module leads to a significant performance drop on the two material understanding datasets in Object Bench, but it improves performance on most of the dynamic physical perception datasets. This is due to the substantial difference in label granularity between the two types of tasks. The text labels currently used for multi-modal alignment contain only coarse-grained object attributes, such as general shape, material, hardness, and roughness, but they do not include fine-grained physical quantities related to contact, such as contact force or pressing speed. As a result, an obvious consequence arises: during multi-modal alignment, samples of the same object pressed with different forces are pulled closer together. This is undesirable for downstream tasks that require distinguishing between different levels of contact force. This issue is actually common in CLIP-style vision–language alignment paradigms (Maninis et al., 2024; Jing et al., 2024; Xie et al.). As the text labels are coarse-grained, multi-modal alignment can lead to suboptimal fine-grained visual perception.

In training AnyTouch 2, two key components are directly guided by the Tactile Dynamic Pyramid: (1) We deliberately select training data that span all tiers of the pyramid. (2) Our task scheduling strategy coordinates the learning of the multi-modal alignment (mainly Tier 4+5 data), action matching (mainly Tier 3 data), and force prediction modules (mainly Tier 1 data). Therefore, to compare pyramid-driven training against training without tiers and thereby demonstrate the value of the Tactile Dynamic Pyramid, we conducted evaluations on four different models: (1) AnyTouch 2 trained on a randomly sampled subset of 248k samples from the full training dataset. This model represents a pyramid-driven method, trained on a data size comparable to that of the other baselines for a fair comparison. (2) AnyTouch 2 trained using only Tier 4+5 data (744k samples in total). This baseline represents the mainstream paradigm of tactile representation learning before the introduction of our Tactile Dynamic Pyramid and the ToucHD dataset. (3) AnyTouch 2 trained using only Tier 1 data (248k samples in total). This baseline corresponds to the unified model on pooled data with task-specific supervision. (4) AnyTouch 2 without task scheduling strategy (248k samples in total). This baseline represents a training setup in which the model does not follow the pyramid-guided, tier-by-tier task curriculum. Instead, all training objectives are activated and optimized jointly from the very beginning of training. We conducted comprehensive comparisons across all offline benchmarks, and the results are presented in Tab. 6. The results demonstrate three key findings: (1) The baseline trained only on Tier 4+5 data performs substantially worse than AnyTouch 2 across all tasks, highlighting the importance of high-tier data (such as our ToucHD dataset) emphasized by the tactile dynamic pyramid. (2) The baseline trained only on Tier 1 data for force prediction tasks fails to outperform AnyTouch 2 on any force-related tasks in Sparsh Bench or ToucHD Bench. This indicates that the tactile dynamic pyramid provides essential guidance on the comprehensive use of training data across tiers. (3) The baseline trained without our task scheduling strategy also under-

Table 8: The impact of ToucHD (Mani) in AnyTouch 2 on real-world manipulation tasks.

| Method | Tier 2 USB Insertion | | Tier 1 Chip Moving |
|---|---|---|---|
| | DG | Mini | Mini |
| **AnyTouch 2** | **0.35** | **0.30** | **0.80** |
| - ToucHD (Mani) | 0.25 | 0.25 | 0.70 |

Table 9: The impact of ToucHD (Force) and shear force labels in AnyTouch 2 on offline benchmarks.

| Frame Number | Object Bench | | Sparsh Bench | | | | ToucHD Bench | |
|---|---|---|---|---|---|---|---|---|
| | TAG | Cloth | Slip (Delta Force) | | Force | | Force | |
| | Acc(↑) | Acc(↑) | F1 Score(↑) / RMSE(↓) | | RMSE(↓) | | RMSE(↓) | |
| | GS | GS | DG | Mini | DG | Mini | DG | Mini |
| **AnyTouch 2** | **76.97** | **42.31** | **86.66 / 87.80** | **97.96 / 80.83** | **624.26** | **202.14** | **894.32** | **1051.03** |
| -Shear Force | 76.32 | 42.18 | 86.08 / 97.25 | 97.89 / 96.33 | 675.83 | 232.34 | 1329.28 | 1707.41 |
| -ToucHD (Force) | 74.33 | 40.87 | 84.91 / 107.43 | 97.85 / 109.37 | 777.41 | 266.43 | 1792.49 | 2424.68 |

performs AnyTouch 2 on all benchmarks, demonstrating the value of the tactile dynamic pyramid in guiding the design of model training strategy. Together, these results underscore the valuable guidance provided by our proposed tactile dynamic pyramid.

We also evaluate the contribution of the ToucHD dataset to dynamic perception in real-world manipulation tasks. As shown in Tab. 7, when the ToucHD dataset is removed from the training data, the model consistently exhibits performance drops across all manipulation tasks, particularly for higher-tier tasks (Tier 1, 2, and 3) that correspond to the ToucHD dataset. As our ToucHD (Mani) is the only Tier 2 dataset and is supposed to contains a large number of tactile dynamic deformation patterns commonly observed during real manipulation. Hence, to quantify the contribution of this Tier 2 data to manipulation performance, we conducted ablation studies on the USB Insertion and Chip Moving tasks. Both tasks involve a variety of dynamic deformation patterns drawn from the 46 tasks in ToucHD (Mani), including pressing, sliding, interactions with fragile objects, and overcoming stiction. The results are presented in Tab. 8. The results show that removing the only Tier 2 dataset from the training data leads to a performance drop in both manipulation tasks. This demonstrates that incorporating Tier 2 training data indeed enables the model to better perceive the dynamic tactile characteristics that arise during real-world manipulation.

To further evaluate the impact of Tier 1 data and its shear force labels on downstream tasks, we conducted comparisons across all offline benchmarks and two real-world manipulation tasks. The results are presented in Tab. 9 and 10. The results show that Tier 1 data, along with its shear force labels, make a clear contribution to various downstream dynamic perception tasks, including force prediction and real-world manipulation tasks. The consistent performance drop observed when removing the shear force labels further highlights the importance of collecting and predicting both shear and normal forces simultaneously. To more precisely quantify the contribution of the shear force labels in the ToucHD (Force) dataset to downstream 3D force prediction tasks—including both shear and normal forces, we also report the prediction errors along each force direction on Sparsh Bench and ToucHD Bench after removing either the ToucHD (Force) dataset entirely or only its shear force labels. The results are shown in Tab. 11 and 12. The results show that removing the shear force labels has a significant impact on the prediction performance along the x and y axes (shear forces) in downstream tasks. Training the model using only z-axis (normal force) labels is clearly insufficient to support accurate shear force prediction. This highlights the advantage of our ToucHD dataset in providing both normal and shear force labels.

Another important parameter of the model is the number of input frames $N$. To evaluate the impact of the input frame count on the model's dynamic perception capability, we compare the performance of our AnyTouch 2 model using 4-frame inputs versus using only 2-frame inputs across all benchmarks. As shown in Tab. 13, the model using 4-frame inputs outperforms the 2-frame variant across all tasks, indicating that denser dynamic tactile information benefits dynamic tactile perception. However, this also presents a trade-off with computational cost, as using 4 frames nearly doubles the token sequence length compared to 2 frames.

Table 10: The impact of ToucHD (Force) and its shear force labels in AnyTouch 2 on real-world manipulation tasks.

| Method | Tier 2 USB Insertion | | Tier 1 Chip Moving |
|---|---|---|---|
| | DG | Mini | Mini |
| **AnyTouch 2** | **0.35** | **0.30** | **0.80** |
| - ToucHD (Force) | 0.25 | 0.20 | 0.70 |
| - Shear Force in ToucHD (Force) | 0.25 | 0.25 | 0.75 |

Table 11: Contribution of ToucHD (Force) and its shear force labels on the shear force and normal force prediction tasks on Sparsh Bench

| Method | Sparsh Bench | | | | | |
|---|---|---|---|---|---|---|
| | Force (RMSE(↓)) | | | | | |
| | DG | | | Mini | | |
| | X (Shear) | Y (Shear) | Z (Normal) | X (Shear) | Y (Shear) | Z (Normal) |
| **AnyTouch 2** | **268.03** | **161.57** | **194.66** | **42.29** | **57.31** | **102.54** |
| -Shear Force | 302.58 | 176.60 | 196.65 | 58.18 | 67.73 | 106.43 |
| -ToucHD (Force) | 308.18 | 215.56 | 253.67 | 76.52 | 78.59 | 111.32 |

Table 12: Contribution of ToucHD (Force) and its shear force labels on the shear force and normal force prediction tasks on ToucHD Bench.

| Method | ToucHD Bench | | | | | |
|---|---|---|---|---|---|---|
| | Force (RMSE(↓)) | | | | | |
| | DG | | | Mini | | |
| | X (Shear) | Y (Shear) | Z (Normal) | X (Shear) | Y (Shear) | Z (Normal) |
| **AnyTouch 2** | **285.89** | **334.37** | **274.06** | **273.48** | **397.17** | **380.38** |
| -Shear Force | 433.24 | 594.77 | 301.26 | 443.54 | 820.00 | 443.87 |
| -ToucHD (Force) | 530.30 | 708.32 | 553.86 | 730.16 | 1012.12 | 682.40 |

Table 13: The impact of the number of input frames in AnyTouch 2 on offline benchmarks.

| Frame Number | Object Bench | | Sparsh Bench | | | | ToucHD Bench | |
|---|---|---|---|---|---|---|---|---|
| | TAG | Cloth | Slip (Delta Force) | | Force | | Force | |
| | Acc(↑) | Acc(↑) | F1 Score(↑) / RMSE(↓) | | RMSE(↓) | | RMSE(↓) | |
| | GS | GS | DG | Mini | DG | Mini | DG | Mini |
| **4 Frames** | **76.97** | **42.31** | **86.66** / 87.80 | **97.96** / 80.83 | **624.26** | **202.14** | **894.32** | **1051.03** |
| 2 Frames | 74.15 | 40.76 | 86.60 / **83.15** | 97.85 / 89.21 | 643.91 | 208.41 | 1076.33 | 1311.27 |

Table 14: Hyper-parameter study on $\lambda_{\text{task}}^{\text{max}}$ and $i_{\text{task}}$

| Parameter | Value | Object Bench Cloth Acc($\uparrow$) GS | Sparsh Bench Force RMSE($\downarrow$) DG | Sparsh Bench Force RMSE($\downarrow$) Mini | ToucHD Bench Force RMSE($\downarrow$) DG | ToucHD Bench Force RMSE($\downarrow$) Mini |
|---|---|---|---|---|---|---|
| $\lambda_{\text{Align}}^{\text{max}}$ | 0.5 | 41.97 | **621.95** | 204.98 | 902.54 | 1076.93 |
| | **1.0** | **42.31** | 624.26 | **202.14** | **894.32** | **1051.03** |
| | 2.0 | 40.13 | 644.81 | 220.53 | 1054.97 | 1195.62 |
| | 4.0 | 39.49 | 671.99 | 235.64 | 1096.87 | 1265.71 |
| $\lambda_{\text{Match}}^{\text{max}}$ | 0.01 | 42.01 | 631.75 | 206.98 | 913.89 | 1084.42 |
| | **0.02** | **42.31** | **624.26** | **202.14** | **894.32** | **1051.03** |
| | 0.05 | 41.67 | 640.46 | 210.93 | 907.52 | 1077.11 |
| | 0.1 | 41.92 | 635.89 | 215.67 | 910.13 | 1085.62 |
| $\lambda_{\text{Force}}^{\text{max}}$ | 0.05 | 42.15 | 630.69 | 210.82 | 923.38 | 1079.62 |
| | **0.1** | **42.31** | **624.26** | **202.14** | **894.32** | 1051.03 |
| | 0.2 | 41.71 | 645.93 | 220.77 | 900.90 | **1012.45** |
| $i_{\text{Align}}$ | 20 (Epoch) | 41.55 | 657.13 | 232.35 | 1036.99 | 1231.94 |
| | **30 (Epoch)** | **42.31** | **624.26** | **202.14** | **894.32** | **1051.03** |
| $i_{\text{Match}}$ | 10 (Epoch) | 41.24 | 632.98 | 207.16 | 909.88 | 1072.65 |
| | **20 (Epoch)** | **42.31** | **624.26** | **202.14** | **894.32** | **1051.03** |
| | 30 (Epoch) | 41.89 | 635.17 | 205.72 | 916.25 | 1085.28 |
| $i_{\text{Force}}$ | 10 (Epoch) | 42.12 | 670.82 | 225.76 | 925.76 | **1048.75** |
| | **20 (Epoch)** | **42.31** | **624.26** | 202.14 | **894.32** | 1051.03 |
| | 30 (Epoch) | 42.01 | 651.79 | **196.09** | 936.41 | 1206.84 |

## A.10 HYPER-PARAMETER STUDY

Our training process incorporates multiple objectives and brings additional hyper-parameters $\lambda_{\text{task}}^{\text{max}}$ and $i_{task}$. To evaluate the stability of our model under hyper-parameter variations, we train and evaluate the model under different hyper-parameter settings. Specifically, we fix the setting of $\lambda_{\text{Align}}^{\text{max}} = 1.0, \lambda_{\text{Match}}^{\text{max}} = 0.02, \lambda_{\text{Force}}^{\text{max}} = 0.1, i_{Align} = 30(\text{Epoch}), i_{Match} = 20(\text{Epoch}), i_{Force} = 20(\text{Epoch})$ as the anchor configuration, and in each experiment we change only one hyper-parameter at a time. The evaluation results are presented in Tab. 14. The findings indicate that our model is not sensitive to these hyper-parameters: although minor fluctuations and noticeable peaks exist, within each parameter's feasible range, our model consistently outperforms the baselines.

We also present the full training loss curves for each task and the overall objective in our AnyTouch 2 model in Fig. 17. It can be observed that the loss for each training objective decreases smoothly. We also observe that the Force Loss $\mathcal{L}_{\text{Force}}$ and Matching Loss $\mathcal{L}_{\text{Match}}$ occasionally become zero for a few iterations after these tasks start. This behavior is expected and stems from how we designed the sampler to stabilize GPU memory usage, as described in Appendix A.4. Since the matching task requires feeding both positive and negative samples into the encoder simultaneously, we modified the sampler to fix the proportion of ToucHD (Force), ToucHD (Sim), and TacQuad samples used for matching in each batch. However, because the sampler prioritizes satisfying the matching sample ratio, a small number of batches (similar to "the last batch") may end up containing no matching samples, resulting in zero Matching Loss and Force Loss. Importantly, this does not affect training because such batches are extremely rare, and when using multi-GPU training, the probability that all GPUs simultaneously receive a batch without these samples is negligible.

## A.11 CROSS-SENSOR GENERALIZATION

To validate our model's ability to extract sensor-invariant features, we conduct additional experiments on the USB Insertion task by switching the tactile sensor at test time. Specifically, we evaluated two settings: (1) fine-tuning the policy network using GelSight Mini data but testing with a DIGIT sensor, and (2) fine-tuning with DIGIT data but testing on a GelSight Mini sensor. We compare our AnyTouch 2 model with T3 baseline and both of the encoders are frozen during fine-tuning.

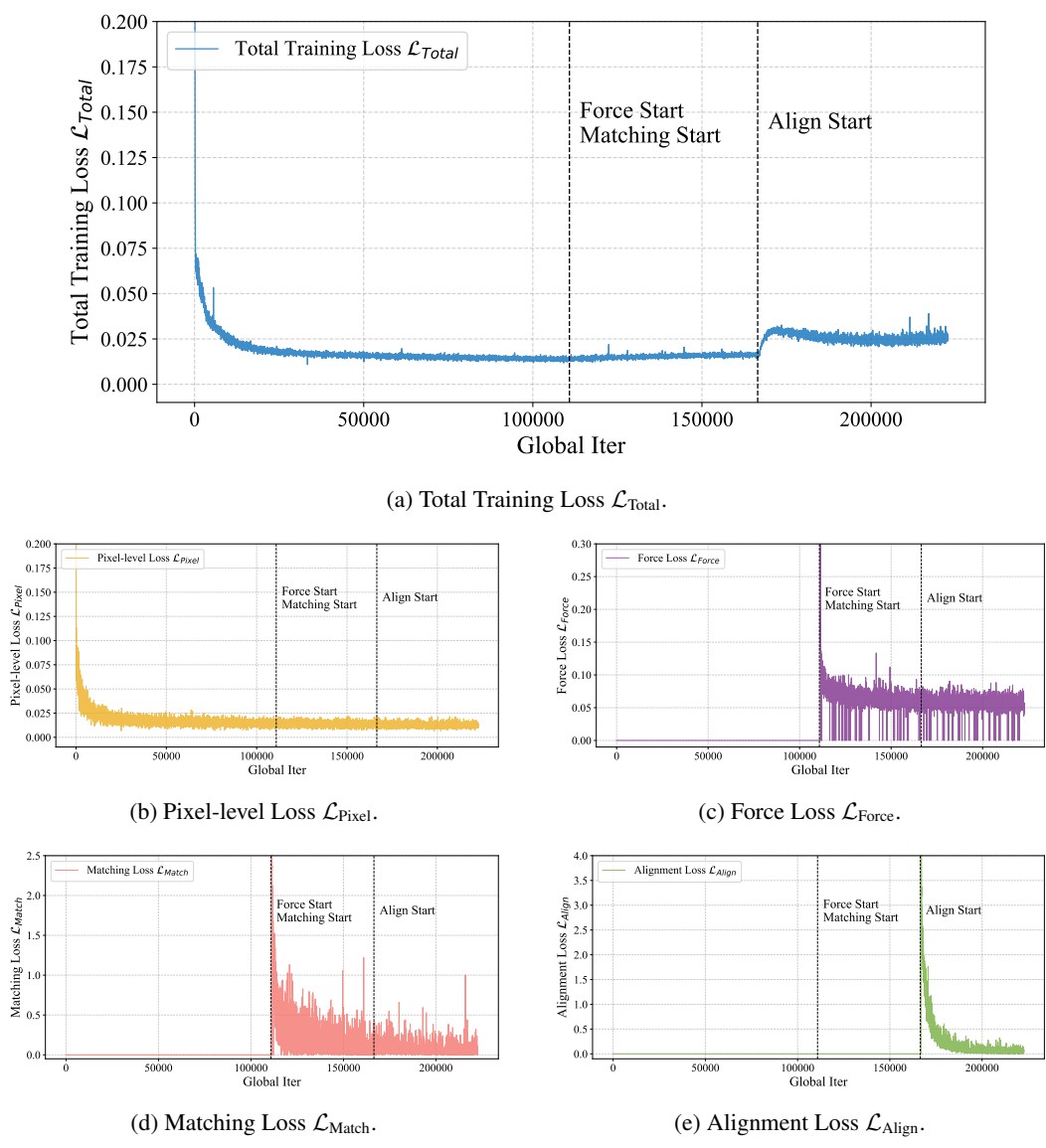

(a) Total Training Loss $\mathcal{L}_{\text{Total}}$.

(b) Pixel-level Loss $\mathcal{L}_{\text{Pixel}}$.

(c) Force Loss $\mathcal{L}_{\text{Force}}$.

(d) Matching Loss $\mathcal{L}_{\text{Match}}$.

(e) Alignment Loss $\mathcal{L}_{\text{Align}}$.

Figure 17: Training loss curves for each task and for the overall objective.

Table 15: Cross-sensor generalization results on the USB Insertion task.

| Method | Training Sensor | Tier 2 USB Insertion | |
|---|---|---|---|
| | | DG | Mini |
| T3 | DG | 0.15 | 0.05 |
| **AnyTouch 2** | | **0.35** | **0.15** |
| T3 | Mini | 0.05 | 0.10 |
| **AnyTouch 2** | | **0.15** | **0.30** |

This setup is similar to CTTP (Rodriguez et al., 2025). However, since CTTP is trained using spatially aligned samples ( collected at exactly the same contact location) from different sensors, while our model uses only coarsely aligned samples (collected at nearby contact locations) and additionally incorporates other training data sources, the training data used by the two approaches are not

Table 16: Generalization results on gel pad changes on GelSight Mini sensor.

| Method | Training Gel Pad | Tier 2 USB Insertion | | Tier 1 Chip Moving | |
|---|---|---|---|---|---|
| | | Gel Pad 1 | Gel Pad 2 | Gel Pad 1 | Gel Pad 2 |
| **AnyTouch 2** | **Gel Pad 1** | 0.30 | 0.30 | 0.80 | 0.75 |

comparable. As a result, it is difficult to conduct a fair comparison, and therefore we do not include a comparison with this method. The results shown in Tab. 15 indicate that when changing the sensor during the test time, our AnyTouch 2 model still outperforms the T3 baseline. This indicates that our model possesses stronger sensor-invariant capabilities than T3. These capabilities primarily benefit from three factors: the integration of large-scale multi-sensor data during training, the unified background removal applied during preprocessing, and the explicit aggregation of representations from different sensors enabled by the multi-modal alignment and cross-sensor matching modules in our model (which follow the same core principles as Contrastive Touch-to-Touch Pretraining). However, we also acknowledge that relying solely on coarsely aligned samples for cross-sensor matching makes it difficult to maintain performance when switching sensors, especially compared with cross-sensor contrastive learning using spatially aligned samples.

To further evaluate our model's robustness to changes in the gel pads of the sensors, we also tested our model on the USB Insertion and Chip Moving tasks using GelSight Mini Sensor. After completing data collection and model training, we replaced the GelSight Mini's gel pad only at test time. The experimental results are shown in Tab. 16. We observe that replacing the gel pad causes only a minor performance drop. This further demonstrates the sensor-invariant capability of our model.

## A.12 LIMITATIONS

Our work still has several limitations that open avenues for future exploration:

- **Unexplored sensors within the ToucHD dataset.** During data collection, we included two marker-based optical tactile sensors, DM-Tac W and GelStereo BioTip. Notably, GelStereo BioTip is also a spherical sensor, which differs significantly in structure from other planar tactile sensors. However, these additional data remain unexplored and are not utilized in the present study. Incorporating them in future work may further enrich the diversity of tactile representations.

- **The force data collection setup can be further improved**. paired tactile–force data can only be collected by moving a specially designed indenter across the sensor surface, which restricts interactions to simplified conditions and excludes a broad range of everyday objects. A more advanced setup capable of capturing tactile–force data during natural object manipulations would significantly enhance the diversity of the dataset.

- **Underutilization of multi-sensor paired manipulation data.**. Although we collect multi-sensor paired data through specifically designed UMI, in this work they are only fed into the model and aligned with the corresponding visual modality, without introducing specialized architectures to exploit cross-sensor synergies. Beyond serving as a complementary cue, the visual modality also holds potential as a predictor of future tactile signals, which remains an underexplored direction.

- **This work is still limited to general optical tactile sensors.** Although achieving general optical tactile representations is already a challenging and significant step, array-based tactile sensors are also very common in robotics. Our framework does not yet integrate such array-based modalities, and extending the model to handle heterogeneous tactile data formats will be an important direction for future research.

## A.13 LLM USAGE

We employed a large language model (LLM) solely for linguistic refinement of this manuscript, such as grammar correction, phrasing improvement, and style polishing. The LLM was not involved in research design, data collection, model development, experiments, or analysis. All scientific contributions, results, and conclusions are entirely the work of the authors.

