# OpenReview forum: "AnyTouch 2: General Optical Tactile Representation Learning For Dynamic Tactile Perception"
_ICLR.cc/2026/Conference — ICLR 2026 Poster_

### Official Review · Reviewer_shzk · 2025-11-01

**Soundness:** 3
**Presentation:** 3
**Contribution:** 3
**Rating:** 8
**Confidence:** 4

**Summary:**

This paper proposed AnyTouch 2 as a general tactile representation learning framework.
Based on their massive hierarchical dataset ToucHD, AnyTouch 2 tried to unify static/dynamic pixel-level information, multi-modal alignment, and dynamic (delta) force information into unified representation, extending the previous AnyTouch 1 architecture to richer tactile information at different levels.
Experiments on perception and manipulation tasks demonstrated the effectiveness of AnyTouch 2.

**Strengths:**

1. The paper is well-motivated and easy to follow. The contributions are sound and will benefit the tactile community.
2. Categorizing tactile tasks into 5 levels provides a good insight into the community. Existing research mostly do not distinguish some of the levels where the tactile modality is actually playing different roles. This insight extends the design of Sparsh with convincing discussion and experiments.
3. Comprehensive experiment and ablation with rich explanation and analysis.

**Weaknesses:**

Simulating those sensors, especially those with markers, is non-trivial. It will be good to further explain the IMPM pipeline for dynamic scenes, as well as how those reconstructed meshes from IMPM is then rendered in Blender for your simulation.

**Questions:**

1. In the multi-modal alignment ablation, do you think these two properties contradicts each other, or is it due to limitations of the text encoder or AnyTouch 2 training scheme?
2. For the force data, please show some visualization of those non-planar sensors' reading. Currently I have no intuition if those sensor data will really help if they are never seen in other levels. An ablation is also appreciated if possible.

---

> ### Author Response · Authors · 2025-11-23
>
> We would like to express our sincere appreciation for your constructive suggestions. We sincerely appreciate your comments that the paper is well-motivated and easy to follow, the contributions are sound and will benefit the tactile community, and the experiments, ablations and analysis are comprehensive.
>
> > Simulating those sensors, especially those with markers, is non-trivial. It will be good to further explain the IMPM pipeline for dynamic scenes, as well as how those reconstructed meshes from IMPM is then rendered in Blender for your simulation.
>
> Thank you for your constructive comments.  In this work, we employ the IMPM (Improved Material Point Method) [1] optical tactile simulation platform to simulate the tactile images for different optical tactile sensors. IMPM extends MPM to more accurately simulate complex manipulations. Each iteration alternates particle-to-grid and grid-to-particle transfers: particle mass, momentum, and deformation information are interpolated to grid nodes, grid velocities are computed with boundary nodes clamped, and updated velocities are interpolated back to update particle velocities, affine fields, and deformation gradients. A key component is the relative-rest check, which stabilizes slip and rotation. During commanded tangential motion, object-particle velocities follow the target motion, while the z-velocities of lower elastomer particles are fixed. IMPM repeats the transfer cycle until the elastomer particle nearest the contact point matches the object’s velocity within a threshold, after which particle positions are updated. The full simulation pipeline consists of two main components: elastomer–object contact simulation and rendering, which are described below.
>
> **Preparation for simulation**
>
> 1. We convert the point cloud models from multiple datasets into a standardized NumPy format, which records the 3D positional information of each particle in the model.
>
> **Elastomer–object contact simulation**
>
> 1. We initialize the grids and particles based on the object’s initial position. Specifically,  the 3D grid dimensions are manually specified, including the number of nodes, their velocities, masses, and the grid size. Particle initial parameters are also defined,  which consist of particle number, position $x\in\mathbb{R}^3$, velocity $v\in\mathbb{R}^3$, mass $m\in\mathbb{R}^+$, affine velocity field $C\in\mathbb{R}^3$, deformation gradient $F\in\mathbb{R}^3$, density $\rho\in\mathbb{R}^+$, Young’s modulus $E\in\mathbb{R}^+$ and Poisson’s ratio $\nu\in\mathbb{R}$. From these, particle volumes and Lamé parameters are computed. The system includes both indenter and elastomer particles. To avoid simulation failure due to indenter particles leaving the domain, any particle positioned outside the grid boundary is removed.
> 2. To reduce the movement time, the indenter is placed so that its center aligns with the elastomer’s center, and its bottom surface is tangent to the top surface of the elastomer.
> 3. The indenter is driven downward using IMPM until the elastomer reaches the target deformation depth. During this process, the simulation continues to advance the indenter step by step until the specified deformation threshold is met.
> 4. Indenter motions in the xy-plane—left/right translations and clockwise/counter-clockwise rotations—are simulated step by step using IMPM until the target pose is reached. At each step, the 3D positions of elastomer particles are recorded, generating a sequence of elastomer deformation states. Motion parameters such as translation distance, rotation angle, and sampling frequency are configurable. In this study, each simulated interaction produces 30 frames capturing the elastomer deformation throughout the motion.
>
> **Rendering**
>
> 1. The reconstructed triangle meshes are imported into Blender.
> 2. For each sensor, the corresponding undeformed background image is projected onto the mesh surface. **As surface geometry deforms during contact, the marker patterns deform accordingly, eliminating the need for manual annotation.**
> 3. LED lighting effects are incorporated according to the sensor design—including LED positions, colors, and power settings—and the corresponding rendered images are generated.
>
> We have further clarified the description of the simulated data collection process in Appendix A.2.1 in the revised version. Thank you again for your valuable comments!
>
> [1] Shen, Z., Sun, Y., Zhang, S., Chen, Z., Sun, H., Sun, F., & Fang, B. (2024). Simulation of optical tactile sensors supporting slip and rotation using path tracing and IMPM. *IEEE Robotics and Automation Letters*.

---

> > ### Author Response · Authors · 2025-11-23
> >
> > > In the multi-modal alignment ablation, do you think these two properties contradicts each other, or is it due to limitations of the text encoder or AnyTouch 2 training scheme?
> >
> > Thank you for your valuable question! In our ablation study, we found that removing the multi-modal alignment module leads to a significant performance drop on the two material understanding datasets in Object Bench, but it improves performance on most of the dynamic physical perception datasets. **This is due to the substantial difference in label granularity between the two types of tasks**. The text labels currently used for multi-modal alignment contain only coarse-grained object attributes, such as general shape, material, hardness, and roughness, but they do not include fine-grained physical quantities related to contact, such as contact force or pressing speed. As a result, an obvious consequence arises: during multi-modal alignment, samples of the same object pressed with different forces are pulled closer together. This is undesirable for downstream tasks that require distinguishing between different levels of contact force.
> >
> > This issue is actually common in CLIP-style vision–language alignment paradigms [2,3,4]. As the text labels are coarse-grained, multi-modal alignment can lead to suboptimal fine-grained visual perception. Correspondingly, their solution is actually straightforward: for example, expanding short, coarse-grained text annotations into longer, fine-grained descriptions. Similarly, the phenomenon observed in AnyTouch 2 could be mitigated by injecting fine-grained information—such as contact physical quantities—into the text annotations. However, previous tactile datasets have rarely addressed this important detail, making it impossible to construct such fine-grained textual labels from existing large-scale material datasets. **This further highlights the significant guiding value of our proposed tactile dynamic pyramid for tactile data collection and model design**. We have expanded this part of the discussion in Appendix A.9 in the revised version. Thank you again for your highly valuable question!
> >
> > [2] Maninis, K. K., Chen, K., Ghosh, S., Karpur, A., Chen, K., Xia, Y., ... & Araujo, A. (2024). TIPS: Text-image pretraining with spatial awareness. *arXiv preprint arXiv:2410.16512*.
> >
> > [3] Jing, D., He, X., Luo, Y., Fei, N., Wei, W., Zhao, H., & Lu, Z. (2024). Fineclip: Self-distilled region-based clip for better fine-grained understanding. *Advances in Neural Information Processing Systems*, *37*, 27896-27918.
> >
> > [4] Xie, C., Wang, B., Kong, F., Li, J., Liang, D., Zhang, G., ... & Yin, Y. FG-CLIP: Fine-Grained Visual and Textual Alignment. In *Forty-second International Conference on Machine Learning*.

---

> > > ### Author Response · Authors · 2025-11-23
> > >
> > > > For the force data, please show some visualization of those non-planar sensors' reading. Currently I have no intuition if those sensor data will really help if they are never seen in other levels. An ablation is also appreciated if possible.
> > >
> > > Thank you for pointing this out. When collecting the ToucHD (Force) subset, we used the GelStereo BioTip, a spherical optical sensor. Nearly 300 markers are distributed across the top surface of the sensor, and their 3D positions are captured by the stereo cameras inside the sensor. To more clearly illustrate the data format and characteristics of this sensor, we have added visualizations of the raw data and the 3D markers obtained as the indenter slides over the GelStereo BioTip sensor surface during force collection, as shown in Figure 8 and 10.
> > >
> > > We collect data from this sensor because, although the sensors integrated into AnyTouch 2 are mainly planar optical tactile sensors, many non-planar and even non-optical tactile sensors are still widely used in practice. While their surface geometry (non-planar) and data representation (3D markers) differ from planar optical tactile sensors, these tactile sensors share the same fundamental principles of converting tactile signals into visual information (either 2D or 3D), indicating clear potential for further integration. **Thus, together with other sensors in ToucHD (Force), the purpose of this part of data is to serve as a bridge between planar optical sensors and non-planar or non-optical tactile sensors, enabling future integration of a broader range of tactile sensors.** Even without including this sensor in other tiers, the aligned data collection process and the unified 3D contact force labels shared across sensors in this subset already provide a large amount of multi-sensor supervision.  This allows the model to transfer dynamic tactile perception knowledge learned from other sensors to this sensor. However, since we currently lack access to additional non-planar and non-optical tactile sensors, it is difficult to directly demonstrate this benefit. Therefore, **exploring how to integrate data from 3D marker–based sensors with other optical tactile sensors based on this part of data, will be an important direction for our future work**. We have expanded this part of the discussion in Appendix A.2.3 in the revised version. Thank you again for pointing this out!

---

### Official Review · Reviewer_LnK5 · 2025-11-01

**Soundness:** 3
**Presentation:** 2
**Contribution:** 2
**Rating:** 2
**Confidence:** 5

**Summary:**

The paper present a tactile dataset and try to build hierarchical perception capabilities. The paper also present a tactile representation learning framework for optical tactile sensors, which fail to show the capabilities of tactile in manipulation.

The paper will be stronger if it focus more on the policy training and representation learning, instead of focusing too much on building such hierarchy pyramid.

The paper try to convey too many contents but loss effective evidence and support, making the paper loss focus.

Incomplete in presentation. Some details on the policy training are missing.

**Strengths:**

Good visualization. The research on related works are comprehensive.

**Weaknesses:**

Minor weakness:


Major weakness:

1. The proposed “tactile dynamic pyramid” is ambitious, but its practical utility is unclear. Tier boundaries are not operationally defined, and the paper does not show that organizing data by tiers yields better representations than training a single, unified model on pooled data with task-specific supervision. Please provide measurable criteria for each tier, evidence of annotation consistency, and ablations comparing (a) pyramid-driven training vs. (b) a unified representation or a task curriculum without tiers.

2. Tier-1 is defined as “supporting precise force-sensitive manipulation,” yet much of this capability can be achieved with press-only data plus straightforward calibration[1], even with non-optical sensors (e.g., force-sensing resistor) that directly measure normal force. The manuscript also appears to conflate normal and shear forces. Please (i) specify which force components are observed/estimated (normal, shear, slip), (ii) describe the labels/ground truth and metrics used to distinguish them, and (iii) demonstrate that Tier-1 data provides incremental benefit over Tier-5 press-only data for force estimation and manipulation success. Force calibration is already contained in the gelsight paper[1].

GelSight: High-Resolution Robot Tactile Sensors for Estimating Geometry and Force

3. Key physical quantities (normal vs shear force, torque, friction coefficient, compliance) are mixed across tiers; some are learnable from press-only with calibrated loads, others require tangential motion. The pyramid does not map cleanly onto physics.

4. With straightforward calibration (even non-optical FSR/arrays), normal force can be estimated robustly. Claiming T1 superiority without showing incremental benefit over calibrated T5 is unconvincing.

5. What is the boundary between force data, manipulation data, and specific action? How do you support your claim that the manipulation dynamics support various complex real-world manipulation tasks, instead of a incremental benefit from the "press-only"?

6. For Tire2, have you contact experiments to support you have extract manipulation dynamics from tactile information? It will be more convincing if your train a dynamics model on the manipulation data and try to prove if your dynamics model could build some casual relationships between tactile and manipulation dynamics of physics. Other additional experiments also work.

7. In figure 3, please specify the difference for "tactile grasping" and "chip moving". It seems both two tasks all try to grasp fragile objects, its just the chip is even more fragile than egg, which require more sensitive tactile instead of require a different sensing modality.

8. Line 306-307, "Among these properties, contact force is fundamental, as it directly governs how objects deform, slip, or respond during manipulation". Follow the authors claim, if contact force is such fundamental, why not directly use 6-axis force sensor on wrist or directly on the gripper. Then you can directly achieve the same capabilities with 6-axis force only dataset.

9. Most of tasks in real-world experiments do not reflect the physics of "slip", which claimed in the previous section. All those for tasks seems highlight the binary contact or not(which can be achieved with simple FSR sensor), and force direction(which can be achieved by the 6-axis force sensor)

10. The major concern would be the overclaim of this paper.

11. Line 481-483 "From the data perspective, the proposed ToucHD dataset serves as the final missing piece, completing a comprehensive dynamic tactile data ecosystem that supports multiple tiers of perception." The author try to build the ecosystem on the tiers which without rigor definition of boundary. Its actually mixing the concept of different types of tactile modalities instead of "serving as the final missing piece".

12. No limitation section in main text.

13. Line 483-485, "our AnyTouch 2 general representation learning framework integrates multi-level objectives across all tiers,  endowing it with comprehensive dy- namic tactile perception capabilities.", the representation learning framework do not support the capabilities of tactile in manipulation. More complex task and solid experiments are required.

14. In caption of Figure 4, "Each dynamic model has a corresponding  dynamic tier, which denotes the highest level of the training data and objectives used in the tactile dynamic pyramid, reflecting the model’s dy- namic perception capability." Can you specify what is "dynamic model", "dynamic tier", "dynamic pyramid" meaning in this context?

15. For four real-world tasks, does the model directly evaluate on those tasks? If the model fine-tuned on downstream task specific data. please provide details on how you fine-tune the model on downstream tasks.

**Questions:**

see weakness above

---

> ### Author Response · Authors · 2025-11-23
>
> We would like to express our sincere appreciation for your constructive suggestions. We sincerely appreciate your comments that the visualization is good and the research on related works is comprehensive.
>
>
>
> > The proposed “tactile dynamic pyramid” is ambitious, but its practical utility is unclear. Tier boundaries are not operationally defined. Please provide measurable criteria for each tier, evidence of annotation consistency
>
> Thank you for your valuable comments. First, we would like to emphasize that **the primary role of our proposed Tactile Dynamics Pyramid is to offer new perspectives and considerations for several existing paradigms in the tactile community, including data collection strategies, model design principles, and the categorization of manipulation tasks**. In this work, our ToucHD dataset was motivated directly by the observation that high-tier dynamic tactile data of the pyramid are markedly scarce. Similarly, the design of AnyTouch 2 explicitly incorporates modules tailored to the dynamic perception capabilities required at different tiers of the pyramid (e.g., force prediction for Tier 1 and action matching for Tier 3). Our selection of downstream manipulation tasks also follows the structure of the pyramid, aiming to comprehensively cover all tiers. This pyramid framework has also been recognized by other reviewers (Review tU7z and shzk).
>
> Second, we want to further clarify the **measurable criteria** of the tiered structure of our Tactile Dynamics Pyramid. These tiers are defined based on the data collection efforts, the types of actions, and the difficulty of obtaining labels:
>
> - **Tier 5 (Press-Only)**: This tier of data is collected by **only pressing the sensor against objects** using either handheld operation or a robot arm. **No detailed action-type annotations or paired force labels are provided.**
> - **Tier 4 (Random Action)**: This tier of data is collected by **pressing the sensor against objects, followed by random sliding and rotation** using either handheld operation or a robot arm. **No detailed action-type annotations or paired force labels are provided.**
> - **Tier 3 (Specific Action)**: This tier of data is collected by **programmatically controlling the sensor to press and slide** along the object surface **following specific predefined actions**. **Detailed action-type labels are available, but no paired force data is provided.**
> - **Tier 2 (Manipulation Data)**: This tier of data is **collected during real object manipulation tasks** using a robot arm or a UMI device. **No paired force data is provided.**
> - **Tier 1 (Force Data)**: This tier of data is **collected by a robot arm equipped with a force sensor**, with either an indenter or an object interacting with the tactile sensor. **This is the only tier that contains paired force labels.**
>
> Within each tier, **the presence or absence of force labels and action-type annotations is consistent**. As the tier level increases, data collection becomes more challenging or requires stricter constraints, and the **data rarity increases**. However, higher-tier data provides richer annotations or more realistic manipulation scenarios, enabling the development of **stronger dynamic tactile perception capabilities**. We have expanded the related descriptions in Section 3 and Appendix A.1 in the revised version. Thank you again for your valuable comments!

---

> > ### Author Response · Authors · 2025-11-23
> >
> > > the paper does not show that organizing data by tiers yields better representations than training a single, unified model on pooled data with task-specific supervision. Please compare (a) pyramid-driven training vs. (b) a unified representation or a task curriculum without tiers.
> >
> > Thank you for your constructive comments. In training AnyTouch 2, two key components are directly guided by the Tactile Dynamic Pyramid: (1) We deliberately select training data that span all tiers of the pyramid. (2) Our task scheduling strategy coordinates the learning of the multi-modal alignment (mainly Tier 4+5 data), action matching (mainly Tier 3 data), and force prediction modules (mainly Tier 1 data). This design enables more efficient integration of these objectives while mitigating task interference. Therefore, to compare pyramid-driven training against training without tiers and thereby demonstrate the value of the Tactile Dynamic Pyramid as you suggested, we conducted evaluations on 4 different models:
> >
> > 1. AnyTouch 2 trained on a randomly sampled subset of 248k samples from the full training set. This model represents a **pyramid-driven method**, trained on a data size comparable to the other baselines for a **fair comparison**.
> > 2. AnyTouch 2 trained using only Tier 4+5 data (744k samples). It represents **the mainstream paradigm of tactile representation learning before the introduction of our Tactile Dynamic Pyramid**.
> > 3. AnyTouch 2 trained using only Tier 1 data (248k samples). It corresponds to **the unified model on pooled data with task-specific supervision** that you suggested.
> > 4. AnyTouch 2 without a task scheduling strategy (248k samples). It represents a training setup in which **the model does not follow the pyramid-guided, tier-by-tier task curriculum**. Instead, all training objectives are activated from the beginning.
> >
> > We conducted comprehensive comparisons across all offline benchmarks, and the results are presented in Table Ⅰ.
> >
> > Table Ⅰ: Comparison between AnyTouch 2 trained with our pyramid-driven strategy and non–pyramid-driven baselines.
> >
> > | **Method**                     | Training Data | **TAG (GS)** Acc(↑) | **Cloth (GS)** Acc(↑) | **Sparsh Slip (DG)** F1 Score(↑) / RMSE(↓) | **Sparsh Slip (Mini)** F1 Score(↑) / RMSE(↓) | **Sparsh Force (DG)** RMSE(↓) | **Sparsh Force (Mini)** RMSE(↓) | **ToucHD Force (DG)** RMSE(↓) | **ToucHD Force (Mini)** RMSE(↓) |
> > | ------------------------------ | ------------- | ------------------- | --------------------- | ------------------------------------------ | -------------------------------------------- | ----------------------------- | ------------------------------- | ----------------------------- | ------------------------------- |
> > | AnyTouch 2                     | 248k          | **69.97**           | **40.82**             | **85.13** / **94.34**                      | **97.80** / **106.89**                       | **679.77**                    | **232.45**                      | **960.58**                    | **1153.19**                     |
> > | AnyTouch 2 (Tier 4+5 Only)     | 744k          | 68.92               | 40.39                 | 84.16 / 110.68                             | 97.67 / 136.36                               | 783.64                        | 257.95                          | 2448.89                       | 2982.46                         |
> > | AnyTouch 2 (Tier 1 Only)       | 248k          | 61.81               | 36.62                 | 84.45 / 98.26                              | 97.60 / 115.32                               | 699.12                        | 240.26                          | 987.91                        | 1172.69                         |
> > | AnyTouch 2 w/o task scheduling | 248k          | 69.24               | 39.91                 | 76.92 / 100.34                             | 97.20 / 139.13                               | 690.39                        | 252.68                          | 1023.67                       | 1342.75                         |
> >
> > The results indicates 3 key findings: (1) The baseline trained only on Tier 4+5 data performs substantially worse than AnyTouch 2 across all tasks, highlighting the importance of high-tier data (such as our ToucHD dataset) emphasized by the tactile dynamic pyramid. (2) The baseline trained only on Tier 1 data for force prediction tasks fails to outperform AnyTouch 2 on any force-related tasks in Sparsh or ToucHD Bench. This indicates that the pyramid provides essential guidance on the comprehensive use of training data across tiers. (3) The baseline trained without our task scheduling strategy also underperforms AnyTouch 2 on all benchmarks, demonstrating the value of the pyramid in guiding the design of the model training strategy. Together, these results underscore the valuable guidance provided by our proposed tactile dynamic pyramid. We have add these results and discussions into Appendix A.9 in the revised version. Thank you again for your constructive comments!

---

> > > ### Author Response · Authors · 2025-11-23
> > >
> > > > Tier-1 is defined as “supporting precise force-sensitive manipulation,” yet much of this capability can be achieved with press-only data plus straightforward calibration[1], even with non-optical sensors (e.g., force-sensing resistor) that directly measure normal force. The manuscript also appears to conflate normal and shear forces. Please (i) specify which force components are observed/estimated (normal, shear, slip), (ii) describe the labels/ground truth and metrics used to distinguish them, and (iii) demonstrate that Tier-1 data provides incremental benefit over Tier-5 press-only data for force estimation and manipulation success. Force calibration is already contained in the gelsight paper[1].
> > >
> > > > With straightforward calibration (even non-optical FSR/arrays), normal force can be estimated robustly. Claiming T1 superiority without showing incremental benefit over calibrated T5 is unconvincing.
> > >
> > > Thank you for your valuable comments. First, we would like to emphasize that in our Tactile Dynamic Pyramid, Tier 5 Press-Only data is collected solely by pressing the sensor against objects, either by handheld operation or using a robot arm. **This means that Tier 5 data does not include any paired force labels.** Notably, many existing large-scale optical tactile datasets belong to this tier, such as TAG, ObjectFolder Real, and TVL. This underscores that contact force, as a critical factor, has been overlooked in tactile data collection. **In fact, the “press-only” data used for calibration that you mentioned, which do include force labels, correspond to Tier 1 in our pyramid. As you noted, even force labels without shear forces are crucial for supporting precise, force-sensitive manipulation. This actually highlights both the significance of our Tactile Dynamic Pyramid and the value of the force-prediction module in AnyTouch 2.** We do not further distinguish between normal and shear forces within Tier 1, as force–tactile paired data are already extremely scarce. Notably, our proposed ToucHD (Force) dataset provides both normal and shear force labels throughout the entire pressing and slipping process of the indenters. Specifically, we collected the tactile frames during the indenter’s pressing and slipping process on the sensor surface, and the corresponding 3D force labels ($F_{x}$, $F_{y}$ for shear force and $F_{z}$ for normal force) for each tactile frame. When training the AnyTouch 2 model, we train the force prediction decoder to predict the corresponding 3D contact forces $(F_{x},F_{y},F_{z})$.
> > >
> > > To further evaluate the impact of Tier 1 data and its shear force labels on downstream tasks, we conducted comparisons across all offline benchmarks and two real-world manipulation tasks. The results are presented in Table Ⅱ and Ⅲ.
> > >
> > > Table Ⅱ: Contribution of ToucHD (Force) and its shear force labels across all offline benchmarks.
> > >
> > > | **Method**                                   | **TAG (GS)** Acc(↑) | **Cloth (GS)** Acc(↑) | **Sparsh Slip (DG)** F1 Score(↑) / RMSE(↓) | **Sparsh Slip (Mini)** F1 Score(↑) / RMSE(↓) | **Sparsh Force (DG)** RMSE(↓) | **Sparsh Force (Mini)** RMSE(↓) | **ToucHD Force (DG)** RMSE(↓) | **ToucHD Force (Mini)** RMSE(↓) |
> > > | -------------------------------------------- | ------------------- | --------------------- | ------------------------------------------ | -------------------------------------------- | ----------------------------- | ------------------------------- | ----------------------------- | ------------------------------- |
> > > | AnyTouch 2                                   | **76.97**           | **42.31**             | **86.66** / **87.80**                      | **97.96** / **80.83**                        | **624.26**                    | **202.14**                      | **894.32**                    | **1051.03**                     |
> > > | AnyTouch 2 w/o Shear Force in ToucHD (Force) | 76.32               | 42.18                 | 86.08 / 97.25                              | 97.89 / 96.33                                | 675.83                        | 232.34                          | 1329.28                       | 1707.41                         |
> > > | AnyTouch 2 w/o ToucHD (Force)                | 74.33               | 40.87                 | 84.91 / 107.43                             | 97.85 / 109.37                               | 777.41                        | 266.43                          | 1792.49                       | 2424.68                         |

---

> > > > ### Author Response · Authors · 2025-11-23
> > > >
> > > > Table Ⅲ: Contribution of ToucHD (Force) and its shear force labels on real-world manipulation tasks.
> > > >
> > > > | **Method**                                   | **USB Insertion** **(DG)** | **USB Insertion** **(Mini)** | **Chip Moving** **(Mini)** |
> > > > | -------------------------------------------- | -------------------------- | ---------------------------- | -------------------------- |
> > > > | AnyTouch 2                                   | **0.35**                   | **0.30**                     | **0.80**                   |
> > > > | AnyTouch 2 w/o Shear Force in ToucHD (Force) | 0.25                       | 0.25                         | 0.75                       |
> > > > | AnyTouch 2 w/o ToucHD (Force)                | 0.25                       | 0.20                         | 0.70                       |
> > > >
> > > > The results show that Tier 1 data, along with its shear force labels, make a clear contribution to various downstream dynamic perception tasks, including force prediction and real-world manipulation tasks.  The consistent performance drop observed when removing the shear force labels further highlights the importance of collecting and predicting both shear and normal forces simultaneously.
> > > >
> > > > To more precisely quantify the contribution of the shear force labels in the ToucHD (Force) dataset to downstream 3D force prediction tasks—including both shear and normal forces, we also report the prediction errors along each force direction on Sparsh Bench and ToucHD Bench after removing either the ToucHD (Force) dataset entirely or only its shear force labels. The results are shown in Table Ⅳ and Ⅴ.
> > > >
> > > > Table Ⅳ: Contribution of ToucHD (Force) and its shear force labels on the shear force and normal force prediction tasks on Sparsh Bench.
> > > >
> > > > | **Method**                                   | **Sparsh Force (DG)** X-axis (Shear) RMSE(↓) | **Sparsh Force (DG)** Y-axis (Shear) RMSE(↓) | **Sparsh Force (DG)** Z-axis (Normal) RMSE(↓) | **Sparsh Force (Mini)** X-axis (Shear) RMSE(↓) | **Sparsh Force (Mini)** Y-axis (Shear) RMSE(↓) | **Sparsh Force (Mini)** Z-axis (Normal) RMSE(↓) |
> > > > | -------------------------------------------- | -------------------------------------------- | -------------------------------------------- | --------------------------------------------- | ---------------------------------------------- | ---------------------------------------------- | ----------------------------------------------- |
> > > > | AnyTouch 2                                   | **268.03**                                   | **161.57**                                   | **194.66**                                    | **42.29**                                      | **57.31**                                      | **102.54**                                      |
> > > > | AnyTouch 2 w/o Shear Force in ToucHD (Force) | 302.58                                       | 176.60                                       | 196.65                                        | 58.18                                          | 67.73                                          | 106.43                                          |
> > > > | AnyTouch 2 w/o ToucHD (Force)                | 308.18                                       | 215.56                                       | 253.67                                        | 76.52                                          | 78.59                                          | 111.32                                          |

---

> > > > > ### Author Response · Authors · 2025-11-23
> > > > >
> > > > > Table Ⅴ: Contribution of ToucHD (Force) and its shear force labels on the shear force and normal force prediction tasks on ToucHD Bench.
> > > > >
> > > > > | **Method**                                   | **ToucHD Force (DG)** X-axis (Shear) RMSE(↓) | **ToucHD Force (DG)** Y-axis (Shear) RMSE(↓) | **ToucHD Force (DG)** Z-axis (Normal) RMSE(↓) | **ToucHD Force (Mini)** X-axis (Shear) RMSE(↓) | **ToucHD Force (Mini)** Y-axis (Shear) RMSE(↓) | **ToucHD Force (Mini)** Z-axis (Normal) RMSE(↓) |
> > > > > | -------------------------------------------- | -------------------------------------------- | -------------------------------------------- | --------------------------------------------- | ---------------------------------------------- | ---------------------------------------------- | ----------------------------------------------- |
> > > > > | AnyTouch 2                                   | **285.89**                                   | **334.37**                                   | **274.06**                                    | **273.48**                                     | **397.17**                                     | **380.38**                                      |
> > > > > | AnyTouch 2 w/o Shear Force in ToucHD (Force) | 433.24                                       | 594.77                                       | 301.26                                        | 443.54                                         | 820.00                                         | 443.87                                          |
> > > > > | AnyTouch 2 w/o ToucHD (Force)                | 530.30                                       | 708.32                                       | 553.86                                        | 730.16                                         | 1012.12                                        | 682.40                                          |
> > > > >
> > > > > The results show that removing the shear force labels has a significant impact on the prediction performance along the x and y axes (shear forces) in downstream tasks. Training the model using only z-axis (normal force) labels is clearly insufficient to support accurate shear force prediction. This highlights the advantage of our ToucHD dataset in providing both normal and shear force labels.
> > > > >
> > > > > We also want to clarify the connection and distinction between our work and the force calibration in the GelSight paper[1]. We both acknowledge that understanding shear and normal forces is crucial for tactile perception, and we both use neural networks to predict 3D contact forces from tactile frame inputs. The key difference is that the force calibration in the GelSight papers is primarily designed for force prediction evaluation, similar to your mentioned "unified model on pooled data with task-specific supervision," and predicts contact forces from single tactile frames. In contrast, our AnyTouch 2 integrates force prediction and delta-force prediction into the pre-training process, enabling it to predict dynamic contact forces as well as force changes over a short time window from a sequence of four consecutive tactile frames. Moreover, we experimentally demonstrate that these force-related pre-training tasks also benefit downstream real-world manipulation tasks, as shown in Table Ⅲ.
> > > > >
> > > > > We have incorporated these results and discussions into Appendix A.2.3 and A.9 in the revised version. Thank you again for your constructive comments!
> > > > >
> > > > >
> > > > >
> > > > > [1] Yuan, W., Dong, S., & Adelson, E. H. (2017). Gelsight: High-resolution robot tactile sensors for estimating geometry and force. *Sensors*, *17*(12), 2762.

---

> > > > > > ### Author Response · Authors · 2025-11-23
> > > > > >
> > > > > > > Key physical quantities (normal vs shear force, torque, friction coefficient, compliance) are mixed across tiers; some are learnable from press-only with calibrated loads, others require tangential motion. The pyramid does not map cleanly onto physics.
> > > > > >
> > > > > > Thank you for your comments. We would like to clarify that the levels in our proposed tactile dynamics pyramid are defined based on the **data collection efforts**, the **types of actions**, and the **difficulty of obtaining labels**. We do not require each tier to correspond to variations in specific key physical quantities, as **these quantities are actually included in the types of actions**. Relying solely on key physical quantities as the basis for tier division would therefore be neither appropriate nor effective. For example, consider a sequence in which an object randomly slides across the sensor surface. Having paired force labels (Tier 1 data) versus having no paired force supervision (Tier 4 data) results in fundamentally different values, **even though the key physical quantities are exactly the same**. This also highlights the new perspectives and considerations that our proposed tactile dynamics pyramid brings to the tactile community, which is also recognized by another reviewer(Reviewer shzk). We have expanded the discussions in Appendix A.1 in the revised version. Thank you again for your comments!
> > > > > >
> > > > > >
> > > > > >
> > > > > > > For Tire2, have you contact experiments to support you have extract manipulation dynamics from tactile information? It will be more convincing if your train a dynamics model on the manipulation data and try to prove if your dynamics model could build some casual relationships between tactile and manipulation dynamics of physics. Other additional experiments also work.
> > > > > >
> > > > > > Thank you for your suggestions. The only Tier 2 dataset we use during training is **ToucHD (Mani)**, which is collected from 46 real-world manipulation tasks and contains a large number of tactile dynamic deformation patterns commonly observed during real manipulation. By training on this dataset, AnyTouch 2 could learn to capture the fine-grained temporal deformation pattern that naturally arises when objects interact, slip, or shift during manipulation. To quantify the contribution of this Tier 2 data to manipulation performance, we conducted ablation studies on two complex real-world manipulation tasks: USB Insertion and Chip Moving. Both tasks involve a variety of dynamic deformation patterns drawn from the 46 tasks in ToucHD (Mani), including pressing, sliding, interactions with fragile objects, and overcoming stiction. The results are presented in Table Ⅵ.
> > > > > >
> > > > > > Table Ⅵ: Contribution of the Tier 2 ToucHD (Mini) data on real-world manipulation tasks.
> > > > > >
> > > > > > | **Method**                   | **USB Insertion** **(DG)** | **USB Insertion** **(Mini)** | **Chip Moving** **(Mini)** |
> > > > > > | ---------------------------- | -------------------------- | ---------------------------- | -------------------------- |
> > > > > > | AnyTouch 2                   | **0.35**                   | **0.30**                     | **0.80**                   |
> > > > > > | AnyTouch 2 w/o ToucHD (Mani) | 0.25                       | 0.25                         | 0.70                       |
> > > > > >
> > > > > > The results show that removing the only Tier 2 dataset from the training data leads to a performance drop in both manipulation tasks. This demonstrates that incorporating Tier 2 training data indeed enables the model to better perceive the dynamic tactile characteristics that arise during real-world manipulation. We have incorporated these results and discussions into Appendix A.9 in the revised version. Thank you again for your constructive comments!

---

> > > > > > > ### Author Response · Authors · 2025-11-23
> > > > > > >
> > > > > > > > Most of tasks in real-world experiments do not reflect the physics of "slip", which claimed in the previous section. All those for tasks seems highlight the binary contact or not(which can be achieved with simple FSR sensor), and force direction(which can be achieved by the 6-axis force sensor)
> > > > > > >
> > > > > > > Thank you for your comments. First, we would like to clarify that among the four real-world manipulation tasks we selected, **three of them inherently involve slip dynamics, including Whiteboard Wiping, USB Insertion, and Chip Moving**. In the Whiteboard Wiping task, the robot must use an eraser to wipe letters off a whiteboard until the surface is completely clean. The interaction involves structured sliding motions—primarily upward sliding of the eraser across the sensor surface. The robot must accurately perceive these sliding cues to dynamically adjust its wiping motion throughout execution. In the USB Insertion task, micro slips frequently occur when the connector aligns, adjusts its orientation, or lightly contacts the port edges. These subtle slip signals are crucial for guiding fine-grained corrections during insertion and ensuring stable alignment. In the Chip Moving task,  subtle displacements between the chip and the sensor occured when the chip touches the bottle during placement. These displacements are critical: if the model fails to detect the chip’s movement with sufficient sensitivity, further lowering may cause the chip to break upon collision. Simply perceiving normal force or the direction of contact force is not sufficient to accomplish these manipulation tasks.
> > > > > > >
> > > > > > > Second, FSR sensors and 6-axis force sensors cannot replace optical tactile sensors, because **our goal is not only to perceive contact forces but also to capture material properties, textures, and fine-grained geometric details to support a broader range of tactile perception tasks** (e.g., Object Bench). Optical tactile sensors have the potential to handle all these tasks, as their data contains far richer tactile cues than force signals alone. However, existing models are still limited in their ability to fully exploit these rich dynamic features, especially in capturing contact forces and subtle deformations. **This is precisely where the value and significance of our work lie**. We have expanded the corresponding discussions in Appendix A.6 in the revised version. Thank you again for your comments!
> > > > > > >
> > > > > > >
> > > > > > >
> > > > > > > > Line 306-307, "Among these properties, contact force is fundamental, as it directly governs how objects deform, slip, or respond during manipulation". Follow the authors claim, if contact force is such fundamental, why not directly use 6-axis force sensor on wrist or directly on the gripper. Then you can directly achieve the same capabilities with 6-axis force only dataset.
> > > > > > >
> > > > > > > Thank you for your comments. We would like to clarify that **obtaining contact forces is not the only purpose of using optical tactile sensors**. The local texture, deformation details, and other fine-grained information provided by these sensors are equally crucial. Specifically, optical tactile sensors offer several irreplaceable advantages:
> > > > > > >
> > > > > > > - **Optical tactile sensors provide high-resolution, local tactile information** (e.g., shear and normal forces at multiple points), which is essential for tasks requiring fine manipulation, material or texture recognition, slippage, or small object features. The 6-axis wrist or gripper force sensors can only measure the force and torque applied to the entire gripper or wrist.
> > > > > > > - **Optical tactile sensors can capture subtle object interactions such as slips or rotational movements of the object relative to the gripper.** A wrist-mounted 6-axis force sensor cannot resolve these fine-grained dynamic events because it integrates all contact forces over the entire gripper.
> > > > > > > - **Optical tactile sensors can support multi-fingered dexterous hands and record detailed tactile information for each finger.** In contrast, using multiple 6-axis sensors at every contact point becomes impractical.
> > > > > > >
> > > > > > > **In summary, optical tactile sensors and 6-axis force sensors are complementary.** While 6-axis force sensors can directly measure forces, their size and configuration limit their ability to accurately capture local contact properties, such as the normal and shear force on the contact surface. Additionally, they cannot provide fine-grained information about surface material and texture, the capability that has also been emphasized by another reviewer (Reviewer nyMK). These limitations of 6-axis force sensors motivate our exploration of optical tactile sensors and their representations. Hence, **extracting contact forces and other unique physical and dynamic information from data collected by optical tactile sensors is undoubtedly a highly valuable research direction.** Thank you again for your comments.

---

> > > > > > > > ### Author Response · Authors · 2025-11-23
> > > > > > > >
> > > > > > > > > In figure 3, please specify the difference for "tactile grasping" and "chip moving". It seems both two tasks all try to grasp fragile objects, its just the chip is more fragile than egg, which require more sensitive tactile instead of require a different sensing modality.
> > > > > > > >
> > > > > > > > Thank you for pointing this out. We would like to clarify that the Tier 5 Tactile Grasping task and the Tier 1 Chip Moving task **differ substantially in the objects being manipulated, the difficulty of the manipulation, and the dynamic characteristics of the tasks**. **In the Tactile Grasping task, the robot grasps white rigid spheres with different surface textures.** The model must use the tactile feedback obtained during grasping to identify the texture type and place the sphere into the corresponding box. This task primarily involves press-only tactile interaction, and follows a classic pick-and-place pipeline. During arm motion, the sphere rarely slips relative to the tactile sensor, making this task the least dynamic among the four tasks.  The main challenge lies in enabling the model to correctly infer the sphere’s material and texture across a wide range of potential grasping locations. **For the Chip Moving task, the robot arm must pick up a very fragile chip from the top of one bottle and place it steadily onto another bottle, all while operating under partial visual occlusion.** This task involves not only pressing actions **but also small displacements between the chip and the sensor** when the chip contacts the bottle during placement. Although these displacements are subtle, they are critical to task success: if the model does not capture the chip’s movement with sufficient sensitivity, continuing to lower the chip may cause it to break upon collision with the bottle. Therefore, during both the picking up and placing phases, the model must capture the subtle dynamic deformations of the chip on the sensor surface in real time and respond with rapid actions. This poses a significant challenge to the model’s dynamic tactile perception capabilities, making it far more difficult than texture recognition in the Tactile Grasping task. The key to successfully completing this task lies in our proposed force and delta-force prediction modules, which enhance the model’s understanding of contact force variations corresponding to subtle object deformations. As shown in Table 9 in Appendix A.9, removing the ToucHD (Force) dataset and the corresponding force prediction tasks leads to a noticeable drop in performance on the Tier 1 Chip Moving task, which strongly supports this conclusion. This also highlights the value of our proposed Tactile Dynamic Pyramid in guiding task grouping, data collection, and model design. We have expanded the corresponding discussions in Appendix A.6 in the revised version. Thank you again for pointing this out!
> > > > > > > >
> > > > > > > >
> > > > > > > >
> > > > > > > > > The major concern would be the overclaim of this paper.
> > > > > > > >
> > > > > > > > Thank you for your comments. We would like to clarify the primary contributions of this work:
> > > > > > > >
> > > > > > > > - We recognize that the dynamic characteristics of existing large-scale tactile datasets are simple, and systematically construct a **Tactile Dynamics Pyramid** that groups tactile datasets according to the levels of dynamic perceptual capabilities they support. This pyramid provides new perspectives and considerations for data collection, model design, and the selection of downstream tasks. **This perspective is recognized by another reviewer (Reviewer shzk) and further supported by the results presented in Table Ⅰ.**
> > > > > > > > - Guided by the Tactile Dynamics Pyramid, we identify a significant scarcity of high tier dynamic tactile data and introduce the **ToucHD** dataset to fill this critical gap, including **simulated atomic action data**, **real-world manipulation data** and **touch-force paired data**. This dataset substantially enhances the model’s dynamic perception capabilities across a wide range of downstream tasks, **supported by extensive comparisons against baselines and thorough ablation studies. Its novelty and clear benefits to the tactile community have also been recognized by other reviewers (Reviewer nyMK and shzk).**
> > > > > > > > - Building on the Tactile Dynamics Pyramid and the ToucHD dataset, we further propose **AnyTouch 2**, a general tactile representation learning framework that unifies sensor-invariant object-level understanding with multi-level dynamic perception capabilities. **The model demonstrates superior performance across a variety of downstream tactile perception tasks, including object property understanding (Object Bench), dynamic physical perception (Sparsh Bench and ToucHD Bench), and real-world manipulation tasks. The thoroughness of our technical details and experiments has also been recognized by other reviewers (Reviewer nyMK, tU7z and shzk).**
> > > > > > > >
> > > > > > > > Therefore, we consider that our work does not have an overclaim issue. It indeed provides substantial value and benefit to the tactile community. Thank you again for your comments.

---

> > > > > > > > > ### Author Response · Authors · 2025-11-23
> > > > > > > > >
> > > > > > > > > > Line 481-483 "From the data perspective, the proposed ToucHD dataset serves as the final missing piece, completing a comprehensive dynamic tactile data ecosystem that supports multiple tiers of perception." The author try to build the ecosystem on the tiers which without rigor definition of boundary. Its actually mixing the concept of different types of tactile modalities instead of "serving as the final missing piece".
> > > > > > > > >
> > > > > > > > > Thank you for your comments. We have previously clarified the boundaries of each tier in the Tactile Dynamic Pyramid. These tiers are not strictly defined by specific tactile physical quantities, but are instead based on the **data collection efforts**, the **types of actions**, and the **difficulty of obtaining labels**. The pyramid clearly highlights the current scarcity of large-scale real-world manipulation data and force-paired data, which motivated the creation of the ToucHD dataset to fill the gap in high-tier (Tier 3, 2, and 1) dynamic tactile data. The value of our ToucHD dataset to the tactile community has been recognized by other reviewers. Furthermore, guided by our Tactile Dynamic Pyramid for data integration, model design, and training strategy, the AnyTouch 2 model trained with our ToucHD dataset outperforms existing models across various downstream tasks. These results have already demonstrated the significance of the proposed Tactile Dynamic Pyramid and the ToucHD dataset. Thank you again for your comments.
> > > > > > > > >
> > > > > > > > >
> > > > > > > > >
> > > > > > > > >
> > > > > > > > >
> > > > > > > > > > No limitation section in main text.
> > > > > > > > >
> > > > > > > > > Thank you for pointing this out. **We would like to clarify that we have already discussed the limitations of our work in detail in Appendix A.12**, including the underutilization of ToucHD, potential improvements in the force data collection process, and the possibility of further extending our approach to non-optical tactile sensors. We placed this discussion in the appendix primarily due to page limitations. We have added a reference to this section in the revised version. Thank you again for pointing this out!
> > > > > > > > >
> > > > > > > > >
> > > > > > > > >
> > > > > > > > > > Line 483-485, "our AnyTouch 2 general representation learning framework integrates multi-level objectives across all tiers, endowing it with comprehensive dynamic tactile perception capabilities.", the representation learning framework do not support the capabilities of tactile in manipulation. More complex task and solid experiments are required.
> > > > > > > > >
> > > > > > > > > > The paper present a tactile dataset and try to build hierarchical perception capabilities. The paper also present a tactile representation learning framework for optical tactile sensors, which fail to show the capabilities of tactile in manipulation.
> > > > > > > > >
> > > > > > > > > Thank you for your comments. We would like to clarify that **we have conducted extensive real-world manipulation experiments in both the main text and the appendix to evaluate the manipulation capabilities of our AnyTouch 2 representation learning framework.** These experiments cover four tasks of varying difficulty—Tactile Grasping, Whiteboard Wiping, USB Insertion, and Chip Moving. Across all four tasks, our AnyTouch 2 model consistently outperforms all baselines, demonstrating that its dynamic tactile perception capabilities are sufficiently strong to support higher manipulation performance, as shown in Figure 4 of the main text.
> > > > > > > > >
> > > > > > > > > In addition, we further include a challenging cross-sensor generalization experiment to evaluate whether the manipulation capabilities learned by our model can successfully generalize across different tactile sensors. Specifically, we evaluated two settings: (1) fine-tuning the policy network using GelSight Mini data but testing with a DIGIT sensor, and (2) fine-tuning with DIGIT data but testing on a GelSight Mini sensor. We compare our AnyTouch 2 model with the T3 baseline and both of the encoders are frozen during fine-tuning. The results are shown in Table Ⅶ.
> > > > > > > > >
> > > > > > > > > Table Ⅶ: Cross-sensor generalization results on the USB Insertion task.
> > > > > > > > >
> > > > > > > > > | **Method** **(Training Sensor)** | **USB Insertion** **(Test DG)** | **USB Insertion** **(Test Mini)** |
> > > > > > > > > | -------------------------------- | ------------------------------- | --------------------------------- |
> > > > > > > > > | T3 (DG)                          | 0.15                            | 0.05                              |
> > > > > > > > > | AnyTouch 2 (DG)                  | **0.35**                        | **0.15**                          |
> > > > > > > > > | T3 (Mini)                        | 0.05                            | 0.10                              |
> > > > > > > > > | AnyTouch 2 (Mini)                | **0.15**                        | **0.30**                          |
> > > > > > > > >
> > > > > > > > > The results show that when changing the sensor during the test time, our AnyTouch 2 model still outperforms the T3 baseline. This indicates that our model possesses stronger sensor-invariant manipulation capabilities than the existing method. We have added these results and the corresponding discussions in Appendix A.11 into the revised version. Thank you again for your comments.

---

> > > > > > > > > > ### Author Response · Authors · 2025-11-23
> > > > > > > > > >
> > > > > > > > > > > In caption of Figure 4, "Each dynamic model has a corresponding dynamic tier, which denotes the highest level of the training data and objectives used in the tactile dynamic pyramid, reflecting the model’s dy- namic perception capability." Can you specify what is "dynamic model", "dynamic tier", "dynamic pyramid" meaning in this context?
> > > > > > > > > >
> > > > > > > > > > Thank you for your questions. "Dynamic model" indicates the model that takes consecutive tactile frames as input. It can capture inter-frame deformation changes and consequently perform better on dynamic perception tasks such as manipulation. Each dataset used to train these models corresponds to a specific tier in our proposed Tactile Dynamic Pyramid ("dynamic pyramid"). The “dynamic tier” of a model indicates the highest tier among its training datasets in the pyramid, reflecting the model’s capability for dynamic tactile perception. We have provided a clearer description of these in Figure 4 in the revised version. Thank you again for your questions!
> > > > > > > > > >
> > > > > > > > > >
> > > > > > > > > >
> > > > > > > > > > > Incomplete in presentation. Some details on the policy training are missing.
> > > > > > > > > >
> > > > > > > > > > > For four real-world tasks, does the model directly evaluate on those tasks? If the model fine-tuned on downstream task specific data. please provide details on how you fine-tune the model on downstream tasks.
> > > > > > > > > >
> > > > > > > > > > Thank you for pointing this out. In all downstream tasks, we kept the tactile encoders frozen to more directly evaluate their perception and generalization capabilities. For all real-world manipulation tasks, we used a frozen ImageNet-pretrained ResNet-50 as the visual encoder and trained a UNet-based Diffusion Policy as the control policy.  The diffusion policy adopted UNet channel sizes of [128,256,512], a positional encoding size of 256, a kernel size of 5, and 8 GroupNorm groups. As the tactile encoder produces a large number of tokens, directly training the policy network on the full token sequence could bring unacceptable costs of GPU memory and time. Hence, we inserted a trainable attentive pooler between each tactile encoder and the diffusion policy. The pooler uses 30 learnable query tokens to extract information from the full tactile token sequence via cross-attention. These 30 pooled tokens then replace the full tactile token sequence as the input to the policy network and are concatenated with the visual features after flattening. We trained the policy network using the AdamW optimizer with a learning rate of 1e-4, for a total of 100 epochs and a batch size of 64. For each task, we randomly sampled 8 trajectories out of 50 as the validation set, and the model with the lowest validation loss was used for real-world evaluation. We have incorporated these details into Appendix A.6 the revised version. Thank you again for your constructive comments!

---

> ### Author Response · Authors · 2025-11-27
> **Thank You for Any Valuable Feedback**
>
> Dear Reviewer LnK5,
>
> We sincerely appreciate your valuable feedback and the time and effort you have devoted to evaluating our work. We kindly ask you to consider our responses and let us know whether they fully or partially address your concerns. We are open to further discussion if needed.
>
> Thank you once again for your valuable reviews. We are looking forward to hearing from you.
>
> Best regards,
>
> The authors

---

### Official Review · Reviewer_tU7z · 2025-11-01

**Soundness:** 3
**Presentation:** 3
**Contribution:** 4
**Rating:** 8
**Confidence:** 4

**Summary:**

This paper introduces a large tactile dataset that includes dynamic tactile data. In addition, the authors propose a general tactile representation learning framework that unifies tactile sensing, visual sensing, semantic information, actions, and forces. The representation is showcased on downstream tasks, including manipulation and offline evaluation.

**Strengths:**

* Fig. 1 clearly explains the role and position of the ToucHD dataset.
* The technical content is thorough, including pixel-level autoencoding, semantic tactile feature learning, and dynamic physical property learning.
* Experiments and benchmarking are comprehensive, covering diverse datasets, methods, and hardware.

**Weaknesses:**

* Although the paper is thorough, the method introduces many task objectives. Are all of them necessary? How does it perform on each individual task? Additional ablations and per-task performance would be convincing.
* Fig. 4 would be clearer with a consistent zero position; the current plot can be misleading.
* The training loss schedule (Eq. 7) appears complex. Does this imply that training the encoder is brittle?
* For readers less familiar with prior tactile datasets, a table summarizing their sizes would help contextualize the scale of ToucHD.

**Questions:**

* The training objective includes many tasks. Do the authors observe training instability? More details on training dynamics would be appreciated. Additionally, how are training hyperparameters (e.g., i_task) determined—trial and error or a principled procedure?
* More details on Section 5.3 would help. Does the diffusion policy also take visual images as input? If not, do the authors assume the gripper has already grasped the object at the beginning of the task?
* For training on a heterogeneous dataset like ToucHD, if a sampled data point lacks information needed to compute a particular loss (e.g., a force pair), how is the loss computed in that case?

---

> ### Author Response · Authors · 2025-11-23
>
> We would like to express our sincere appreciation for your constructive suggestions. We sincerely appreciate your comments that the role and position of our ToucHD dataset is clear, the technical content is thorough and experiments and benchmarking are comprehensive.
>
>
>
> > Although the paper is thorough, the method introduces many task objectives. Are all of them necessary? How does it perform on each individual task? Additional ablations and per-task performance would be convincing.
>
> Thank you for asking this question. We have moved the detailed ablation studies for each task into the main text and added additional ablation experiments and discussions in Appendix A.9. The experimental results show that each module in the model makes a positive contribution to the specific tactile perception capability it is designed to support. Together, these training objectives enable our model to become a general optical tactile representation model capable of supporting a wide range of tactile perception tasks. Thank you again for pointing this out!
>
>
>
> > Fig. 4 would be clearer with a consistent zero position; the current plot can be misleading.
>
> Thank you for pointing this out. In the original Figure 4, we selected different y-axis zero positions for each experiment to better highlight the performance gaps between models. To minimize any potential misleading effects while still emphasizing these differences, we have adjusted zero positions of the Tactile Grasping, Whiteboard Wiping and USB Insertion tasks to a unified value. The Chip Moving task, however, retains its original zero position because its success rates are generally lower than those of the other tasks. We have updated Figure 4 in the revised version. Thank you again for pointing this out!
>
>
>
> > For readers less familiar with prior tactile datasets, a table summarizing their sizes would help contextualize the scale of ToucHD.
>
> Thank you for your constructive comments. In fact, we have already provided statistical information in Appendix A.3 for each tactile dataset used to train AnyTouch 2, including the dataset name, its tier within our tactile dynamics pyramid, the modalities paired with touch, the types of sensors included, the total dataset size, and the actual dataset size used after dynamic redundancy filtering, as shown in Table 5 in the appendix. It is clear that our dataset holds a significant advantage over existing tactile datasets in both total tactile frame count and the number of frames actually used, highlighting the value and contribution of our dataset. Thank you again for pointing this out!

---

> > ### Author Response · Authors · 2025-11-23
> >
> > > The training loss schedule (Eq. 7) appears complex. Does this imply that training the encoder is brittle?
> >
> > > The training objective includes many tasks. Do the authors observe training instability? More details on training dynamics would be appreciated. Additionally, how are training hyperparameters (e.g., i_task) determined—trial and error or a principled procedure?
> >
> > Thank you for your valuable comments. Since our goal is to develop a general tactile model that can support a wide range of downstream tactile perception tasks, the training process must incorporate multiple objectives to comprehensively equip the model with such capabilities. Therefore, the training loss inevitably consists of multiple objectives. However, this does not mean that the training process is fragile. During our experiments, we conducted grid searches over the hyper-parameters $\lambda_{task}^{max} $  and  $i_{task}$, and the results are shown in Table Ⅰ, Ⅱ, Ⅲ, Ⅳ, Ⅴ and Ⅵ.
> >
> > Table Ⅰ: Hyper-parameter study on $\lambda_{Align}^{max}$.
> >
> > | $\lambda^{max}_{Align}$ | Cloth (GS) Acc(↑) | **Sparsh Force (DG)** RMSE(↓) | **Sparsh Force (Mini)** RMSE(↓) | **ToucHD Force (DG)** RMSE(↓) | **ToucHD Force (Mini)** RMSE(↓) |
> > | ----------------------- | ----------------- | ----------------------------- | ------------------------------- | ----------------------------- | ------------------------------- |
> > | 0.5                     | 41.97             | **621.95**                    | 204.98                          | 902.54                        | 1076.93                         |
> > | **1.0**                 | **42.31**         | 624.26                        | **202.14**                      | **894.32**                    | **1051.03**                     |
> > | 2.0                     | 40.13             | 644.81                        | 220.53                          | 1054.97                       | 1195.62                         |
> > | 4.0                     | 39.49             | 671.99                        | 235.64                          | 1096.87                       | 1265.71                         |
> >
> > Table Ⅱ: Hyper-parameter study on $\lambda_{Match}^{max}$.
> >
> > | $\lambda^{max}_{Match}$ | Cloth (GS) Acc(↑) | **Sparsh Force (DG)** RMSE(↓) | **Sparsh Force (Mini)** RMSE(↓) | **ToucHD Force (DG)** RMSE(↓) | **ToucHD Force (Mini)** RMSE(↓) |
> > | ----------------------- | ----------------- | ----------------------------- | ------------------------------- | ----------------------------- | ------------------------------- |
> > | 0.01                    | 42.01             | 631.75                        | 206.98                          | 913.89                        | 1084.42                         |
> > | **0.02**                | **42.31**         | **624.26**                    | **202.14**                      | **894.32**                    | **1051.03**                     |
> > | 0.05                    | 41.67             | 640.46                        | 210.93                          | 907.52                        | 1077.11                         |
> > | 0.1                     | 41.92             | 635.89                        | 215.67                          | 910.13                        | 1085.62                         |
> >
> > Table Ⅲ: Hyper-parameter study on $\lambda_{Force}^{max}$.
> >
> > | $\lambda^{max}_{Force}$ | Cloth (GS) Acc(↑) | **Sparsh Force (DG)** RMSE(↓) | **Sparsh Force (Mini)** RMSE(↓) | **ToucHD Force (DG)** RMSE(↓) | **ToucHD Force (Mini)** RMSE(↓) |
> > | ----------------------- | ----------------- | ----------------------------- | ------------------------------- | ----------------------------- | ------------------------------- |
> > | 0.05                    | 42.15             | 630.69                        | 210.82                          | 923.38                        | 1079.62                         |
> > | **0.1**                 | **42.31**         | **624.26**                    | **202.14**                      | **894.32**                    | 1051.03                         |
> > | 0.2                     | 41.71             | 645.93                        | 220.77                          | 900.90                        | **1012.45**                     |

---

> > > ### Author Response · Authors · 2025-11-23
> > >
> > > Table Ⅳ: Hyper-parameter study on $i_{Align}$.
> > >
> > > | $i_{Align}$    | Cloth (GS) Acc(↑) | **Sparsh Force (DG)** RMSE(↓) | **Sparsh Force (Mini)** RMSE(↓) | **ToucHD Force (DG)** RMSE(↓) | **ToucHD Force (Mini)** RMSE(↓) |
> > > | -------------- | ----------------- | ----------------------------- | ------------------------------- | ----------------------------- | ------------------------------- |
> > > | 20 (Epoch)     | 41.55             | 657.13                        | 232.35                          | 1036.99                       | 1231.94                         |
> > > | **30 (Epoch)** | **42.31**         | **624.26**                    | **202.14**                      | **894.32**                    | **1051.03**                     |
> > >
> > > Table Ⅴ: Hyper-parameter study on $i_{Match}$.
> > >
> > > | $i_{Match}$    | Cloth (GS) Acc(↑) | **Sparsh Force (DG)** RMSE(↓) | **Sparsh Force (Mini)** RMSE(↓) | **ToucHD Force (DG)** RMSE(↓) | **ToucHD Force (Mini)** RMSE(↓) |
> > > | -------------- | ----------------- | ----------------------------- | ------------------------------- | ----------------------------- | ------------------------------- |
> > > | 10 (Epoch)     | 41.24             | 632.98                        | 207.16                          | 909.88                        | 1072.65                         |
> > > | **20 (Epoch)** | **42.31**         | **624.26**                    | **202.14**                      | **894.32**                    | **1051.03**                     |
> > > | 30 (Epoch)     | 41.89             | 635.17                        | 205.72                          | 916.25                        | 1085.28                         |
> > >
> > > Table Ⅵ: Hyper-parameter study on $i_{Force}$.
> > >
> > > | $i_{Force}$    | Cloth (GS) Acc(↑) | **Sparsh Force (DG)** RMSE(↓) | **Sparsh Force (Mini)** RMSE(↓) | **ToucHD Force (DG)** RMSE(↓) | **ToucHD Force (Mini)** RMSE(↓) |
> > > | -------------- | ----------------- | ----------------------------- | ------------------------------- | ----------------------------- | ------------------------------- |
> > > | 10 (Epoch)     | 42.12             | 670.82                        | 225.76                          | 925.76                        | **1048.75**                     |
> > > | **20 (Epoch)** | **42.31**         | **624.26**                    | 202.14                          | **894.32**                    | 1051.03                         |
> > > | 30 (Epoch)     | 42.01             | 651.79                        | **196.09**                      | 936.41                        | 1206.84                         |
> > >
> > > The findings indicate that our model is not sensitive to these hyper-parameters: although minor fluctuations and noticeable peaks exist, within each parameter’s feasible range, our model consistently outperforms the baselines. We additionally present the full training loss curves for each task in the AnyTouch 2 model in Figure 17 in the appendix. It can be observed that the loss for each training objective decreases smoothly. We also observe that the Force Loss $\mathcal{L}\_{Force}$ and Matching Loss $\mathcal{L}_{Match}$ occasionally become zero for a few iterations after these tasks start. This behavior is expected and stems from how we designed the sampler to stabilize GPU memory usage, as described in Appendix A.5. Since the matching task requires feeding both positive and negative samples into the encoder simultaneously, we modified the sampler to fix the proportion of ToucHD (Force), ToucHD (Sim), and TacQuad samples used for matching in each batch. However, because the sampler prioritizes satisfying the matching sample ratio, a small number of batches (similar to “the last batch”) may end up containing no matching samples, resulting in zero Matching Loss and Force Loss. Importantly, this does not affect training because such batches are extremely rare, and when using multi-GPU training, the probability that all GPUs simultaneously receive a batch without these samples is negligible. We have included these results and the corresponding discussion in Appendix A.10 in the revised version. Thank you again for your valuable questions!

---

> > > > ### Author Response · Authors · 2025-11-23
> > > >
> > > > > More details on Section 5.3 would help. Does the diffusion policy also take visual images as input? If not, do the authors assume the gripper has already grasped the object at the beginning of the task?
> > > >
> > > > Thank you for your constructive comments. In fact, we have already provided a detailed description of these four real-world manipulation tasks in Appendix A.6. In all four manipulation tasks, we simultaneously recorded and used visual images from a third-person camera and tactile frames from the optical tactile sensor, as completing these tasks requires certain visual cues. In the tactile grasping and chip moving tasks, the gripper of the robotic arm does not initially grasp the object but instead maintains a certain distance from it. This is because determining tactile attributes and grasping fragile objects based on tactile inputs at different contact locations is itself the core challenge of these two tasks. In contrast, for the Whiteboard wiping and USB insertion tasks, the primary role of the tactile modality lies in the manipulations performed after the object has been grasped, rather than in the grasping action itself. Therefore, in these two tasks, the gripper starts by firmly holding the object to be manipulated. We have expanded this part of the discussion in Appendix A.6 in the revised version. Thank you again for pointing this out!
> > > >
> > > >
> > > >
> > > > > For training on a heterogeneous dataset like ToucHD, if a sampled data point lacks information needed to compute a particular loss (e.g., a force pair), how is the loss computed in that case?
> > > >
> > > > Thank you for pointing out this important point. During training, if a sample lacks the label required for a specific training objective, it is excluded from the loss computation for that objective. For example, if a sample does not have a paired text description, it will not be considered when computing the tactile-text contrastive loss within the batch (details are provided in Appendix A.7). Similarly, if a sample does not have a corresponding force label, it will not be fed into the force decoder. In summary, all loss terms that require additional labels are computed only on the corresponding available subset within each batch.
> > > >
> > > > In particular, the number of available samples for the cross-sensor matching and action matching tasks within each batch has a considerable impact on both loss computation and GPU memory usage, since completing these tasks requires feeding both positive and negative samples into the encoder simultaneously. Therefore, to stabilize the loss computation and memory consumption during training, we rewrote the dataset sampler to ensure that the proportion of samples involved in the matching tasks remains as consistent as possible across batches. We have provided a more detailed description of this aspect in Appendix A.5 in the revised version. Thank you again for pointing this out!

---

> > > > > ### Comment · Reviewer_tU7z · 2025-11-26
> > > > >
> > > > > My questions are thoroughly addressed! I wiil keep my rating

---

> > > > > > ### Author Response · Authors · 2025-11-26
> > > > > >
> > > > > > Thank you very much for your thoughtful comments and constructive feedback! Your suggestions are highly valuable for improving our work, and we truly appreciate the time and effort you dedicated to reviewing the paper.

---

### Official Review · Reviewer_nyMK · 2025-11-02

**Soundness:** 3
**Presentation:** 4
**Contribution:** 3
**Rating:** 8
**Confidence:** 4

**Summary:**

To address the problem that previous tactile datasets and models only focus on object-level features while lack temporal dynamics, this paper presents ToucHD, which is, to the best of my knowledge, the first large-scale dataset that systematically collects tactile data paired with actions and manipulation tasks. The paper also proposes AnyTouch 2, which is a unified tactile representation learning model that captures dynamics of various tactile sensors. Experiments on various benchmarks demonstrate strong performance across different tasks, ranging from object-level understanding to force-aware manipulation.

**Strengths:**

1. [Useful tactile dataset] To the best of my knowledge, this is the first work that captures large-scale tactile data on action-specific and manipulation tasks with a systematic pipeline. The combination of dynamic tactile perception and robotic manipulation is becoming more important as the tasks requires more dexterity and fine-grained control. The dataset may potentially be a foundation upon which future tactile robotic policies can be developed.

2. [Strong pre-trained encoder] As is shown in Figure 2, the paper introduces a training paradigm that includes multiple stages. The paradigm makes it possible to train the model on a combination of various datasets (as shown in Appendix Table 4), resulting in a large-scale training set that models both semantics and dynamics of diverse sensors. Results in Table 1 and Figure 4 also show the significant improvement of the proposed model on modeling dynamics when compared to previous baselines. The pretrained tactile encoder and potentially be a base model for future tactile and robotic research.

3. [Comprehensive technical details] The paper provides a very comprehensive appendix that includes technical details of both the data collection and model training, which makes it easy-to-understand.

**Weaknesses:**

1. [Lack of sensor-invariant evidence] Although claimed to be sensor-invariant (by performing contrastive learning on different sensors), there's no clear evidence showing that the learned represenation is generalizable across different tactile sensors. In practice, there are two major problems when using a pretrained tactile encoder: (i) is it generalizable across different sensors? and (ii) is it model performance robust when the gel pads of the sensors are replaced (since for some sensors, the tactile readings might be signiciantly changed even if the gel pad is replaced with a new one)? For the first problem, it would be interesting to freeze the pretrained encoder and train a decoder on a downstream task with one sensor, and then test if the decoder still works when we change to another sensor (as what is done in Contrastive Touch-to-Touch Pretraining [1]). For the second problem, one simple way to test it would be repeating the current evaluation with another gel pad.

2. [Lack of material diversity] Despite being diverse in dynamics, the proposed ToucHD dataset is mostly constructed by probing daily objects and 3D-printed indenters (as is shown in Appendix Table 2 and Figure 8). This might result in the lack of material diversity in the dataset. This also probably explains why AnyTouch 2 is underperforming AnyTouch 1 on the TAG benchmark (as shown in Table 1), since the TAG benchmark mainly evaluates the material understanding ability of the model. The model might potentially be further improved by (i) adding a material classification head to the pretraining stage and (ii) leveraging more tactile datasets that focus on capturing diverse materials (e.g. TAG and TaRF [2] both captures various materials in outdoor scenes).

3. [Grouping of atomic actions] As is explained in Line 285-287, the tactile videos are grouped into 8 atomic actions during the action matching stage. I'm curious about: (i) how to group the complicated manipulation actions into these atomic actions, and does it require large amounts of manual labelling? (ii) is it possible to perfectly represent the manipulation actions with only 8 atomic actions?


[1] Rodriguez, Samanta, Yiming Dou,  William van den Bogert, Miquel Oller, Kevin So, Andrew Owens, and Nima  Fazeli. "Contrastive touch-to-touch pretraining." In *2025 IEEE International Conference on Robotics and Automation (ICRA)*, pp. 5857-5863. IEEE, 2025.
[2] Dou, Yiming, Fengyu Yang, Yi Liu, Antonio Loquercio, and Andrew Owens. "Tactile-augmented radiance fields." In *Proceedings of the IEEE/CVF Conference on Computer Vision and Pattern Recognition*, pp. 26529-26539. 2024.

**Questions:**

In addition to the questions mentioned in the weaknesses section, here are several clarification questions that I'm curious about:

1. When collecting manipulation data, why do you choose to use UMI+left hand instead of fully using tele-operation with two grippers? Would this lead to bias in the captured data and limit its generaliability in pure robotic tasks?

2. Is there a specific reason for using a different robotic arm in the Chip Moving task (as is mentioned in Line 1102-1105), i.e., is the representation specific to different embodiments for different tasks?

3. Some of the material understanding benchmarks only require one single image as input (e.g. TAG), but multiple frames are used in the experiments shown in Appendix Table 7. What is the setting of this experiment?

4. Upon acceptance, will the proposed dataset and pretrained models be released to the public?

---

> ### Author Response · Authors · 2025-11-23
>
> We would like to express our sincere appreciation for your constructive suggestions. We sincerely appreciate your comments that our ToucHD dataset is useful, our AnyTouch 2 model is strong and the technical details are comprehensive.
>
>
>
> > [Lack of sensor-invariant evidence] Although claimed to be sensor-invariant ...... another gel pad.
>
> Thank you for your valuable suggestions. First, we would like to clarify that in both our offline benchmarks and real-world manipulation experiments, we used at least two different optical tactile sensors. Specifically, the offline benchmarks involve GelSight, DIGIT, and GelSight Mini sensors, while the real-world manipulation tasks use DIGIT and GelSight Mini. Moreover, in all downstream evaluations, our AnyTouch 2 encoder remains frozen and we only fine-tune the task-specific decoders. This setup already demonstrates the model’s ability to generalize across different tactile sensors. To further validate our model’s ability to extract sensor-invariant features, we conducted additional experiments on the USB Insertion task by switching the tactile sensor at test time, as you suggested. Specifically, we evaluated two settings: (1) fine-tuning the policy network using GelSight Mini data but testing with a DIGIT sensor, and (2) fine-tuning with DIGIT data but testing on a GelSight Mini sensor. We compare our AnyTouch 2 model with the T3 baseline and both of the encoders are frozen during fine-tuning. Since Contrastive Touch-to-Touch Pretraining is trained using **spatially aligned samples** ( collected at exactly the same contact location) from different sensors, while our model uses only **coarsely aligned samples** (collected at nearby contact locations) and additionally incorporates other training data sources, the training data used by the two approaches are not comparable. As a result, it is difficult to conduct a fair comparison, and therefore we do not include a comparison with their method. The results are shown in Table Ⅰ.
>
> Table Ⅰ: Cross-sensor generalization results on the USB Insertion task.
>
> | **Method** **(Training Sensor)** | **USB Insertion** **(Test DG)** | **USB Insertion** **(Test Mini)** |
> | -------------------------------- | ------------------------------- | --------------------------------- |
> | T3 (DG)                          | 0.15                            | 0.05                              |
> | AnyTouch 2 (DG)                  | **0.35**                        | **0.15**                          |
> | T3 (Mini)                        | 0.05                            | 0.10                              |
> | AnyTouch 2 (Mini)                | **0.15**                        | **0.30**                          |
>
> The results show that when changing the sensor during the test time, our AnyTouch 2 model still outperforms the T3 baseline. This indicates that our model possesses stronger sensor-invariant capabilities than T3. These capabilities primarily benefit from three factors: the integration of large-scale multi-sensor data during training, the unified background removal applied during preprocessing, and the explicit aggregation of representations from different sensors enabled by the multi-modal alignment and cross-sensor matching modules in our model (which follow the same core principles as Contrastive Touch-to-Touch Pretraining). However, we also acknowledge that relying solely on coarsely aligned samples for cross-sensor matching makes it difficult to maintain performance when switching sensors, especially compared with cross-sensor contrastive learning using spatially aligned samples.
>
> To further evaluate our model’s robustness to changes in the gel pads of the sensors, we also tested our model on the USB Insertion and Chip Moving tasks using GelSight Mini Sensor. After completing data collection and model training, we replaced the GelSight Mini’s gel pad only at test time. The experimental results are shown in Table Ⅱ.
>
> Table Ⅱ: Generalization results on gel pad changes on GelSight Mini sensor.
>
> | **Method** **(Training Gel Pad)** | **USB Insertion** **(Test Gel Pad 1)** | **USB Insertion** **(Test Gel Pad 2)** | **Chip Moving** **(Test Gel Pad 1)** | **Chip Moving** **(Test Gel Pad 2)** |
> | --------------------------------- | -------------------------------------- | -------------------------------------- | ------------------------------------ | ------------------------------------ |
> | AnyTouch 2 (Gel Pad 1)            | 0.30                                   | 0.30                                   | 0.80                                 | 0.75                                 |
>
> We observe that replacing the gel pad causes only a minor performance drop. This further demonstrates the sensor-invariant capability of our model. We have added these results and the corresponding discussions into Appendix A.11 in the revised version. Thank you again for your valuable suggestions!

---

> > ### Author Response · Authors · 2025-11-23
> >
> > > [Lack of material diversity] Despite being diverse in dynamics ...... various materials in outdoor scenes).
> >
> > Thank you for your constructive suggestions. We strongly agree with your insight that material diversity is important, as it plays a crucial role in the applicability of tactile perception models. We also acknowledge that the ToucHD (Force) subset of our ToucHD dataset contains only 3D-printed materials, while ToucHD (Mani) includes only a limited range of everyday object types. Therefore, we made lots of efforts to maximize this diversity when collecting ToucHD (Sim). We took full advantage of tactile simulation to generate data and easily scale the number of objects, incorporating over 1,000 objects to collect contact data. These objects cover more than 10 different material types across five major environments—household, office, video, industrial, and natural, surpassing the material diversity of several existing large-scale tactile datasets such as YCB-Slide (10 material types) and ObjectFolder Real (7 material types). We also acknowledge that despite this diversity, the realism limitations of tactile simulation and the finite variety of manipulation tasks still leave room for further improvement. Therefore, when training AnyTouch 2, we additionally incorporate material-rich datasets such as TAG, TVL, and TacQuad to further enhance material diversity as you suggested. When leveraging these material-rich data for pretraining, we agree that directly incorporating a material classifier is a reasonable and effective approach, as also demonstrated in prior work [1]. At the same time, material labels can be represented more comprehensively through textual descriptions, which can capture additional properties such as hardness, roughness, and contact location. Therefore, in AnyTouch 2, we use the multi-modal alignment module to align tactile samples with their paired text or images, enabling the model to understand richer material information. This can be seen as a more flexible extension of adding a material classification head, as you suggested.
> >
> > When collecting ToucHD (Sim) via tactile simulation, we also explored the feasibility of augmenting training data using neural representations from datasets such as ObjectFolder [2], TaRF [3], and Tactile Functasets [4] as you suggested. While these datasets can increase material diversity, they cannot directly render tactile images during dynamic contact, providing only large numbers of static images. Therefore, these data essentially belong to Tier 5 of the tactile dynamics pyramid, offering few advantages compared to tactile simulators that can render dynamic contact processes. Nonetheless, we believe that incorporating these static images into the model training could also be beneficial, and **we also believe an important future direction for tactile neural representations is to incorporate dynamic contact parameters and enable rendering of dynamic processes.** This also reflects the guiding values of our proposed tactile dynamics pyramid.
> >
> > In training AnyTouch 2, we used the TAG dataset for pretraining, just as AnyTouch 1 did. AnyTouch 1 places strong emphasis on material properties and adopts a relatively high multi-modal alignment strength, resulting in good performance on material recognition tasks. However, this comes at the cost of dynamic tactile perception, leading to significant performance disadvantages across tasks in Sparsh Bench, ToucHD Bench and real-world scenarios. In contrast, our proposed AnyTouch 2 maintains strong object-attribute understanding while placing greater emphasis on the model’s dynamic perception capabilities. Consequently, although AnyTouch 2 performs slightly worse than the material-focused AnyTouch 1 on the TAG material recognition task, it still clearly outperforms other baseline models and achieves the best dynamic perception performance overall. We have expanded these discussions in Appendix A.2.1 in the revised version. Thank you again for your constructive comments!
> >
> > [1] Zhao, J., Ma, Y., Wang, L., & Adelson, E. (2025, January). Transferable Tactile Transformers for Representation Learning Across Diverse Sensors and Tasks. In *Conference on Robot Learning* (pp. 3766-3779). PMLR.
> >
> > [2] Gao, R., Si, Z., Chang, Y. Y., Clarke, S., Bohg, J., Fei-Fei, L., ... & Wu, J. (2022). Objectfolder 2.0: A multisensory object dataset for sim2real transfer. In *Proceedings of the IEEE/CVF conference on computer vision and pattern recognition* (pp. 10598-10608).
> >
> > [3] Dou, Y., Yang, F., Liu, Y., Loquercio, A., & Owens, A. (2024). Tactile-augmented radiance fields. In *Proceedings of the IEEE/CVF Conference on Computer Vision and Pattern Recognition* (pp. 26529-26539).
> >
> > [4] Li, S., Rodriguez, S., Dou, Y., Owens, A., & Fazeli, N. (2025, May). Tactile Functasets: Neural Implicit Representations of Tactile Datasets. In *2025 IEEE International Conference on Robotics and Automation (ICRA)* (pp. 3219-3225). IEEE.

---

> > > ### Author Response · Authors · 2025-11-23
> > >
> > > > [Grouping of atomic actions] As is explained in Line 285-287, the tactile videos are grouped into 8 atomic actions during the action matching stage. I'm curious about: (i) how to group the complicated manipulation actions into these atomic actions, and does it require large amounts of manual labelling? (ii) is it possible to perfectly represent the manipulation actions with only 8 atomic actions?
> > >
> > > Thank you for your valuable questions. The definitions of the eight atomic actions in our ToucHD dataset: sliding in the four directions (up, down, left, right), clockwise and counterclockwise rotation, pressing, and leaving were **determined based on an extensive survey of robotic manipulation tasks**, particularly those involving tactile sensing. We found that, in most tasks [5,6,7,8,9,10,11], the types of tactile deformations are relatively structured and typically involve motions in a single direction—either along one of the six directions of the X, Y, and Z axes or along one of the two rotational directions on the sensor surface (i.e., translations parallel and perpendicular to the surface, and rotations around the surface normal). In some complex tasks [12,13], combinations of these eight atomic actions may occur—for example, sliding in a certain direction while pressing. Nevertheless, as the minimal fundamental action units, these eight atomic actions are sufficient. **The labels of these atomic actions exist only in ToucHD (Sim) and ToucHD (Force), and they can be obtained in a very simple manner without any manual annotation**. For ToucHD (Sim), the data are collected by predefining the relative position and motion trajectory between the object and the sensor surface in the tactile simulator. Thus, once the spatial trajectory corresponding to each atomic action is specified, the 30-frame dynamic tactile video for that action can be directly obtained based on its type. For ToucHD (Force), we ingeniously obtain the atomic action labels by leveraging the force data paired with the tactile frames. Specifically, we locate segments in each trail where the forces along the X, Y, and Z axes change smoothly, and determine the action type based on the direction of these changes. Since we did not perform rotational motions when collecting force data (to avoid damaging the sensor gel), the ToucHD (Force) subset does not contain clockwise or counterclockwise rotation samples. Since other datasets do not provide similar motion information, we cannot automatically determine the corresponding atomic action labels for their tactile samples. Therefore, we did not annotate atomic actions for the other datasets. However, we also acknowledge that atomic action labels for some datasets can be roughly obtained using certain image processing techniques—for example, by calculating the frame-to-frame Laplacian variance in press-only Tier 5 datasets to locate pressing and leaving segments. We have expanded this part of discussions in A.2.1 in the revised version. Thank you again for your valuable questions!
> > >
> > >
> > >
> > > [5] Lee, M. A., Zhu, Y., Zachares, P., Tan, M., Srinivasan, K., Savarese, S., ... & Bohg, J. (2020). Making sense of vision and touch: Learning multimodal representations for contact-rich tasks. *IEEE Transactions on Robotics*, *36*(3), 582-596.
> > >
> > > [6] Li, H., Zhang, Y., Zhu, J., Wang, S., Lee, M. A., Xu, H., ... & Wu, J. (2023, March). See, Hear, and Feel: Smart Sensory Fusion for Robotic Manipulation. In *Conference on Robot Learning* (pp. 1368-1378). PMLR.
> > >
> > > [7] Zhang, C., Hao, P., Cao, X., Hao, X., Cui, S., & Wang, S. (2025). Vtla: Vision-tactile-language-action model with preference learning for insertion manipulation. *arXiv preprint arXiv:2505.09577*.
> > >
> > > [8] Liu, F., Li, C., Qin, Y., Xu, J., Abbeel, P., & Chen, R. (2025). Vitamin: Learning contact-rich tasks through robot-free visuo-tactile manipulation interface. *arXiv preprint arXiv:2504.06156*.
> > >
> > > [9] Yu, K., Han, Y., Wang, Q., Saxena, V., Xu, D., & Zhao, Y. (2023). Mimictouch: Leveraging multi-modal human tactile demonstrations for contact-rich manipulation. *arXiv preprint arXiv:2310.16917*.
> > >
> > > [10] Hao, P., Zhang, C., Li, D., Cao, X., Hao, X., Cui, S., & Wang, S. (2025). Tla: Tactile-language-action model for contact-rich manipulation. *arXiv preprint arXiv:2503.08548*.
> > >
> > > [11] Jiang, S., Zhao, S., Fan, Y., & Yin, P. (2025). GelFusion: Enhancing Robotic Manipulation under Visual Constraints via Visuotactile Fusion. *arXiv preprint arXiv:2505.07455*.
> > >
> > > [12] Xue, H., Ren, J., Chen, W., Zhang, G., Fang, Y., Gu, G., ... & Lu, C. (2025). Reactive diffusion policy: Slow-fast visual-tactile policy learning for contact-rich manipulation. *arXiv preprint arXiv:2503.02881*.
> > >
> > > [13] Zhao, J., Kuppuswamy, N., Feng, S., Burchfiel, B., & Adelson, E. (2025). PolyTouch: A Robust Multi-Modal Tactile Sensor for Contact-rich Manipulation Using Tactile-Diffusion Policies. *arXiv preprint arXiv:2504.19341*.

---

> > > > ### Author Response · Authors · 2025-11-23
> > > >
> > > > > When collecting manipulation data, why do you choose to use UMI+left hand instead of fully using tele-operation with two grippers? Would this lead to bias in the captured data and limit its generaliability in pure robotic tasks?
> > > >
> > > > Thank you for your valuable questions. We agree that directly teleoperating two grippers to perform manipulation tasks is the least biased way to collect data. However, this approach also has several limitations. First, data collection through teleoperation is less efficient than using a handheld UMI device. In our experience, the total time required to collect a single trajectory with a UMI is nearly half that of teleoperation, which results in a substantial efficiency gain when collecting data at scale. Second, teleoperation does not provide direct force feedback. This can significantly affect performance in fine manipulation tasks, further reducing efficiency and even lowering the success rate. In contrast, when collecting data with a handheld UMI, contact forces are directly transmitted to the operator’s arm, allowing them to receive continuous real-time force feedback throughout the task. This helps improve the quality of task execution. However, after we modified the UMI device by adding two tactile sensors, the overall setup became bulkier, and using dual UMIs to collect data would make many tasks difficult to perform. Therefore, we switched to a UMI+hand collaboration setup for large-scale data collection, which is essentially a trade-off.
> > > >
> > > > As our work primarily focuses on tactile representation while vision serves only as an alignment modality, the negative effects and biases introduced by the UMI+hand collaborative setup are limited. Moreover, many existing studies [9,14,15,16] have shown that even human-hand manipulation data can help improve the generalization ability of robotic manipulation. We have added this discussion to Appendix A.2.2 of the revised version and will further explore this aspect in future work. Thank you again for your highly valuable questions!
> > > >
> > > > [14] Wang, B., Zhang, J., Dong, S., Fang, I., & Feng, C. (2024). Vlm see, robot do: Human demo video to robot action plan via vision language model. *arXiv preprint arXiv:2410.08792*.
> > > >
> > > > [15] Zhou, J., Ma, T., Lin, K. Y., Wang, Z., Qiu, R., & Liang, J. (2025). Mitigating the human-robot domain discrepancy in visual pre-training for robotic manipulation. In *Proceedings of the Computer Vision and Pattern Recognition Conference* (pp. 22551-22561).
> > > >
> > > > [16] Ye, S., Jang, J., Jeon, B., Joo, S. J., Yang, J., Peng, B., ... & Seo, M. Latent Action Pretraining from Videos. In *The Thirteenth International Conference on Learning Representations*.
> > > >
> > > >
> > > >
> > > > > Is there a specific reason for using a different robotic arm in the Chip Moving task (as is mentioned in Line 1102-1105), i.e., is the representation specific to different embodiments for different tasks?
> > > >
> > > > Thank you for pointing this out. We use the AGILEX Piper robotic arm for the Tactile Grasping, Whiteboard Wiping, and USB Insertion tasks because, as a lightweight arm mounted on an isomorphic master–slave platform, it allows us to conveniently collect manipulation data through teleoperation. However, it also has some drawbacks, such as limited precision and slight vertical jitter caused by its mechanical structure. Therefore, for tasks that require more precise manipulation, such as the Chip Moving task, we use the larger xArm 6 robotic arm and collect data via a space mouse. In addition, evaluating our model’s performance across different embodiments is also an important consideration. We have added this discussion to Appendix A.6 in the revised version. Thank you again for pointing this out!

---

> > > > > ### Author Response · Authors · 2025-11-23
> > > > >
> > > > > > Some of the material understanding benchmarks only require one single image as input (e.g. TAG), but multiple frames are used in the experiments shown in Appendix Table 7. What is the setting of this experiment?
> > > > >
> > > > > Thank you for pointing this out. Since our model can accept multi-frame tactile input, we input consecutive $N=4$ (or 2) frames $(T_{t-3},T_{t-2},T_{t-1},T_{t}$ (or $(T_{t-3},T_{t}$) to the model for both the TAG and Cloth material recognition tasks. Using a single frame as input is indeed sufficient for most material classifications, but continuous tactile changes may further enhance the understanding of some material properties, such as hardness. In addition, for the comparative experiments with static models that only accept single-frame input, we ensured a fair comparison by processing $N=2$ frames for these models. Specifically, the $N$ frames were temporally unfolded into a batch of $B\times N $ independent samples for the static model. The final prediction was then obtained by averaging the output features across all $N$ frames for each original sample. We have expanded on these details in Appendix A.5 in the revised version. Thank you again for asking this question!
> > > > >
> > > > >
> > > > >
> > > > > > Upon acceptance, will the proposed dataset and pretrained models be released to the public?
> > > > >
> > > > > Thank you for pointing this out. We will certainly release all resources including our code, dataset, model checkpoint, and 3D assets immediately upon acceptance.

---

### Author Response · Authors · 2025-11-23
**Response to All Reviewers**

We would like to express our sincere gratitude to the reviewers for their comprehensive and thoughtful reviews, constructive suggestions, and kind recognition of **the strong performance** of our **model** and its **potential to advance future research in robotics** (Reviewer nyMK and shzk), the **novelty and significance** of our **dataset** and the **tactile dynamic pyramid** (Reviewer nyMK, tU7z and shzk), the **clarity of our motivation and visualizations** (Reviewer tU7z, LnK5 and shzk), the **thoroughness of our experiments and technical details** (Reviewer nyMK, tU7z and shzk), and the **insightful contribution and benefit to to the tactile community** (Reviewer nyMK and shzk).

We list our responses to several key concerns here:

[**Sensor-invariant experiments**] To further evaluate our model’s cross-sensor generalization capability, we additionally conducted experiments that replaced the sensor and the sensor’s gel pads at test time. The results in Appendix A.11 show that under these settings, our method still outperforms the existing baseline, demonstrating that our model possesses strong sensor-invariant capabilities.

[**Ablation and hyper-parameter study**] To evaluate the stability of our model under hyper-parameter variations as well as the contribution of each component of AnyTouch 2 model and ToucHD dataset, we conducted extra ablation and hyper-parameter experiments. The results in Appendix A.9 and A.10 show that our model is robust to hyper-parameter choices, consistently outperforming the baselines across a wide range of settings. Moreover, every component of the model and dataset provides a positive contribution to the tactile perception capabilities it supports.

[**Additional visualizations**] We visualized the training loss curves of AnyTouch 2 in Appendix A.10 and the marker distribution of the non-planar GelStereo BioTip sensor during force collection in Appendix A.2.3. These visualizations further demonstrate the stability of our training process as well as the diversity of our dataset.

[**Release of the resources**] We will release all resources including our code, dataset, model checkpoint, and 3D assets immediately upon acceptance.

---

### Author Response · Authors · 2025-11-30
**Summary of Rebuttal**

Dear Area Chair and Reviewers,

We would like to express our sincere gratitude to the reviewers and the Area Chair for the time and effort in evaluating our work, as well as for the valuable feedback that helped improve the work. We are also grateful that the reviewers recognized **the strong performance** of our **model** and its **potential to advance future research in robotics** (Reviewer nyMK and shzk), the **novelty and significance** of our **dataset** and the **Tactile Dynamic Pyramid** (Reviewer nyMK, tU7z and shzk), the **clarity of our motivation and visualizations** (Reviewer tU7z, LnK5 and shzk), the **thoroughness of our experiments and technical details** (Reviewer nyMK, tU7z and shzk), and the **contribution and benefit to the tactile community of our work** (Reviewer nyMK and shzk).

During the discussion phase, we provided detailed responses to each reviewer, comprehensively covering all of their concerns. **Reviewer tU7z explicitly stated that our replies thoroughly addressed their questions.** We also provide the following summary of several shared concerns raised by the reviewers, along with our corresponding responses:

- **More comprehensive ablation studies on our Tactile Dynamics Pyramid, ToucHD dataset, and AnyTouch 2 model**

  We conducted extensive additional ablation studies to thoroughly examine how our Tactile Dynamics Pyramid guides model training and how different ToucHD subsets and shear force labels impact the performance of our AnyTouch 2 model across downstream tasks. The results demonstrate that the Tactile Dynamics Pyramid effectively guides the training paradigm, while both dataset subsets and shear force labels consistently enhance performance. We have added the relevant experiments and discussions to Appendix A.9.

- **More comprehensive hyper-parameter studies and training loss visualizations**

  We conducted additional hyper-parameter experiments to evaluate the stability of our model under hyper-parameter variations. The experimental results show that our model is robust to hyper-parameter choices, consistently outperforming the baselines across a wide range of settings. Moreover, the full training loss curves for each task and the overall objective in our AnyTouch 2 model demonstrate that the training process of our model is stable. We have added the relevant experiments, visualizations and discussions to Appendix A.10.

- **Cross-sensor generalization experiments**

  We additionally conducted experiments that replaced the sensor and the sensor’s gel pads at test time to evaluate our model’s cross-sensor generalization capability. The results show that under these settings, our method still outperforms the existing baseline, demonstrating that our model possesses strong sensor-invariant capabilities. We have added the relevant experiments and discussions to Appendix A.11.

We have also provided a more detailed explanation of the criteria of the tiered structure of our Tactile Dynamic Pyramid. We further clarify that the tiered structure is defined based on the data collection efforts, the types of actions, and the difficulty of obtaining labels. These discussions have been added to Section 3 and Appendix A.1.

The new experiments and analyses described above help to substantially enhance the quality of our work. We once again express our sincere gratitude to the Reviewers and Area Chair for their time, effort, valuable feedback and constructive suggestions.



Best regards,

The Authors of Submission 3806

---

### Meta-Review · Area_Chair_N2FU · 2026-01-05

**Summary:**

In the initial reviews, 3 reviewers recommended acceptance and 1 recommended rejection.

nyMK recommends acceptance, citing the strength of the dataset and the quality of the learned feature representation. They also asked for evidence that the representation is indeed sensor-invariant, raised concerns about the diversity of the materials, and raised questions about the action grouping. In the rebuttal, the authors clarify that the benchmarks cover a range of different touch sensors in real-world manipulation tasks and that the features are frozen for the downstream task. They follow the reviewer's suggestion and perform an experiment where an out-of-distribution touch sensor is used at test time, finding that the model performs well. They also discussed the challenges of improving material diversity and describe their attempts to address this, and how the atomic action schemes were derived. tU7z is also in favor of acceptance, based on the comprehensiveness of the benchmark/dataset and the reasonable technical approach. They asked for additional ablations for the pretraining method and learning loss schedule, questioning whether the complexity is necessary. In the rebuttal, the authors reported the results of varying the weight of different losses and clarify the real-world experiments. The reviewer responded to say that this addressed their issues. LnK5 recommends rejection and raises a number of concerns, including the evidence for the "tactile dynamic pyramid", the benefit of "Tier-1" data, conflating shear and normal force, performance on real-world tasks, the lack of a limitations section, and (more broadly) what they perceive to be overclaiming. The authors address this in the rebuttal in a number of ways, including A) clarifying the definition of different tiers in the pyramid, B) by conducting new experiments that train the model with different tiers of data, C) ablating the shear force labels. Reviewer shzk recommends acceptance, praising the structure of the dataset, but raises concerns around the simulation.

On balance, the AC weighs the benefits of the proposed dataset and representation learning method over the weaknesses raised by LnK5. However, the AC agress with the concerns raised by LnK5, particularly around the evidence for the claims around the "tactile dynamic pyramid" and the benefits of structuring the data this way, and urgest the authors to address these concerns in the revision (along with the other concerns raised by the reviewers, especially nyMK and LnK5).

**Reviewer Concerns:**

nyMK requested extra experimental evidence that the features generalize between sensors. I think that the new experiments (shown in Table 1) may partly address this concern. tU7z says in a comment that the rebuttal addressed their concerns. The rebuttal provides a number of different experiments to address the questions from LnK5 (see above), including experiments that mix different susbets of the dataset (to address concerns around the structure of the pyramid) and ablating different force labels. For shzk, the authors describe the elastomer–object contact simulation in more detail.

**Reviewer Scores:**

nyMK was in favor of acceptance. I see no reason to believe that the score would have decreased, given that the authors partly addressed the concern about generalization through new experiments. tU7z said that the rebuttal addressed their concerns. As mentioned above, the authors conduct a large number of experiments to address LnK5's concerns. I am not sure whether these experiments would result in the reviewer raising their score, given their large number of concerns, though it may. Reviewer shzk recommends acceptance, and I see no reason to believe that they would decrease or increase their score.

---

### Decision · Program_Chairs · 2026-01-26

Accept (Poster)